# Does a Hybrid Space-Aware Randomized Defense Improve Empirical and Certified Adversarial Robustness?

Joy Dhar [1]  Manish Kumar Pandey [2]  Behzad Bozorgtabar [3]  Nayyar Zaidi [4]  Wenyu Zhang [5]  Wei-Hong Li [6]  Tingting Mu [7]
Dwarikanath Mahapatra [8]  Mahsa Baktashmotlagh [9]  Trung Le [10]  Chen Chen [11]  Sajib Mistry [12]  Camila Gonzalez [13]
Samira Ebrahimi Kahou [14]  Lina Yao [15]  Piotr Koniusz [15][16]  Robert B Fisher [17]  Dinh Phung [10]  Bohyung Han [18]
Nuno Vasconcelos [19]  Pietro Lio [20]

## Abstract

We introduce *Hybrid Space-aware Stochastic Convolution Attention Noise* (`HySCAN`), a hybrid randomized defense that helps close the long-standing gap between *provable* robustness under $\ell_2$ certificates and *empirical* robustness against strong $\ell_\infty$ attacks, while maintaining strong generalization across diverse imaging benchmarks. `HySCAN` jointly explores complementary sources of stochasticity at both training and inference: (i) implicit weight-space randomness via stochastic-aware Random Weights, and (ii) explicit feature-space randomness via Stochastic Attention Noise Injection modules. By incorporating randomness at both the parameter and representation levels, `HySCAN` enables meaningful certified guarantees while improving empirical robustness in practice. Comprehensive experiments on diverse imaging datasets e.g., `CelebA`, `CIFAR-10` and `CIFAR-100`, `ImageNet-1k`, `HAM10000`, and `NIH Chest X-ray` demonstrate that `HySCAN` outperforms existing certified and empirical defenses, improving certified robustness by up to $\approx 9.6\%$ and empirical robustness by up to $\approx 5\%$ without reducing clean accuracy. Code- HySCAN

---

[1]Indian Institute Of Technology Ropar [2]RoentGen Health [3]Aarhus University [4]Deakin University [5]Center for AI Safety [6]University of Bristol [7]University of Manchester [8]Khalifa University of Science, Technology and Research [9]The University of Queensland [10]Monash University [11]University of Sheffield [12]Curtin University [13]Medizinische Universität Wien [14]University of Calgary [15]The University of New South Wales (UNSW) [16]Data6❤CSIRO [17]University of Edinburgh [18]Seoul National University [19]University of California, San Diego [20]University of Cambridge. Correspondence to: Joy Dhar <joy.22csz0003@iitrpr.ac.in>.

*Proceedings of the 43rd International Conference on Machine Learning*, Seoul, South Korea. PMLR 306, 2026. Copyright 2026 by the author(s).

## 1. Introduction

Deep vision architectures have achieved remarkable performance across diverse real-world applications, yet they remain vulnerable to adversarial perturbations[1] (Carlini & Wagner, 2017; Dhar et al., 2026a). This vulnerability is particularly concerning in high-stakes applications such as clinical decision support and fraud detection (Dhar et al., 2026a). In response, researchers have explored a variety of methods to enhance the robustness of deep neural networks. In particular, empirical defenses—most notably adversarial training (Madry et al., 2018; Ding et al., 2018; Shafahi et al., 2019; Sriramanan et al., 2021; Cheng et al., 2023)—aim to enhance robustness by iteratively optimizing on adversarially perturbed examples. However, such defenses are not fully trustworthy: they are frequently broken by stronger or more carefully crafted attacks (Carlini & Wagner, 2017; Yuan et al., 2021; Hendrycks et al., 2021; Duan et al., 2021; Li et al., 2023; Dhar et al., 2026a). These limitations have renewed interest in methods that offer *certified robustness*, i.e., formal guarantees that a classifier's prediction remains unchanged within a specified perturbation range (Raghunathan et al., 2018; Hao et al., 2022; Dhar et al., 2026a).

Among certification methods, Randomized Smoothing (`RS`) (Lécuyer et al., 2019; Cohen et al., 2019) has emerged as one of the most scalable approaches for modern vision architectures. `RS` yields $\ell_2$ robustness certificates by defining a *smoothed* classifier that aggregates predictions of a base classifier under i.i.d. Gaussian perturbations of the input at inference time (Xia et al., 2024). A key limitation, however, is the inherent accuracy–radius trade-off induced by a fixed noise scale: larger noise often increases certified radii but can degrade clean accuracy, whereas smaller noise preserves accuracy but certifies only small radii (Xia et al., 2024). To mitigate this, prior work has explored inference-time (often input-dependent) adaptation, including `RS`-specific approaches (Alfarra et al., 2022; Súkeník et al., 2022; Hong et al., 2022) as well as broader inference-

---

[1]Adversarial examples are modified input images by adding small perturbations that can cause misclassification with high success rates.

time heuristics (Croce et al., 2022). In practice, such defenses can face limitations under modern $\ell_\infty$ adversarial attack suites (e.g., AutoAttack (AA), and its APGD variants) (Croce & Hein, 2020), highlighting a persistent gap between certified guarantees and empirical robustness.

This gap becomes even more consequential when moving from natural-image to medical-imaging benchmarks. Several studies suggest that medical-domain pipelines may be more vulnerable to adversarial perturbations than their natural-image counterparts (Dong et al., 2024; Yao et al., 2021; Ma et al., 2021; Finlayson et al., 2019). A plausible contributing factor is that medical-imaging datasets can differ markedly in sample size, feature statistics, acquisition artifacts, and task structure (Dong et al., 2024). These observations motivate the following question:

> *Can we design a defense that simultaneously improves certified robustness and empirical robustness, and does so reliably across both natural-image and medical-vision benchmarks?*

To address this question, we propose **Hybrid Space-aware Stochastic Convolution Attention Noise (HySCAN)**, a randomized defense architecture that improves certified $\ell_2$ and empirical $\ell_\infty$ robustness by injecting *independently resampled, attention-modulated noise* into the convolutional layers during both training and inference. Crucially, our certification accounts for *all* sources of randomness—both the input Gaussian perturbations used by randomized smoothing and HySCAN's internal resampling—thereby preserving standard RS $\ell_2$ guarantees while strengthening robustness to strong $\ell_\infty$ attacks across natural-vision and medical-imaging benchmarks.

HySCAN replaces standard convolutional blocks with two complementary stochastic attention-conditioned modules: (i) *Random Weights with Attention Noise (RWAN)*, which applies structured, attention-gated perturbations to convolutional weights, thereby implicitly inducing heteroscedastic weight-space stochasticity concentrated on salient channels; and (ii) *Stochastic Attention Noise Injection (SANI)*, which injects attention-modulated perturbations into intermediate features to explicitly smooth the representation geometry while preserving discriminative cues. By distributing controlled stochasticity across both weights and representations, HySCAN preserves RS certificates and improves the certified–empirical robustness trade-off across diverse image benchmarks. The key contributions of this paper are:

1. We introduce HySCAN, a plug-and-play hybrid stochastic defense architecture that enables provable $\ell_2$ robustness certification while improving empirical robustness against strong $\ell_\infty$ adversarial attacks.
2. We propose two attention-modulated stochastic modules—RWAN (weight-space stochasticity via attention-gated perturbations) and SANI (feature-space stochastic-

ity via attention-scaled injection)—and show how their hybridization enables controlled smoothing of both parameters and representations.
3. We benchmark HySCAN against strong baselines on eight natural-image and medical-imaging datasets, and observe consistent improvements in both certified and empirical robustness.

## 2. Related Work

**Certified randomized defenses.** Randomized smoothing (RS) provides scalable $\ell_2$ robustness certification by converting a base classifier $f$ into a Gaussian-smoothed classifier

$$F(x) = \arg\max_y \mathbb{P}\left[f(x + \varepsilon) = y\right], \; \varepsilon \sim \mathcal{N}(0, \sigma^2 I) \quad (1)$$

whose prediction can be certified to remain invariant within an $\ell_2$ ball around (x) (Cohen et al., 2019). Prior methods strengthen RS-style certification by improving the noise stability of the base model, for example through noise-aware/consistency regularization (Jeong & Shin, 2020), denoising-based preprocessing (Carlini et al., 2023), or adversarial training tailored to the smoothing distribution (Salman et al., 2019). While classical RS injects randomness only at the input, more recent variants tighten certified radii by jointly introducing multiple (including internal) noise sources, e.g., ARS (Lyu et al., 2024), DRS (Xia et al., 2024), IRS (Ugare et al., 2023). In contrast, HySCAN incorporates stochasticity beyond input perturbations: stochastic-attention noise implicitly modulates weight-perturbation dynamics and explicitly injects representation noise for feature-level smoothing, while preserving RS-style $\ell_2$ certification. Prior HyCAS (Dhar et al., 2026a) uses Lipschitz-controlled hybrid streams with smoothing-induced feature-space stochasticity, but does not inject smoothing-induced stochasticity in filter space. In contrast, HySCAN learns a shared attention-noise module that couples RWAN weight-space and SANI feature-space stochasticity within a single randomized base predictor.

**Empirical Randomized Defenses.** Complementary to certified approaches, empirical randomized defenses—such as CTRW (Ma et al., 2023), RPF (Dong & Xu, 2023), PNI (He et al., 2019), and Learn2Perturb (Jeddi et al., 2020)—have largely focused on *empirical* robustness (often under $\ell_\infty$ threat models) by injecting stochasticity during training and/or inference. However, these defenses offer no formal worst case guarantees (Yang et al., 2022; Liu et al., 2021), and can be brittle against adversarial attacks that explicitly account for the defense's randomness over the internal noise (Athalye et al., 2018; Tramèr et al., 2020).

Existing defenses are often specialized for either certified $\ell_2$ or empirical $\ell_\infty$ robustness. HySCAN unifies both regimes by coupling RS-style certification with empirical adversarial robustness through internally resampled, attention-noise guided stochastic modules (i.e., RWAN, SANI) that

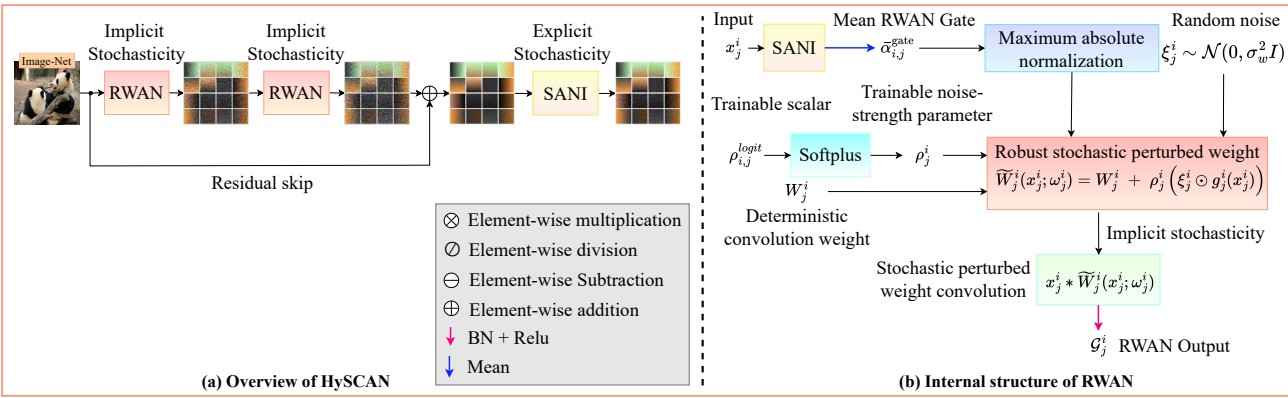

*Figure 1.* (a) Overview of the proposed `HySCAN` block, which replaces a standard convolutional block (e.g., the residual block in `ResNet` family (He et al., 2016)). `HySCAN` incorporates two complementary defense modules: (b) `RWAN`, which replaces each convolution layer and learns weight-space perturbations to induce implicit stochasticity, and `SANI`, which performs noisy feature-space representation learning to introduce explicit stochasticity. Together, they improve both certified and empirical adversarial robustness.

remain active during both training and inference. Our experiments show that replacing standard convolutional block with `HySCAN` incurs only a modest clean-accuracy drop, while improving certified $\ell_2$ guarantees (under weight-perturbation- and feature-noise guided smoothing) and strengthening robustness to strong $\ell_\infty$ adversarial attacks. Furthermore, while defenses are frequently evaluated in a single regime (e.g., natural vision datasets), with comparatively limited attention to medical imaging benchmarks, we show that `HySCAN` performs well across diverse imaging benchmarks (Tables 1–4 and Figs. 3–5). Further details appear in Appendix A.1.

## 3. `HySCAN`: Proposed Methodology

Randomization is a widely explored defense primitive because it makes the adversary's optimization problem stochastic (Ma et al., 2025). It can be leveraged to improve adversarial robustness during training and/or inference. However, existing randomized defenses often struggle to strengthen adversarial robustness consistently across threat models (e.g., certified and empirical adversarial attacks) and heterogeneous benchmarks (e.g., natural images and medical imaging) (Dong et al., 2024). A key reason is that, when injected in a naïve or feature-agnostic manner, *randomness does not reliably translate into robustness*. Many methods (e.g., `DRS`, `ARS`, `RPF`, `CTRW`) add noise only during training or inject noise at a single locus (e.g., only in weights or only in features), typically without conditioning on the input or the model's salient features. Such localized, unstructured perturbations can be attenuated by adaptive adversaries under both certified and empirical settings and can hinder the learning of informative representations, resulting in limited robustness gains.

To address these limitations, we propose **Hybrid Stochastic Convolution Attention Noise** (`HySCAN`) module, a plug-in defense block that replaces each convolutional block in standard `CNN` backbones with a hybrid design that couples *implicit* and *explicit* stochasticity via attention-conditioned noise. `HySCAN` consists of two complementary components: *(i)* **Random Weights with Attention Noise** (`RWAN`), which introduces *implicit* stochasticity by injecting attention-gated randomness into convolutional kernels (i.e., weight-space perturbations); and *(ii)* **Stochastic Attention Noise Injection** (`SANI`), which introduces *explicit* stochasticity by injecting attention-modulated randomness into intermediate features at the end of each block (i.e., feature-space perturbations). Together, `RWAN` and `SANI` distribute stochasticity across both parameters (i.e., weights) and representations, aiming to improve adversarial robustness while maintaining a leading clean-accuracy trade-off.

Let $x \in \mathbb{R}^{H \times W \times C}$ denote an input feature map with spatial dimensions $(H, W)$ and $C$ channels, and $y$ denote its label. We augment a base classifier $f_\Theta$ (parameters $\Theta$) by integrating `HySCAN` blocks by replacing each convolution block (e.g., residual block) throughout the network. Let $\varepsilon \sim \mathcal{N}(0, \sigma^2 I)$ denote isotropic Gaussian noise used for randomized smoothing (with dimension matching the network input). We use $\omega$ to denote the internal randomness instantiated by `RWAN` (including sampled weight noise and any auxiliary randomness used to form the `RWAN` gate), and $\psi$ for the internal randomness instantiated by `SANI` at the *feature-injection* stage. The resulting smoothed classifier is

$$F_\Theta(x) = \arg\max_{c \in \mathcal{Y}} \mathbb{P}_{\varepsilon, \omega, \psi}[f_\Theta(x + \varepsilon; \omega, \psi) = c], \quad (2)$$

i.e., $F_\Theta(\cdot)$ predicts the class most likely returned by the randomized network under the joint randomness from smoothing noise $\varepsilon$ and internal `HySCAN` randomness $(\omega, \psi)$. This has the following properties.

### 3.1. Certified guarantees and empirical robustness

Let $B_p(\epsilon) = \{\delta \in \mathbb{R}^d : \|\delta\|_p \le \epsilon\}$ denote the $\ell_p$ ball (vectorized inputs). Although the base network $f_\Theta(\cdot; \omega, \psi)$

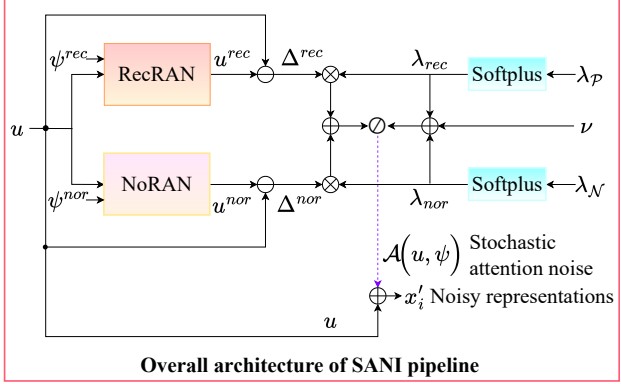

*Figure 2.* Overview of the Stochastic Attention Noise Injection (SANI) module with recursive RecRAN and non-recursive NoRAN branches for feature-space noise injection.

is randomized, the smoothed classifier $F_\Theta$ of Eq. (2) is defined by marginalizing over the joint randomness $(\varepsilon, \omega, \psi)$ and is therefore deterministic at the population level.

**Certified $\ell_2$ robustness guarantees** For $c \in \mathcal{Y}$, define the class probability induced by the randomized network in Eq. (2) as

$$p_c(x) = \mathbb{P}_{\varepsilon,\omega,\psi}[f_\Theta(x + \varepsilon; \omega, \psi) = c]. \quad (3)$$

Let $c_A = \arg\max_c p_c(x)$, and define $p_A = p_{c_A}(x)$ and $p_B = \max_{c \neq c_A} p_c(x)$. To confirm that HySCAN remains certifiable under randomized smoothing, we restate the standard RS guarantee (Cohen et al., 2019) to the joint-randomness predictor in Eq. (2), as formalized in Theorem 3.1.

**Theorem 3.1** (Randomized-smoothing certificate adopted from (Cohen et al., 2019) for HySCAN). *Consider the smoothed classifier $F_\Theta$ in Eq. (2) constructed with Gaussian smoothing noise $\varepsilon \sim \mathcal{N}(0, \sigma^2 I)$. If $p_A > p_B$, then the smoothed prediction is invariant to perturbations in an $\ell_2$ ball $B_2(r_2(x))$ of valid certified radius*

$$r_2(x) = \frac{\sigma}{2}\Big(\Phi^{-1}(p_A) - \Phi^{-1}(p_B)\Big), \quad (4)$$

*i.e.*

$$\forall \delta \in B_2(r_2(x)): \quad F_\Theta(x + \delta) = c_A, \quad (5)$$

*where $\Phi^{-1}$ is the inverse CDF of a standard normal random variable. This certificate remains valid for any internal HySCAN randomness $(\omega, \psi)$ because $p_A$ and $p_B$ are defined under the full joint probability $p_c(x) = \mathbb{P}_{\varepsilon,\omega,\psi}[\cdot]$.*

*Proof.* See App. A.3.2.

More details can be found App. A.3.1.

**Empirical $\ell_\infty$ adversarial robustness.** Since HySCAN is randomized at test time, $\ell_\infty$ adversarial attacks are evaluated with the model randomness active, i.e., RWAN/SANI randomness is resampled across attack iterations (each *stochastic forward pass* draws fresh noise $(\omega, \psi)$, and gradients are

obtained by backpropagating through that same realization); see (Eq. 2) and Algorithm 3. Let $U := (\omega, \psi) \sim \nu$ denote the induced joint internal randomness.

Fix an input $x$ and a perturbation $\delta \in B_\infty(\varepsilon_\infty)$ for the randomized base classifier $f_\Theta(\cdot; U)$ (as defined above). Let $a : \mathbb{R}^{|\mathcal{Y}|} \to \mathbb{R}$ be any differentiable attack objective applied to the classifier's differentiable output scores, and define the *single-pass* input gradient under a fresh draw $U$ as

$$\hat{g}_U(\delta) := \nabla_\delta \, a(f_\Theta(x + \delta; U)). \quad (6)$$

Define the mean direction after marginalizing joint randomness,

$$\mu(\delta) := \mathbb{E}_U[\hat{g}_U(\delta)], \quad u(\delta) := \frac{\mu(\delta)}{\|\mu(\delta)\|_2} \quad (\mu(\delta) \neq 0), \quad (7)$$

and the scalar alignment of a single-pass gradient with the mean direction,

$$Z_U(\delta) := \langle \hat{g}_U(\delta), \, u(\delta) \rangle. \quad (8)$$

**Proposition 3.2** (Directional gradient dispersion under joint randomness). *Assume $\mathbb{E}_U\|\hat{g}_U(\delta)\|_2^2 < \infty$ and $\mu(\delta) \neq 0$. Then the probability that a* single *stochastic-gradient step is directionally misaligned with the mean direction satisfies*

$$\mathbb{P}_U\big(Z_U(\delta) \leq 0\big) \leq \frac{\mathrm{Var}_U\big(Z_U(\delta)\big)}{\mathrm{Var}_U\big(Z_U(\delta)\big) + \|\mu(\delta)\|_2^2}. \quad (9)$$

*Moreover, writing $U = (\omega, \psi)$, this directional variance decomposes (law of total variance) as*

$$\mathrm{Var}_{\omega,\psi}\big(Z_{\omega,\psi}(\delta)\big) = \mathbb{E}_\omega\Big[\mathrm{Var}_\psi\big(Z_{\omega,\psi}(\delta) \mid \omega\big)\Big] + \mathrm{Var}_\omega\Big(\mathbb{E}_\psi[Z_{\omega,\psi}(\delta) \mid \omega]\Big). \quad (10)$$

*Proof.* See App. A.3.3.

Proposition 3.2 characterizes directional gradient dispersion under HySCAN's resampled internal randomness. Theorem A.1 (**see App. A.2**) offers a supervised-loss upper bound that formalizes the empirical $\ell_\infty$ robustness implication of this stochastic-gradient view.

### 3.2. Implementation

An overview of the implementation of HySCAN is given in **Fig. 1** and pseudo-code is provided in **App. A.8–A.9**. Each HySCAN block processes its input feature map through two cascaded modules: RWAN followed by SANI. Let $x_i$ denote the input to block $i$. RWAN is implemented as a residual branch with $J_i$ stochastic convolutional layers; we denote the resulting RWAN transformation by $\mathcal{G}_i^{J_i}(\cdot; \omega_i)$, where $\omega_i$ aggregates the sampled perturbations and gating randomness across these $J_i$ layers. SANI is denoted by $\mathcal{S}_i(\cdot; \psi_i)$, where $\psi_i$ parameterizes the sampled feature perturbations injected at the end of the block. The HySCAN output is

$$z_i(x_i) = \mathcal{S}_i\Big(x_i + \mathcal{G}_i^{J_i}(x_i; \omega_i); \psi_i\Big). \quad (11)$$

*Table 1.* Certified accuracy (%) of HySCAN and existing defenses on `ImageNet` (Deng et al., 2009) and CIFAR-10 (Krizhevsky, 2009). All methods are assessed at two noise levels. Bold denotes the best result.

| Methods | $\sigma$ | ImageNet | | | | | | | | CIFAR-10 | | | | | | | |
|---|---|---|---|---|---|---|---|---|---|---|---|---|---|---|---|---|---|
| | | Certified accuracy at predetermined $\ell_2$ radius $r$ (%) | | | | | | | | Certified accuracy at predetermined $\ell_2$ radius $r$ (%) | | | | | | | |
| | | 0.00 | 0.25 | 0.50 | 0.75 | 1.00 | 1.25 | 1.50 | 2.0 | 0.00 | 0.25 | 0.50 | 0.75 | 1.00 | 1.25 | 1.50 | 2.0 |
| RS (Cohen et al., 2019) | 0.25 | 67.1 | 48.7 | 0 | 0 | 0 | 0 | 0 | 0 | 75.3 | 60.2 | 43.4 | 26.1 | 0 | 0 | 0 | 0 |
| | 0.50 | 57.3 | 45.9 | 36.8 | 28.7 | 0 | 0 | 0 | 0 | 65.2 | 54.1 | 41.3 | 32.4 | 23.2 | 14.7 | 9.34 | 0 |
| ARS (Lyu et al., 2024) | 0.25 | 71.1 | 61.4 | 52.7 | 43.1 | 39.1 | 33.4 | 26.7 | 0 | 84.1 | 67.3 | 51.4 | 39.1 | 30.9 | 21.1 | 16.2 | 0 |
| | 0.50 | 68.1 | 58.7 | 50.3 | 43.4 | 39.1 | 34.5 | 30.6 | 22.4 | 78.4 | 63.7 | 50.2 | 38.9 | 31.8 | 23.3 | 19.7 | 8.47 |
| DRS (Xia et al., 2024) | 0.25 | 70.6 | 61.2 | 51.8 | 42.7 | 38.4 | 32.6 | 25.4 | 0 | 83.4 | 65.8 | 50.2 | 34.5 | 24.7 | 15.8 | 10.5 | 0 |
| | 0.50 | 67.6 | 58.2 | 49.6 | 42.8 | 35.6 | 33.2 | 29.8 | 21.3 | 78.1 | 62.1 | 48.7 | 35.8 | 24.5 | 17.9 | 12.9 | 4.6 |
| IRS (Ugare et al., 2023) | 0.25 | 68.4 | 58.5 | 46.2 | 38.7 | 32.1 | 19.3 | 10.8 | 0 | 78.6 | 63.2 | 47.5 | 30.8 | 19.6 | 10.3 | 5.72 | 0 |
| | 0.50 | 62.4 | 50.9 | 41.5 | 34.7 | 27.3 | 20.2 | 13.8 | 6.31 | 71.3 | 58.5 | 44.1 | 33.3 | 24.1 | 15.7 | 11.4 | 2.2 |
| HyCAS (Dhar et al., 2026a) | 0.25 | **72.3** | 63.9 | 55.6 | 46.4 | 40.7 | 35.2 | 29.7 | 5.42 | **85.4** | 70.1 | 56.7 | 44.3 | 36.5 | 29.6 | 22.9 | 8.52 |
| | 0.50 | **69.2** | 60.6 | 53.9 | 45.6 | 41.1 | 36.3 | 32.7 | 24.8 | **80.7** | 65.3 | 54.8 | 44.3 | 36.8 | 30.3 | 23.4 | 12.5 |
| **HySCAN (Ours)** | 0.25 | 71.9 | **65.2** | **56.7** | **47.9** | **42.3** | **37.1** | **30.7** | **7.15** | 85.2 | **70.4** | **57.3** | **45.5** | **37.9** | **30.7** | **24.1** | **9.94** |
| | 0.50 | 68.9 | **61.5** | **55.1** | **46.8** | **42.3** | **37.4** | **33.9** | **26.2** | 80.3 | **66.4** | **56.1** | **45.8** | **38.3** | **31.7** | **25.1** | **13.9** |

The block output is fed to the next block by setting $x_{i+1} = z_i(x_i)$. In this hybrid design, RWAN provides *implicit* stochasticity by perturbing effective convolutional filters, while SANI provides *explicit* stochasticity by injecting structured, attention-conditioned perturbations into the block representation.

### 3.2.1. RWAN: RANDOM WEIGHTS WITH ATTENTION NOISE

Randomizing convolutional weights can hinder adversarial optimization by making the effective filter stochastic across forward passes. However, uniformly injecting i.i.d. Gaussian noise to all kernel coefficients is feature-agnostic and largely stationary: attackers can often recover usable gradients by averaging over stochastic queries, while indiscriminate perturbations can degrade discriminative filters. RWAN instead induces structured, input-conditioned weight-space stochasticity, contributing to HySCAN's certified–empirical robustness trade-off. Using a SANI-derived attention–noise response from the current feature map, RWAN gates perturbation magnitudes *per input channel*, yielding *heteroscedastic* kernel randomization. Channels with stronger responses receive larger perturbations, whereas stable or less informative channels remain near-deterministic—making the randomness harder to average out by attackers while more preserving salient filters. **Fig. 1(b)** shows RWAN's internal structure. We next detail RWAN's design stages.

**Attention-gated stochastic convolution.** Consider RWAN layer $j$ in block $i$. Let $x_j^i \in \mathbb{R}^{H \times W \times C_j^i}$ denote its input feature map, and let the base kernel (i.e., standard convolution weights) be $W_j^i \in \mathbb{R}^{K_j^i \times K_j^i \times C_j^i \times \hat{C}_j^i}$. At each forward pass, RWAN samples Gaussian noise in *weight space*

$$\xi_j^i \sim \mathcal{N}(0, \sigma_w^2 I), \qquad \xi_j^i \in \mathbb{R}^{K_j^i \times K_j^i \times C_j^i \times \hat{C}_j^i}. \quad (12)$$

This noise is then gated using an attention-noise response

$\alpha_{i,j}^{\text{gate}}(x_j^i) \in \mathbb{R}^{H \times W \times C_j^i}$ produced by the SANI mechanism (Sec. 3.3). The response $\alpha_{i,j}^{\text{gate}}(x_j^i)$ is first compressed into a per-input-channel gate by averaging over batch ($N$) and spatial dimensions ($W, N$)

$$\bar{\alpha}_{i,j}^{\text{gate}} = \text{Mean}_{N,H,W}\left(\alpha_{i,j}^{\text{gate}}(x_j^i)\right) \in \mathbb{R}^{C_j^i}, \quad (13)$$

and applying max-absolute normalization:

$$g_j^i(x_j^i) = \frac{\bar{\alpha}_{i,j}^{\text{gate}}}{\max_c \left|\bar{\alpha}_{i,j}^{\text{gate}}[c]\right| + b}, \quad (14)$$

where $b = 10^{-6}$ ensures numerical stability. The gate $g_j^i$ is then broadcast to match the kernel shape $K_j^i \times K_j^i \times C_j^i \times \hat{C}_j^i$ and modulates the injected weight noise, to produce the stochastic RWAN weights

$$\widetilde{W}_j^i(x_j^i; \omega_j^i) = W_j^i + \rho_j^i \left(\xi_j^i \odot g_j^i(x_j^i)\right), \ \rho_j^i = \text{SP}(\rho_{i,j}^{\text{logit}}) \geq 0, \quad (15)$$

where $\odot$ is element-wise multiplication, $\rho_{i,j}^{\text{logit}}$ is a trainable noise-strength parameter[2] and SP($\cdot$) denotes SoftPlus activation. Finally, RWAN replaces the deterministic convolution weight $W_j^i$ with the robust stochastic perturbed weight $\widetilde{W}_j^i$ and performs convolution using $\widetilde{W}_j^i$, thereby injecting attention-gated implicit stochasticity into the weight space:

$$\mathcal{G}_j^i(x_j^i; \omega_j^i) = \phi\Big(\text{BN}\big(x_j^i * \widetilde{W}_j^i(x_j^i; \omega_j^i)\big)\Big), \quad (16)$$

where $*$ denotes convolution, BN($\cdot$) is batch normalization, and $\phi(\cdot)$ is ReLU. Composing these layers yields the RWAN residual branch $\mathcal{G}_i^{J_i}(\cdot; \omega_i)$ used in Eq. (11).

We next establish two structural properties of RWAN's induced weight perturbation $\Delta W_j^i = \widetilde{W}_j^i - W_j^i$: (i) the gate bounds its magnitude so injected noise cannot be amplified

_______

[2] We initialize $\rho_{i,j}^{\text{logit}}$ to a negative value so that $\rho_j^i$ starts small and increases only when beneficial.

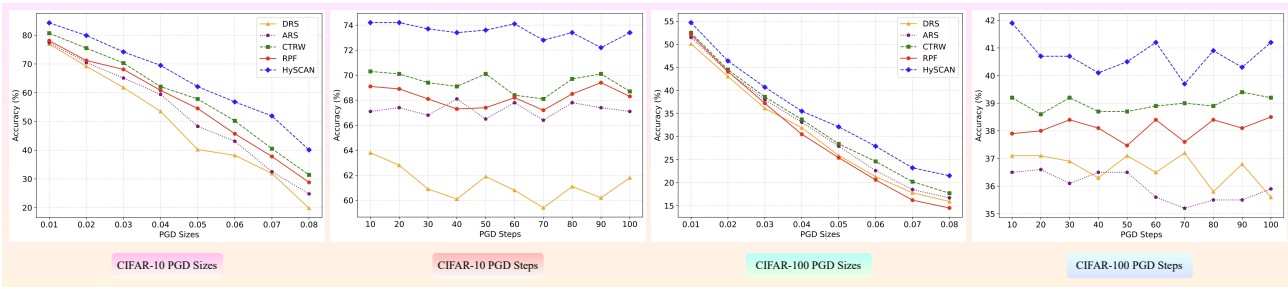

*Figure 3.* Empirical robustness of The HySCAN evaluation against stronger PGD attacks on CIFAR-10 and CIFAR-100.

beyond the learnable scale $\rho_j^i$ (Lemma 3.3); and (ii) under Gaussian base noise, $\Delta W_j^i$ is conditionally Gaussian with gate-modulated (heteroscedastic) variance (Proposition 3.4), enabling tractable reasoning under HySCAN's joint randomness. We start by bounding $\|\Delta W_j^i\|_F$.

**Lemma 3.3** (RWAN gate bounds weight-perturbation magnitude). *Let $g_j^i(x_j^i) \in \mathbb{R}^{C_j^i}$ denote the normalized gate in Eq. (14) with $b > 0$, and let $\Delta W_j^i = \widetilde{W}_j^i - W_j^i$ be the RWAN weight perturbation in Eq. (15). Then $\|g_j^i(x_j^i)\|_\infty \leq 1$ and*

$$\|\Delta W_j^i\|_F \leq \rho_j^i \|\xi_j^i\|_F. \tag{17}$$

*Proof.* See App. A.4.1.

The following proposition characterizes the conditional law of $\Delta W_j^i$ under Gaussian base noise.

**Proposition 3.4** (RWAN induces input-conditioned (heteroscedastic) Gaussian weight perturbations). *Assume $\xi_j^i \sim \mathcal{N}(0, \sigma_w^2 I)$ in Eq. (12) and $\xi_j^i \perp x_j^i$. Conditioned on $x_j^i$, the RWAN perturbation $\Delta W_j^i$ in Eq. (15) is Gaussian distribution. Moreover, for any kernel coefficient $\Delta W_j^i[q]$ indexed by $q \in (k_h, k_w, c, \hat{c}) \in \{1, \ldots, K_j^i\}^2 \times \{1, \ldots, C_j^i\} \times \{1, \ldots, \hat{C}_j^i\}$, and $\hat{c} \in \{1, \ldots, \hat{C}_j^i\}$,*

$$\mathbb{E}\left[\Delta W_j^i[q] \mid x_j^i\right] = 0, \ \text{Var}\left(\Delta W_j^i[q] \mid x_j^i\right) = (\rho_j^i)^2 \sigma_w^2 \left(g_j^i(x_j^i)[c]\right)^2 \tag{18}$$

*Finally,*

$$\mathbb{E}\left[\|\Delta W_j^i\|_F^2 \mid x_j^i\right] = (\rho_j^i)^2 \sigma_w^2 \|G_j^i(x_j^i)\|_F^2 \leq (\rho_j^i)^2 \sigma_w^2 K_j^{i,2} C_j^i \hat{C}_j^i. \tag{19}$$

*where $G_j^i(x_j^i)$ denotes the gate broadcast to the kernel shape.*

*Proof.* See App. A.4.2.

### 3.3. SANI: Stochastic Attention Noise Injection

**Motivation.** While RWAN diversifies weights by perturbing kernels, weight-space randomness alone does not explicitly smooth intermediate representations: the induced stochasticity can be partially averaged out by adversarial attacks, and may not sufficiently regularize representation geometry. SANI complements RWAN by injecting *explicit* feature-

space perturbations at the end of each HySCAN block, encouraging stochastic smoothing of intermediate features while preserving salient cues through attention conditioning. It also provides the attention-noise *responses* used by RWAN to gate weight perturbations, ensuring that weight-space noise remains structured rather than indiscriminate. We next discuss the main building blocks of SANI.

**Dual-branch attention-noise learner.** Given an input feature map $u \in \mathbb{R}^{H \times W \times C}$ (in our usage, either $u = x_j^i$ for per-layer gating or $u = x_i'$ for block-level injection), SANI produces stochastic attention noise $\mathcal{A}(u; \psi)$ with the two complementary stochastic branches of **Fig. 2**. This attention-noise is a normalized nonnegative mixture

$$\mathcal{A}(u; \psi) = \frac{\lambda_{\text{rec}} \Delta^{\text{rec}} + \lambda_{\text{nor}} \Delta^{\text{nor}}}{\gamma}, \ \ \gamma = \lambda_{\text{rec}} + \lambda_{\text{nor}} + \upsilon, \tag{20}$$

where $\upsilon = 10^{-6}$ ensures numerical stability, of two *residual increments* relative to $u$,

$$\Delta^{\text{rec}} = u^{\text{rec}} - u, \qquad \Delta^{\text{nor}} = u^{\text{nor}} - u. \tag{21}$$

$u^{\text{rec}}$ is the output of a Recursive Randomized Attention Noise (RecRAN) branch,

$$u^{\text{rec}} = \text{RecRAN}(u; \psi^{\text{rec}}), \tag{22}$$

which implements a recursive stochastic attention modulation. $u^{\text{nor}}$ is the output of Non-Recursive Randomized Attention Noise (NoRAN)

$$u^{\text{nor}} = \text{NoRAN}(u; \psi^{\text{nor}}), \tag{23}$$

a non-recursive branch that injects complementary channel-aware perturbations in a single pass. The implementation details of the two branches (i.e., RecRAN and NoRAN) are presented in **App. A.5 and A.6**. $\psi = (\psi^{\text{rec}}, \psi^{\text{nor}})$ collects the branch-specific internal randomness and are resampled per forward pass. This two branch architecture yields an adaptive trade-off: RecRAN captures *progressive* stochastic smoothing, while NoRAN contributes *complementary*, randomized, channel-aware noisy representation learning.

**Why dual-branch?** In SANI, RecRAN applies progressive smoothing that stabilizes features for robustness, but

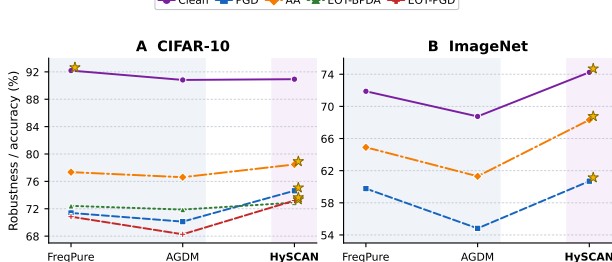

*Figure 4.* Empirical robustness (%) of HySCAN versus prior diffusion-based defenses (e.g., FreqPure (Pei et al., 2025) and AGDM (Lin et al., 2025)) on CIFAR-10 and ImageNet. CIFAR-10 is evaluated under PGD, AutoAttack (AA), EOT-PGD, and adaptive EOT-BPDA attacks; ImageNet under PGD and AA attacks.

it can over-smooth and limit perturbation diversity. NoRAN complements this with a non-recursive, single-pass, channel-aware heteroscedastic injection that improves diversity but lacks RecRAN's progressive invariance. We therefore jointly exploit both effects to preserve both properties and improve certified and empirical adversarial robustness (see Appendix A.11).

**Roles of SANI in HySCAN.** Within block $i$, RWAN contains $J_i$ layers indexed by $j$. Let $x_j^i$ denote the input to RWAN layer $j$ and $\hat{x}_j^i = \mathcal{G}_j^i(x_j^i; \omega_j^i)$ its output. The attention noise $\mathcal{A}(u; \psi)$ of Eq. 20 is used in two complementary places of HySCAN.

*(i) Implicit (**RWAN**) use, per-layer.* At each RWAN layer $j$, the *attention-noise response* produced by SANI is used to form the RWAN gate

$$\alpha_{i,j}^{\text{gate}}(\hat{x}_j^i) = \mathcal{A}(\hat{x}_j^i; \omega_{i,j}^{\text{gate}}) \in \mathbb{R}^{H \times W \times C}, \qquad (24)$$

where $\omega_{i,j}^{\text{gate}}$ denotes the random draws used inside $\mathcal{A}$ for gating; we subsume these draws into $\omega$ in Eq. (2). Crucially, $\alpha_{i,j}^{\text{gate}}$ is *not* added to the feature stream at layer $j$; it is only pooled to gate weight noise (Eq. (14)).

*(ii) Explicit (feature) use, per-block.* After completing the $J_i$ RWAN layers in block $i$, SANI injects an additive perturbation at the end of the block, thereby yielding an explicitly randomized intermediate representations. Let

$$x_i' = x_i + \mathcal{G}_i^{J_i}(x_i; \omega_i) \qquad (25)$$

denote the pre-SANI block output (ref. Eq. (11)). SANI injects an additive attention-modulated perturbation:

$$\hat{x}_i' = \mathcal{S}_i(x_i'; \psi_i) = x_i' + \alpha_i^{\text{inj}}, \qquad \alpha_i^{\text{inj}} = \mathcal{A}(x_i'; \psi_i), \quad (26)$$

where $\psi_i$ denotes the SANI randomness used for *feature injection* at block $i$ (this is the $\psi$ in Eq. (2)).

**Lemma 3.5** (SANI fusion forms a sub-convex nonnegative mixture). *Let $\mathcal{A}(u; \psi)$ be the SANI fused perturbation in Eq. (20) with $\lambda_{\text{rec}}, \lambda_{\text{nor}} \geq 0$ (Eq. (29)) and $\gamma = \lambda_{\text{rec}} + \lambda_{\text{nor}} + \upsilon$ for $\upsilon > 0$. Define weights $w_{\text{rec}} = \lambda_{\text{rec}}/\gamma$ and*

$w_{\text{nor}} = \lambda_{\text{nor}}/\gamma$. *Then $w_{\text{rec}}, w_{\text{nor}} \geq 0$ and $w_{\text{rec}} + w_{\text{nor}} \leq 1$, and therefore*

$$\mathcal{A}(u; \psi) = w_{\text{rec}}\Delta^{\text{rec}} + w_{\text{nor}}\Delta^{\text{nor}}, \qquad (27)$$

*i.e., SANI is a normalized nonnegative mixture of branch increments (with an explicit shrinkage due to $\upsilon$).*

*Proof.* See App. A.4.3.

**Proposition 3.6** (SANI fusion is scale-controlled under any convex norm). *Under the setting of Lemma 3.5, for any convex norm $\| \cdot \|$ (e.g., $\ell_2$ or Frobenius), the fused perturbation satisfies*

$$\|\mathcal{A}(u; \psi)\| \leq w_{\text{rec}}\|\Delta^{\text{rec}}\| + w_{\text{nor}}\|\Delta^{\text{nor}}\| \leq \max\{\|\Delta^{\text{rec}}\|, \|\Delta^{\text{nor}}\|\}. \qquad (28)$$

*Thus, SANI cannot increase the perturbation magnitude beyond the larger of the two branch increments.*

*Proof.* See App. A.4.4.

The mixture weights are parameterized via softplus:

$$\lambda_{\text{rec}} = \text{SoftPlus}(\lambda_\mathcal{P}) \geq 0, \quad \lambda_{\text{nor}} = \text{SoftPlus}(\lambda_\mathcal{N}) \geq 0, \qquad (29)$$

with learnable scalars $\lambda_\mathcal{P}, \lambda_\mathcal{N}$ (broadcast over $H \times W \times C$). Softplus enforces nonnegativity, while normalization by $\gamma$ controls perturbation scale and enables the model to learn the relative contributions of RecRAN and NoRAN without trivially increasing both weights.

### 3.4. Training objective.

HySCAN is trained under the *joint randomness* as the smoothed predictor in Eq. (2): Gaussian smoothing noise $\varepsilon \sim \mathcal{N}(0, \sigma^2 I)$ and internal HySCAN randomness $U := (\omega, \psi) \sim \nu$, where $\omega$ denotes RWAN's weight/gating randomness and $\psi$ denotes SANI's feature-injection randomness. Both $\varepsilon$ and $U$ are resampled on each *forward* pass. We apply a supervised loss $\ell(\cdot, \cdot)$ (e.g., cross-entropy) to the score vector output of the base classifier $f_\Theta(\cdot; U)$ (prediction is $\arg\max_k f_{\Theta,k}(\cdot; U)$). To preserve RS-compatibility for $\ell_2$ certification (Thm. 3.1) while improving empirical $\ell_\infty$ robustness, we minimize the hybrid objective

$$\min_\Theta \mathcal{L}_{\text{HySCAN}}(\Theta) = \mathbb{E}_{(x,y)\sim\mathcal{D}}\Big[\mathbb{E}_{\varepsilon,U}\, \ell\big(f_\Theta(x + \varepsilon; U), y\big)$$
$$+ \kappa \max_\delta \mathbb{E}_{\varepsilon,U}\, \ell\big(f_\Theta(x + \delta + \varepsilon; U), y\big)\Big]. \qquad (30)$$

where $\delta : \{\|\delta\|_\infty \leq \epsilon_\infty\}$ and $\kappa \geq 0$ controls the certified–empirical trade-off ($\kappa = 0$ gives RS-compatible training). In practice, the inner maximization in Eq. (30) is approximated with a multi-step first-order attack (e.g., PGD/APGD), using a fresh *forward-pass* draw of $U$ at each attack step.

## 4. Experimental Results

***The experimental setup—datasets, implementation details, and baseline descriptions—is provided in Appendix A.7.***

*Table 2.* Certified accuracy (%) of `HySCAN` and state-of-the-art defenses on `NIH-CXR` (Wang et al., 2017), `HAM10000` (Tschandl et al., 2018), and `CelebA` (Liu et al., 2015). All approaches are assessed under three noise levels. Bold indicates the best results.

| | | NIH-CXR | | | HAM10000 | | | CelebA | | |
|---|---|---|---|---|---|---|---|---|---|---|
| Approaches | $\sigma$ | $\ell_2$ radius $r$ | | | $\ell_2$ radius $r$ | | | $\ell_2$ radius $r$ | | |
| | | 0.0 | 0.50 | 1.0 | 0.0 | 0.50 | 1.0 | 0.0 | 0.50 | 1.0 |
| RS (Cohen et al., 2019) | 0.25 | 77.4 | 43.5 | 15.7 | 94.6 | 53.2 | 10.5 | 92.8 | 45.7 | 0 |
| | 0.50 | 73.3 | 39.9 | 21.8 | 89.3 | 52.1 | 12.2 | 87.7 | 47.8 | 10.5 |
| | 1.0 | 66.4 | 42.9 | 22.8 | 84.7 | 54.3 | 21.2 | 81.4 | 51.6 | 18.8 |
| ARS (Lyu et al., 2024) | 0.25 | 79.1 | 58.4 | 32.5 | 96.7 | 57.4 | 31.3 | 95.2 | 53.3 | 27.4 |
| | 0.50 | 74.9 | 54.7 | 33.3 | 91.9 | 55.1 | 32.8 | 91.3 | 53.9 | 30.4 |
| | 1.0 | 69.9 | 52.9 | 34.1 | 86.9 | 57.4 | 34.6 | 85.3 | 59.2 | 31.6 |
| HyCAS (Dhar et al., 2026a) | 0.25 | **81.6** | 61.9 | 38.6 | **97.2** | 60.5 | 35.4 | **96.8** | 58.1 | 33.7 |
| | 0.50 | **76.2** | 58.6 | 40.9 | **93.1** | 60.4 | 36.6 | **92.7** | 59.3 | 34.8 |
| | 1.0 | **71.7** | 60.6 | 41.4 | **88.2** | 61.9 | 38.5 | **87.7** | 62.3 | 36.9 |
| **HySCAN (Ours)** | 0.25 | 81.2 | **63.1** | **39.4** | 96.9 | **62.4** | **37.2** | 96.5 | **59.7** | **34.9** |
| | 0.50 | 76.1 | **59.6** | **42.3** | 92.8 | **61.2** | **37.9** | 92.5 | **60.7** | **36.1** |
| | 1.0 | 71.4 | **61.1** | **42.9** | 87.7 | **63.3** | **39.5** | 87.4 | **63.9** | **38.4** |

## 4.1. Certified Adversarial Robustness

**Natural vision benchmarks.** Tables 1–2 summarize certified accuracy on three natural-image benchmarks—`ImageNet` (Deng et al., 2009), `CIFAR-10` (Krizhevsky, 2009), and `CelebA` (Liu et al., 2015)—across the evaluated noise–radius pairs $(\sigma, r)$, and compare against prior randomized-smoothing defenses. `HySCAN`[3] achieves the highest certified accuracy across the evaluated nonzero certified radii, establishing a new state of the art in certified robustness, while maintaining clean accuracy at $r = 0$ comparable to the leading baseline. On `ImageNet` and `CIFAR-10`, `HySCAN` improves over the leading prior `ARS` (Lyu et al., 2024) baseline by up to $\approx 9.6$ percentage points for $r > 0$, and improves the nonzero-radius certified entries over the prior `HyCAS` (Dhar et al., 2026a) baseline by up to $\approx 1.9$ percentage points.

In the low-noise, high-radius regime $(\sigma, r) = (0.25, 2.0)$, `HySCAN` achieves higher certified accuracy than `HyCAS` on both `ImageNet` and `CIFAR-10`. With larger smoothing noise ($\sigma = 0.50$), `HySCAN` remains robust, achieving 42.3% and 38.3% certified accuracy at $r = 1.0$, and 26.2% and 13.9% at $r = 2.0$, on `ImageNet` and `CIFAR-10`, respectively. On `CelebA`, `HySCAN` reaches 59.7–63.9% at $r = 0.5$ and 34.9–38.4% at $r = 1.0$ across $\sigma \in \{0.25, 0.50, 1.0\}$. Overall, these results show that `HySCAN`'s gains are stable across datasets and noise–radius pairs, rather than being confined to a single benchmark or operating point, suggesting that the hybrid space-aware stochasticity induced by `RWAN` and `SANI` strengthens the `HySCAN`-integrated smoothed classifier without requiring a large clean accuracy sacrifice.

**Medical vision benchmarks.** `HySCAN` also general-

---

[3] `HySCAN` denotes our proposed randomized-defense stochastic base architecture throughout. For certified defense, we evaluate it using the `RS`-style smoothed classifier in Eq. (2), which gives the certified guarantees; for empirical defense, we train/evaluate it with the adversarial-training objective in Eq. (30).

izes beyond natural images, showing the same certified-robustness trend on `NIH-CXR` (Wang et al., 2017) and `HAM10000` (Tschandl et al., 2018) across $\sigma \in \{0.25, 0.50, 1.0\}$ in Table 2. At $(r, \sigma) = (1.0, 1.0)$, `HySCAN` achieves 42.9% / 39.5% certified accuracy on `NIH-CXR` / `HAM10000`, improving over `ARS` (Lyu et al., 2024) by +8.8 / +4.9 percentage points and over the prior `HyCAS` baseline by +1.5 / +1.0 percentage points. At smaller radii, `HySCAN` maintains high certified accuracy (e.g., 81.2%–96.9% on `NIH-CXR` and `HAM10000` at $\sigma = 0.25$), indicating that its gains in certified robustness do not require large clean-accuracy degradation. These results are consistent with structure-aware internal stochasticity: instead of injecting indiscriminate noise, `RWAN` and `SANI` concentrate attention-noise-induced stochasticity around salient weight- and feature-space responses.

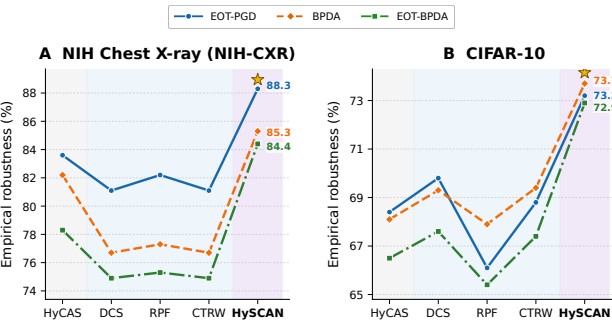

*Figure 5.* Empirical robustness (%) under `EOT-PGD` and adaptive attacks (e.g., `BPDA` and `EOT-BPDA` (Athalye et al., 2018; Tramèr et al., 2020)) on `NIH-CXR` and `CIFAR-10` benchmarks.

**Effect of the smoothing noise level $\sigma$.** As expected for randomized smoothing, increasing $\sigma$ traces out an accuracy–robustness frontier (Table 1). For `HySCAN`, raising $\sigma$ from 0.25 to 0.50 yields clear gains at larger radii with minimal changes at intermediate radii. On `ImageNet`, the certified accuracy at $r$=1.0 stays at 42.3%, whereas performance at $r$=2.0 climbs from 7.15% to 26.2%. On `CIFAR-10`, accuracy at $r$=0.75 changes only slightly (45.5% → 45.8%), while at $r$=2.0 it improves from 9.94% to 13.9%. These results highlight a tunable certified frontier in which $\sigma$ governs the robustness–accuracy trade-off; `RWAN` and `SANI` together help to shift this frontier upward.

## 4.2. Empirical Adversarial Robustness

Tables 3–4 report $\ell_\infty$ robust accuracy against `APGD-20` (Croce & Hein, 2020) and AutoAttack (`AA-20`) (Croce & Hein, 2020) at $\epsilon \in \{8, 16\}/255$. Across four medical imaging benchmarks—`NCT-CRC-HE-100K` (Kather et al.), `NIH-CXR` (Wang et al., 2017), `EyePACS` (Eye-PACS, 2015), and `HAM10000` (Tschandl et al., 2018)—`HySCAN` attains 55.2–90.7% robust accuracy while maintaining 75.1–91.5% clean accuracy. Relative to the strongest baseline (often `CTRW` (Ma et al., 2023)), `HySCAN` improves

*Table 3.* Empirical robustness (accuracy, %) against $\ell_\infty$ adversarial attacks (APGD-20 (Croce & Hein, 2020) and AA-20 (Croce & Hein, 2020)) on NCT-CRC-HE-100K (Kather et al.) (left) and NIH-CXR (Wang et al., 2017) (right) at perturbation strength $\epsilon \in \{8/255, 16/255\}$. Values in parentheses $(\cdot)$ indicate standard deviations.

| Method | NCT-CRC-HE-100K | | | | | NIH-CXR | | | | |
| --- | --- | --- | --- | --- | --- | --- | --- | --- | --- | --- |
| | Clean | APGD-20 | | AA-20 | | Clean | APGD-20 | | AA-20 | |
| | | 8/255 | 16/255 | 8/255 | 16/255 | | 8/255 | 16/255 | 8/255 | 16/255 |
| DRS (Xia et al., 2024) | 85.9 (2.25) | 75.1 (2.61) | 65.2 (3.82) | 73.5 (3.21) | 63.7 (3.94) | 83.9 (2.33) | 73.1 (2.41) | 62.9 (3.23) | 71.6 (3.12) | 61.9 (3.81) |
| ARS (Lyu et al., 2024) | 86.8 (2.14) | 75.9 (2.71) | 66.1 (3.52) | 74.6 (3.11) | 64.5 (3.73) | 84.8 (2.22) | 75.1 (3.01) | 64.7 (3.28) | 72.8 (3.11) | 62.8 (3.72) |
| AT (Madry et al., 2018) | 92.2 (1.82) | 77.8 (2.51) | 68.7 (3.12) | 76.3 (2.83) | 66.2 (3.61) | 89.1 (1.91) | 74.7 (2.52) | 66.9 (3.41) | 74.2 (2.93) | 64.1 (3.70) |
| DCS (Ma et al., 2025) | 90.3 (1.93) | 84.5 (2.72) | 73.0 (3.25) | 83.3 (2.74) | 71.6 (3.46) | 87.2 (2.05) | 82.4 (2.41) | 71.7 (3.21) | 81.7 (2.72) | 69.6 (3.45) |
| CTRW (Ma et al., 2023) | 90.4 (1.62) | 87.6 (2.29) | 76.7 (3.12) | 86.7 (2.44) | 75.2 (3.22) | 88.4 (1.73) | 85.1 (2.23) | 73.1 (3.22) | 84.5 (2.48) | 72.6 (3.41) |
| RPF (Dong & Xu, 2023) | 91.1 (1.71) | 86.1 (2.33) | 73.9 (3.33) | 84.2 (2.62) | 72.4 (3.41) | 88.4 (1.82) | 83.7 (2.49) | 71.9 (3.29) | 82.5 (2.71) | 70.8 (3.52) |
| HyCAS (Dhar et al., 2026a) | 91.3 (2.63) | 90.4 (2.82) | 79.3 (3.52) | 88.2 (2.63) | 76.7 (3.34) | 89.5 (1.64) | 88.6 (2.33) | 77.3 (3.14) | 86.9 (2.42) | 74.4 (3.33) |
| **HySCAN** (Ours) | **91.5** (2.25) | **90.7** (1.69) | **80.2** (2.61) | **89.5** (1.24) | **77.3** (2.98) | **90.1** (0.87) | **88.9** (1.75) | **78.1** (2.82) | **87.3** (1.98) | **75.1** (2.59) |

*Table 4.* Empirical robustness (accuracy, %) against $\ell_\infty$ adversarial attacks (APGD-20 and AA-20) on EyePACS (EyePACS, 2015) (left) and HAM10000 (Tschandl et al., 2018) (right) at $\epsilon \in \{8/255, 16/255\}$. $(\cdot)$ denotes the standard deviations.

| Method | EyePACS | | | | | HAM10000 | | | | |
| --- | --- | --- | --- | --- | --- | --- | --- | --- | --- | --- |
| | Clean | APGD-20 | | AA-20 | | Clean | APGD-20 | | AA-20 | |
| | | 8/255 | 16/255 | 8/255 | 16/255 | | 8/255 | 16/255 | 8/255 | 16/255 |
| DRS (Xia et al., 2024) | 71.9 (3.86) | 58.3 (2.61) | 47.4 (3.71) | 56.5 (2.92) | 45.7 (3.94) | 68.9 (3.28) | 53.4 (3.84) | 43.2 (3.43) | 51.6 (4.12) | 41.8 (3.81) |
| ARS (Lyu et al., 2024) | 72.9 (3.97) | 59.9 (2.91) | 48.8 (3.61) | 57.6 (2.94) | 46.5 (3.73) | 69.8 (3.22) | 53.9 (3.88) | 44.1 (3.13) | 52.7 (4.10) | 42.8 (3.71) |
| AT (Madry et al., 2018) | 78.2 (2.91) | 60.0 (2.72) | 50.1 (3.52) | 58.3 (2.83) | 48.2 (3.61) | 75.2 (2.94) | 56.1 (3.94) | 46.5 (3.75) | 54.2 (3.93) | 44.2 (3.80) |
| DCS (Ma et al., 2025) | 76.4 (2.94) | 66.8 (2.53) | 55.2 (3.68) | 65.3 (3.74) | 53.6 (3.46) | 73.2 (2.94) | 62.9 (3.48) | 51.7 (3.09) | 61.4 (3.72) | 49.5 (3.45) |
| CTRW (Ma et al., 2023) | 76.4 (2.84) | 70.1 (2.53) | 57.7 (3.35) | 69.7 (2.64) | 56.1 (3.31) | 74.3 (2.75) | 64.7 (3.33) | 52.8 (3.32) | 63.3 (3.48) | 51.2 (3.52) |
| RPF (Dong & Xu, 2023) | 77.1 (2.90) | 67.8 (2.57) | 56.1 (3.12) | 66.4 (2.73) | 54.4 (3.44) | 74.3 (2.86) | 64.1 (3.41) | 51.9 (3.42) | 62.6 (3.71) | 50.4 (3.58) |
| HyCAS (Dhar et al., 2026a) | 77.6 (2.79) | 72.6 (2.72) | 60.5 (3.43) | 71.8 (2.82) | 58.3 (3.32) | 74.6 (2.74) | 67.8 (3.43) | 55.3 (3.14) | 65.8 (3.42) | 53.1 (3.33) |
| **HySCAN** (Ours) | **78.3** (1.92) | **73.9** (1.57) | **62.2** (2.91) | **73.1** (2.25) | **60.5** (2.88) | **75.1** (1.84) | **68.3** (2.75) | **57.2** (2.86) | **66.9** (3.17) | **55.2** (2.62) |

robust accuracy by approximately $+2.1$ to $+5.0$ percentage points across both attacks and both $\epsilon$ budgets; compared to DCS (Ma et al., 2025), gains reach up to $+7.8$ points (e.g., EyePACS under AA-20 at $\epsilon = 8/255$). These results are consistent with HySCAN's joint stochasticity, which couples hybrid space-aware RWAN (weight-space randomness) with SANI (feature-space randomness), yielding stronger empirical robustness without a large clean-accuracy sacrifice.

Fig. 4 further evaluates HySCAN against recent diffusion-based defenses (e.g., FreqPure (Pei et al., 2025) and AGDM (Lin et al., 2025)). Across CIFAR-10 and ImageNet benchmarks, HySCAN achieves higher adversarial robustness under the evaluated $\ell_\infty$ attacks, including PGD (Madry et al., 2018), AA (Croce & Hein, 2020), EOT-PGD, and EOT-BPDA (Athalye et al., 2018; Tramèr et al., 2020). This gain is consistent with HySCAN's attention-noise-induced smoothing: RWAN induces weight-space stochasticity, while SANI induces feature-space stochasticity, exposing the adversary to jointly resampled randomness across both spaces. Fig. 5 confirms this behavior under adaptive stochastic attacks, where HySCAN remains ahead under EOT-PGD, BPDA (Athalye et al., 2018; Tramèr et al., 2020), and EOT-BPDA attacks on both NIH-CXR and CIFAR-10 datasets.

Fig. 3 further strengthen robustness under stronger PGD (Madry et al., 2018) configurations. On CIFAR-10, at $\epsilon = 0.08$, HySCAN retains an $\approx 9$ point advantage over the closest competitor. On CIFAR-100, HySCAN remains consistently above prior defenses as the number of PGD steps increases, preserving an $\approx 2$ point margin at 100 steps. This behavior is consistent with HySCAN's design principle: stochasticity is resampled and distributed across both weights and representations, forcing extended optimization to contend with a moving, structured randomness landscape rather than a single localized noise source. ***Further details of Fig. 3, additional experimental results, ablation studies, and an extended discussion with asymptotic analysis can be found in Appendices A.10, A.11, and A.12, respectively.***

# 5. Conclusion

We propose HySCAN, a hybrid randomized adversarial defense framework that injects internal randomness at two complementary levels: implicit weight-space perturbations and explicit feature-space noise smoothing. By jointly leveraging these mechanisms, HySCAN bridges certified $\ell_2$ guarantees with strong empirical robustness, yielding improved robustness without compromising performance on clean data. Experiments across heterogeneous imaging datasets show state-of-the-art certified accuracy and strong empirical robustness against powerful $\ell_\infty$ attacks. Future work includes developing tighter $\ell_\infty$ certificates, designing more efficient certification samplers, and integrating HySCAN into multi-modal pipelines.

## Impact Statement

This paper aims to improve the reliability of image-recognition models under adversarial perturbations, with potential benefit in safety-sensitive applications such as medical imaging. Its guarantees are limited to the evaluated threat models, and the added stochasticity increases computational cost during inference and certification.

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

# A. Appendix

## A.1. Extended Related Study: Randomized Certified and Empirical Defenses

**Certified defenses via randomized smoothing.** Randomized smoothing (RS) constructs a *smoothed classifier* $g$ by predicting the most likely label under isotropic Gaussian perturbations of the input, $x + \varepsilon$ with $\varepsilon \sim \mathcal{N}(0, \sigma^2 I)$, and then certifies an $\ell_2$ radius by lower-bounding the top-class probability and upper-bounding the strongest competitor (Xia et al., 2024). The resulting certificate is derived via a Neyman–Pearson style argument and yields a radius that grows with $\sigma$ and with the probability gap between the top and runner-up classes (e.g., (Cohen et al., 2019)). RS remains one of the most scalable certification approaches because it is architecture-agnostic and can be

*Table 5.* Scope of representative certified, empirical, and hybrid defences. A ✓ indicates that the property is explicitly addressed, or the domain is reported, in the original paper.

| Method | Certified | Empirical | Natural images | Medical images |
|---|---|---|---|---|
| RS (Cohen et al., 2019) | ✓ | | ✓ | |
| ARS (Lyu et al., 2024) | ✓ | | ✓ | |
| DRS (Xia et al., 2024) | ✓ | | ✓ | |
| IRS (Ugare et al., 2023) | ✓ | | ✓ | |
| Learn2Perturb (Jeddi et al., 2020) | | ✓ | ✓ | |
| PNI (He et al., 2019) | | ✓ | ✓ | |
| CAP (Xiang et al., 2023) | | ✓ | | ✓ |
| RPF (Dong & Xu, 2023) | | ✓ | ✓ | |
| CTRW (Ma et al., 2023) | | ✓ | ✓ | |
| **HySCAN (Ours)** | ✓ | ✓ | ✓ | ✓ |

applied to modern deep vision models with Monte-Carlo estimation at test time (Cohen et al., 2019; Xia et al., 2024).

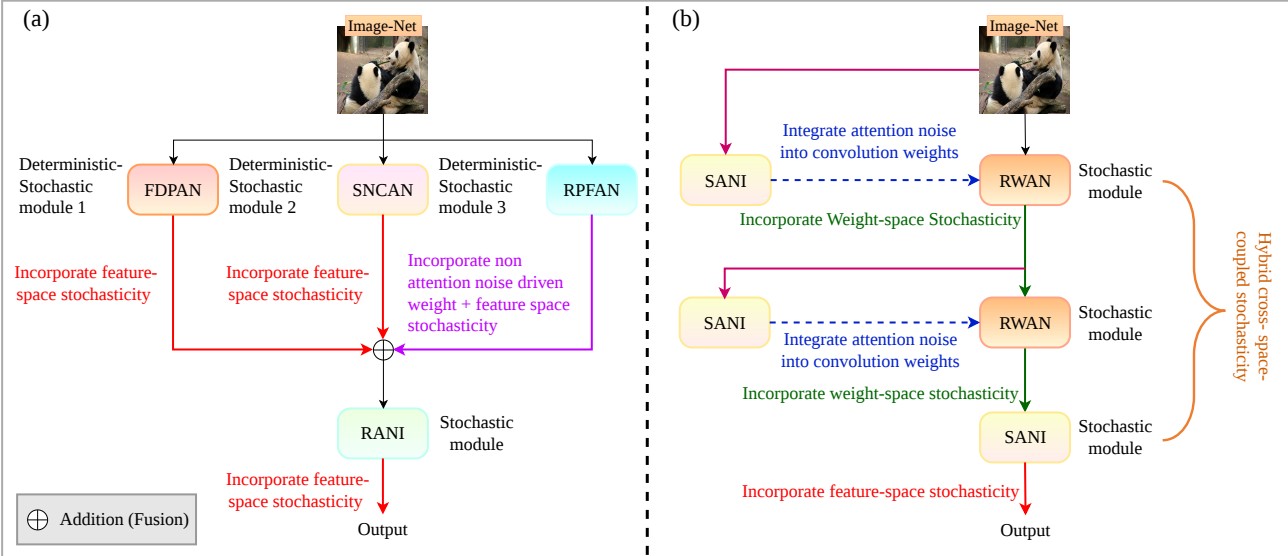

*Figure 6.* Methodological comparison between HyCAS (Dhar et al., 2026a) and our HySCAN. (a) HyCAS is a Lipschitz-controlled multi-stream architecture composed of FDPAN, SNCAN, and RPFAN, followed by RANI; in this design, weight-space stochasticity arises mainly from RPFAN, while feature-space stochasticity is injected separately by RANI. (b) HySCAN instead introduces a hybrid cross-space-coupled stochastic block inside the backbone, where RWAN applies SANI aware attention-conditioned, input-dependent heteroscedastic perturbations in weight space and SANI injects attention-modulated perturbations in feature space. Crucially, the same attention-noise learner is used in two complementary roles—per-layer RWAN gating and per-block SANI feature injection—so weight-space and feature-space stochasticity are explicitly shared and coordinated rather than added independently. This block-level coupling is the core methodological distinction between HySCAN and HyCAS.

**Training-time strengthening of smoothing.** Since RS guarantees ultimately depend on how stable the *base classifier* is under Gaussian corruption, a line of work focuses on objectives that directly align training with the smoothing distribution. (Salman et al., 2019) propose adversarially training the *smoothed* model by coupling Gaussian noise with worst-case perturbations during training, improving certified robustness at a given noise level. In a complementary direction, (Jeong & Shin, 2020) improve smoothed certification by encouraging prediction consistency across multiple noisy views of the same input, explicitly regularizing the base classifier toward noise-invariant decision boundaries.

**Tighter certificates beyond input-only smoothing.** Recent RS variants attempt to reduce conservativeness or improve the practicality of certification. Dual randomized smoothing (Xia et al., 2024) addresses the curse of dimensionality by constructing certificates through smoothing in lower-dimensional representations (e.g., via structured down-sampling/splitting) and combining the resulting guarantees. Incremental randomized smoothing (Ugare et al., 2023) amortizes certification by

reusing information from a previously certified model to certify a *modified* model (e.g., after pruning/quantization) with fewer additional samples, improving certification efficiency. Adaptive randomized smoothing (Lyu et al., 2024) extends the smoothing/DP view to *adaptive multi-step* randomized defenses and provides certificates against $\ell_\infty$-bounded perturbations for composed mechanisms, broadening the class of certifiable randomized pipelines beyond one-shot input corruption.

**Empirical defenses with learned stochasticity or domain priors.** On the empirical side, adversarial training (Dhar et al., 2026b) remains a dominant baseline: it directly optimizes worst-case loss under $\ell_\infty$ attacks (typically `PGD`) and can yield strong observed robustness but does not provide formal certificates (Madry et al., 2018; Dhar et al., 2025). A complementary family of defenses injects *trainable* stochasticity inside the network. Parametric Noise Injection (`PNI`) (He et al., 2019) learns layer-wise noise magnitudes jointly with model parameters (often within a min–max training loop), using learned randomness as regularization against adversarial perturbations. Learn2Perturb (Jeddi et al., 2020) similarly learns feature-space perturbation modules that act as trainable stochastic augmentation across layers. Architectural randomization has also been explored through sampling weights from learned distributions during training and inference (`CTRW`; (Ma et al., 2023)) or freezing part of early convolutions as random projection filters (`RPF`; (Dong & Xu, 2023)). Finally, domain-informed defenses such as contour attention preserving (`CAP`) (Xiang et al., 2023) harden medical-image classifiers by reinforcing clinically meaningful boundary cues.

**`HyCAS` versus `HySCAN`.** `HyCAS` (Dhar et al., 2026a) is the recent prior hybrid randomized defense, but its stochastic design and control mechanism differ from `HySCAN`. `HyCAS` combines Lipschitz-controlled hybrid streams with separately controlled filter- and feature-space stochastic components. In contrast, `HySCAN` uses a shared attention-noise learner: the same `SANI`-derived attention-noise response gates `RWAN`'s input-conditioned weight-space stochasticity and also drives block-level feature-space stochasticity (**see Fig. 6**). Thus, `HySCAN` couples both stochastic spaces inside a single randomized base predictor, rather than combining separately controlled stochastic modules.

`HySCAN` addresses this persistent gap, which remains between (i) *certified* robustness that is typically strongest/easiest to obtain in $\ell_2$ via smoothing, and (ii) *empirical* robustness evaluated against strong $\ell_\infty$ attacks (e.g., `PGD`, `APGD`, `AutoAttack`; (Madry et al., 2018; Croce & Hein, 2020)). `HySCAN` targets this mismatch by injecting *fresh, resampled* stochasticity *inside* the convolutional pipeline at both parameter and representation levels (`RWAN` and `SANI`), while retaining a randomized-smoothing style certificate that marginalizes over all randomness. This design differs from input-only smoothing and from single-location stochastic defenses by using attention-modulated, multi-level randomization to jointly improve certified $\ell_2$ guarantees and empirical $\ell_\infty$ robustness under modern evaluations. It is hereby noted that we *compare the properties for novel adversarial defense approach for enhancing adversarial robustness against existing baselines, demonstrating how `HySCAN` uniquely overcomes each identified research gap* (**see Table 5**).

### A.2. Extended Theoretical Analysis

**Theorem A.1** (Empirical $\ell_\infty$ robust-risk bound for `HySCAN`). *Fix a labeled example $(x, y)$ and an $\ell_\infty$ budget $\epsilon_\infty$. Let $U$, $f_\Theta$, and the supervised loss $\ell$ be as defined in Eq. (2) and Eq. (30), and let $\hat{y}_\Theta(z; U)$ denote the hard prediction of the randomized base predictor. Assume that $\ell$ is nonnegative and satisfies the loss-dominance condition*

$$\mathbf{1}\{\hat{y}_\Theta(z; U) \neq y\} \leq \frac{\ell(f_\Theta(z; U), y)}{\tau_\ell}$$

*for some $\tau_\ell > 0$ and all $(z, U)$; for the standard cross-entropy instantiation, one may take $\tau_\ell = \log 2$. Define the stochastic supervised risk*

$$L_{x,y}(\delta) = \mathbb{E}_U[\ell(f_\Theta(x + \delta; U), y)],$$

*its worst-case $\ell_\infty$ adversarial counterpart*

$$L_{x,y,\infty}^{\mathrm{adv}} = \sup_{\delta \in B_\infty(\epsilon_\infty)} L_{x,y}(\delta),$$

*and the local $\ell_\infty$ sensitivity*

$$G_\infty(x, y) = \sup_{\delta \in B_\infty(\epsilon_\infty)} \|\nabla_\delta L_{x,y}(\delta)\|_1.$$

*If $L_{x,y}$ is differentiable on $B_\infty(\epsilon_\infty)$, then*

$$\sup_{\delta \in B_\infty(\epsilon_\infty)} \mathbb{P}_U[\hat{y}_\Theta(x + \delta; U) \neq y] \leq \min\left\{1, \frac{L_{x,y,\infty}^{\mathrm{adv}}}{\tau_\ell}\right\} \leq \min\left\{1, \frac{L_{x,y}(0) + \epsilon_\infty G_\infty(x, y)}{\tau_\ell}\right\}.$$

*Proof.* Fix a labeled example $(x, y)$ and an arbitrary perturbation $\delta \in B_\infty(\epsilon_\infty)$. Throughout the proof, the probability and expectation are taken over the internal randomness $U$ of the randomized base predictor.

**Step 1: from loss dominance to stochastic error probability.** By the assumed loss-dominance condition, for every input $z$ and every realization of the internal randomness $U$,

$$\mathbf{1}\{\hat{y}_\Theta(z; U) \neq y\} \leq \frac{\ell(f_\Theta(z; U), y)}{\tau_\ell}.$$

Applying this condition to $z = x + \delta$ gives

$$\mathbf{1}\{\hat{y}_\Theta(x + \delta; U) \neq y\} \leq \frac{\ell(f_\Theta(x + \delta; U), y)}{\tau_\ell}.$$

Taking expectation over $U$ on both sides yields

$$\mathbb{E}_U\left[\mathbf{1}\{\hat{y}_\Theta(x + \delta; U) \neq y\}\right] \leq \frac{\mathbb{E}_U\left[\ell(f_\Theta(x + \delta; U), y)\right]}{\tau_\ell}.$$

The expectation of an indicator equals the probability of its event. Therefore,

$$\mathbb{P}_U\left[\hat{y}_\Theta(x + \delta; U) \neq y\right] \leq \frac{L_{x,y}(\delta)}{\tau_\ell}.$$

**Step 2: take the worst case over the $\ell_\infty$ ball.** Since the preceding inequality holds for every $\delta \in B_\infty(\epsilon_\infty)$, taking the supremum over the same perturbation set gives

$$\sup_{\delta \in B_\infty(\epsilon_\infty)} \mathbb{P}_U\left[\hat{y}_\Theta(x + \delta; U) \neq y\right] \leq \frac{\sup_{\delta \in B_\infty(\epsilon_\infty)} L_{x,y}(\delta)}{\tau_\ell}.$$

By the definition of $L_{x,y,\infty}^{\mathrm{adv}}$, this becomes

$$\sup_{\delta \in B_\infty(\epsilon_\infty)} \mathbb{P}_U\left[\hat{y}_\Theta(x + \delta; U) \neq y\right] \leq \frac{L_{x,y,\infty}^{\mathrm{adv}}}{\tau_\ell}.$$

Because the left-hand side is a probability, it is also bounded by one. Hence,

$$\sup_{\delta \in B_\infty(\epsilon_\infty)} \mathbb{P}_U\left[\hat{y}_\Theta(x + \delta; U) \neq y\right] \leq \min\left\{1, \frac{L_{x,y,\infty}^{\mathrm{adv}}}{\tau_\ell}\right\}.$$

**Step 3: control the adversarial supervised risk by local sensitivity.** It remains to upper-bound $L_{x,y,\infty}^{\mathrm{adv}}$. Fix any $\delta \in B_\infty(\epsilon_\infty)$ and consider the line segment $t\delta$ for $t \in [0, 1]$. Since $B_\infty(\epsilon_\infty)$ is convex, we have $t\delta \in B_\infty(\epsilon_\infty)$ for every $t \in [0, 1]$. By differentiability of $L_{x,y}$ on this ball and the fundamental theorem of calculus applied along the line segment from 0 to $\delta$,

$$L_{x,y}(\delta) - L_{x,y}(0) = \int_0^1 \langle \nabla_\delta L_{x,y}(t\delta), \delta \rangle \, dt.$$

Using Hölder's inequality with the dual pair $(\ell_1, \ell_\infty)$,

$$\langle \nabla_\delta L_{x,y}(t\delta), \delta \rangle \leq \|\nabla_\delta L_{x,y}(t\delta)\|_1 \|\delta\|_\infty.$$

Since $\delta \in B_\infty(\epsilon_\infty)$, $\|\delta\|_\infty \leq \epsilon_\infty$. Moreover, because $t\delta \in B_\infty(\epsilon_\infty)$, the definition of $G_\infty(x, y)$ gives

$$\|\nabla_\delta L_{x,y}(t\delta)\|_1 \leq G_\infty(x, y) \qquad \text{for all } t \in [0, 1].$$

Combining these inequalities,

$$L_{x,y}(\delta) - L_{x,y}(0) \leq \int_0^1 \epsilon_\infty G_\infty(x, y) \, dt = \epsilon_\infty G_\infty(x, y).$$

Thus, for every $\delta \in B_\infty(\epsilon_\infty)$,

$$L_{x,y}(\delta) \leq L_{x,y}(0) + \epsilon_\infty G_\infty(x, y).$$

Taking the supremum over $\delta \in B_\infty(\epsilon_\infty)$ yields

$$L_{x,y,\infty}^{\mathrm{adv}} = \sup_{\delta \in B_\infty(\epsilon_\infty)} L_{x,y}(\delta) \leq L_{x,y}(0) + \epsilon_\infty G_\infty(x, y).$$

**Step 4: combine the two bounds.** Substituting the sensitivity bound on $L^{\mathrm{adv}}_{x,y,\infty}$ into the probability bound from Step 2 gives

$$\sup_{\delta \in B_\infty(\epsilon_\infty)} \mathbb{P}_U\left[\hat{y}_\Theta(x+\delta;U) \neq y\right] \leq \min\left\{1, \frac{L^{\mathrm{adv}}_{x,y,\infty}}{\tau_\ell}\right\} \leq \min\left\{1, \frac{L_{x,y}(0) + \epsilon_\infty G_\infty(x,y)}{\tau_\ell}\right\}.$$

This proves the theorem. □

For the standard cross-entropy loss, $\ell_{\mathrm{CE}}(f_\Theta(z;U), y) = -\log p_y(z;U)$, where $p_y(z;U)$ is the softmax probability assigned to the true class. If $\hat{y}_\Theta(z;U) \neq y$, then some class $k \neq y$ has $p_k(z;U) \geq p_y(z;U)$, which implies $p_y(z;U) \leq 1/2$. Therefore,

$$\ell_{\mathrm{CE}}(f_\Theta(z;U), y) = -\log p_y(z;U) \geq \log 2,$$

and hence

$$\mathbf{1}\{\hat{y}_\Theta(z;U) \neq y\} \leq \frac{\ell_{\mathrm{CE}}(f_\Theta(z;U), y)}{\log 2}.$$

Thus, for cross-entropy, the loss-dominance condition holds with $\tau_\ell = \log 2$.

### A.3. Proofs for Certified Guarantees and Attack-Based Bounds

#### A.3.1. A GAUSSIAN SHIFT LEMMA ON PRODUCT SPACES

We first state an auxiliary lemma that formalizes how probabilities of measurable events change when the input to a Gaussian-smoothed classifier is shifted by a perturbation $\delta$. Crucially, we state it on a *product space* to explicitly accommodate internal randomness (e.g., $(\omega, \psi)$) that is resampled independently of the Gaussian noise.

**Lemma A.2** (Gaussian shift bound on a product space). *Let $\sigma > 0$ and $\delta \in \mathbb{R}^d$. Let $\nu$ be any probability measure on an auxiliary space $\Omega$ (e.g., the space of internal randomness). Define two distributions on $\mathbb{R}^d \times \Omega$:*

$$P = \mathcal{N}(x, \sigma^2 I) \times \nu, \qquad Q = \mathcal{N}(x + \delta, \sigma^2 I) \times \nu.$$

*Then for any measurable set $A \subseteq \mathbb{R}^d \times \Omega$, writing $p = P(A)$, we have*

$$Q(A) \geq \Phi\left(\Phi^{-1}(p) - \frac{\|\delta\|_2}{\sigma}\right), \tag{31}$$

*and also*

$$Q(A) \leq \Phi\left(\Phi^{-1}(p) + \frac{\|\delta\|_2}{\sigma}\right). \tag{32}$$

*Proof.* We prove the lower bound (31); the upper bound (32) follows by applying the same argument to $A^c$ and using symmetry of the Gaussian CDF.

**Step 1: reduce to a hypothesis testing extremal problem.** Consider binary hypothesis testing between

$$H_0 : (Z, U) \sim P, \qquad H_1 : (Z, U) \sim Q,$$

where $Z \in \mathbb{R}^d$ is the Gaussian variable and $U \in \Omega$ is the auxiliary random variable with distribution $\nu$. A test is any measurable function $\varphi : \mathbb{R}^d \times \Omega \to \{0, 1\}$. Interpreting $\varphi = 1$ as "reject $H_0$", the Type-I error and power are

$$\alpha(\varphi) = P(\varphi = 1), \qquad \beta(\varphi) = Q(\varphi = 1).$$

Fix a target acceptance probability under $P$, namely $P(A) = p$. If we set the *acceptance* region to be $A$ and the rejection region to be $A^c$, then $\alpha(\varphi) = P(A^c) = 1 - p$ and $\beta(\varphi) = Q(A^c) = 1 - Q(A)$. Thus, *minimizing $Q(A)$ among sets $A$ with $P(A) = p$* is equivalent to *maximizing $Q(A^c)$ among tests with fixed $P(A^c) = 1 - p$.*

**Step 2: compute the likelihood ratio and observe it ignores $U$.** Since $P$ and $Q$ share the same $\nu$ factor, their Radon–Nikodym derivative depends only on $Z$:

$$\frac{dQ}{dP}(z, u) = \frac{d\mathcal{N}(x + \delta, \sigma^2 I)}{d\mathcal{N}(x, \sigma^2 I)}(z) = \exp\left(\frac{\langle \delta, z - x \rangle}{\sigma^2} - \frac{\|\delta\|_2^2}{2\sigma^2}\right).$$

Hence, the likelihood ratio is a monotone function of the one-dimensional projection $\langle \delta, z - x \rangle$ and does *not* depend on $u$.

**Step 3: apply Neyman–Pearson to identify the extremal event.** By the Neyman–Pearson lemma, the most powerful level-$\alpha$ test rejects $H_0$ on a super-level set of the likelihood ratio. Equivalently, the rejection region that maximizes $Q(\text{reject})$ subject to $P(\text{reject}) = \alpha$ is a halfspace in the direction of $\delta$:

$$A_\star^c = \left\{ (z, u) : \langle \delta,\, z - x \rangle \geq t \right\} \quad \text{for some threshold } t \in \mathbb{R}.$$

Because $U$ cancels from the likelihood ratio, the optimal region does not need to depend on $u$. Therefore, among all $A$ with $P(A) = p$ (i.e., $P(A^c) = 1 - p$), the set minimizing $Q(A)$ is achieved (in the extremal sense) by such a halfspace.

**Step 4: compute $Q(A_\star)$ for the halfspace.** Let $v = \delta/\|\delta\|_2$ when $\delta \neq 0$ (the $\delta = 0$ case is trivial). Define the scalar random variable under $P$:

$$\hat{T} = \langle v, Z - x \rangle.$$

Under $P$, $\hat{T} \sim \mathcal{N}(0, \sigma^2)$; under $Q$, $\hat{T} \sim \mathcal{N}(\|\delta\|_2, \sigma^2)$. Choose the halfspace so that $P(A_\star) = p$, i.e.,

$$A_\star = \{(z, u) : \langle v, z - x \rangle \leq \sigma\, \Phi^{-1}(p)\}.$$

Then under $Q$,

$$Q(A_\star) = \mathbb{P}\left[ \hat{T} \leq \sigma\, \Phi^{-1}(p) \,\Big|\, \hat{T} \sim \mathcal{N}(\|\delta\|_2, \sigma^2) \right] = \Phi\!\left( \Phi^{-1}(p) - \tfrac{\|\delta\|_2}{\sigma} \right).$$

Since $A_\star$ minimizes $Q(A)$ among all measurable $A$ with $P(A) = p$, every such $A$ must satisfy $Q(A) \geq Q(A_\star)$, proving (31).

**Step 5: derive the upper bound.** Apply (31) to $A^c$:

$$Q(A^c) \geq \Phi\!\left( \Phi^{-1}(P(A^c)) - \tfrac{\|\delta\|_2}{\sigma} \right) = \Phi\!\left( \Phi^{-1}(1 - p) - \tfrac{\|\delta\|_2}{\sigma} \right).$$

Thus,

$$Q(A) = 1 - Q(A^c) \leq 1 - \Phi\!\left( \Phi^{-1}(1 - p) - \tfrac{\|\delta\|_2}{\sigma} \right) = \Phi\!\left( \Phi^{-1}(p) + \tfrac{\|\delta\|_2}{\sigma} \right),$$

where the last equality uses $\Phi^{-1}(1 - p) = -\Phi^{-1}(p)$ and $1 - \Phi(a) = \Phi(-a)$. This proves (32). $\qquad\square$

### A.3.2. PROOF OF THEOREM 3.1 (RANDOMIZED SMOOTHING CERTIFICATE)

*Proof of Theorem 3.1.* In the following stages, we delve into details of proofs regarding randomized smoothing certificate.

**Step 1: rewrite class probabilities as probabilities of measurable events.** Let the internal randomness space be $\Omega$ and write $U = (\omega, \psi) \sim \nu$ for its distribution. Let $Z = x + \varepsilon$ with $\varepsilon \sim \mathcal{N}(0, \sigma^2 I)$. For each class $c \in \mathcal{Y}$ define the measurable set

$$A_c = \{(z, u) \in \mathbb{R}^d \times \Omega : f_\Theta(z; u) = c\}.$$

Then, by definition,

$$p_c(x) = \mathbb{P}_{\varepsilon, U}\left[ f_\Theta(x + \varepsilon; U) = c \right] = \mathbb{P}_{(Z,U) \sim \mathcal{N}(x, \sigma^2 I) \times \nu}\left[ (Z, U) \in A_c \right] = P(A_c),$$

where $P = \mathcal{N}(x, \sigma^2 I) \times \nu$.

Now take any perturbation $\delta \in \mathbb{R}^d$. Define $Q = \mathcal{N}(x + \delta, \sigma^2 I) \times \nu$. Then

$$p_c(x + \delta) = \mathbb{P}_{\varepsilon, U}\left[ f_\Theta(x + \delta + \varepsilon; U) = c \right] = Q(A_c).$$

**Step 2: lower bound the shifted top-class probability.** Let $c_A = \arg\max_c p_c(x)$, $p_A = p_{c_A}(x)$. Apply Lemma A.2 to the set $A_{c_A}$ with $p = P(A_{c_A}) = p_A$. We obtain

$$p_{c_A}(x + \delta) = Q(A_{c_A}) \geq \Phi\!\left( \Phi^{-1}(p_A) - \tfrac{\|\delta\|_2}{\sigma} \right). \tag{33}$$

**Step 3: upper bound every shifted non-top class probability.** Fix any $c \neq c_A$. Apply the *upper* bound in Lemma A.2 to $A_c$ with $p = P(A_c) = p_c(x)$:

$$p_c(x + \delta) = Q(A_c) \leq \Phi\Big(\Phi^{-1}(p_c(x)) + \tfrac{\|\delta\|_2}{\sigma}\Big). \tag{34}$$

By definition $p_B = \max_{c \neq c_A} p_c(x)$, so $p_c(x) \leq p_B$ for all $c \neq c_A$. Since $\Phi^{-1}(\cdot)$ and $\Phi(\cdot)$ are monotone increasing, (34) implies

$$p_c(x + \delta) \leq \Phi\Big(\Phi^{-1}(p_B) + \tfrac{\|\delta\|_2}{\sigma}\Big) \qquad \text{for all } c \neq c_A. \tag{35}$$

**Step 4: enforce strict dominance at the perturbed point.** If $\|\delta\|_2 < r_2(x)$ where $r_2(x)$ is defined in Eq. (4), then

$$\tfrac{\|\delta\|_2}{\sigma} < \frac{1}{2}\Big(\Phi^{-1}(p_A) - \Phi^{-1}(p_B)\Big).$$

Rearranging gives

$$\Phi^{-1}(p_A) - \tfrac{\|\delta\|_2}{\sigma} > \Phi^{-1}(p_B) + \tfrac{\|\delta\|_2}{\sigma}.$$

Applying the monotonicity of $\Phi$ yields

$$\Phi\Big(\Phi^{-1}(p_A) - \tfrac{\|\delta\|_2}{\sigma}\Big) > \Phi\Big(\Phi^{-1}(p_B) + \tfrac{\|\delta\|_2}{\sigma}\Big).$$

Combining with (33) and (35) gives

$$p_{c_A}(x + \delta) > p_c(x + \delta) \qquad \text{for all } c \neq c_A.$$

Therefore $c_A$ remains the unique maximizer of $p_c(x + \delta)$, and thus

$$F_\Theta(x + \delta) = c_A.$$

This proves the invariance for all $\delta$ with $\|\delta\|_2 < r_2(x)$.

**Boundary case $\|\delta\|_2 = r_2(x)$.** At $\|\delta\|_2 = r_2(x)$ the above inequalities become non-strict when using only the bounds. In practice one may state the guarantee for the open ball, or assume a fixed deterministic tie-breaking rule. The standard randomized-smoothing literature typically states robustness for $\|\delta\|_2 < r_2(x)$; this is the core certified guarantee. $\qquad\square$

A.3.3. PROOF OF PROPOSITION 3.2 (DIRECTIONAL GRADIENT DISPERSION UNDER JOINT RANDOMNESS)

*Proof.* Fix input $x$ and $\delta \in B_\infty(\varepsilon_\infty)$. Throughout, the *only* randomness is HySCAN's internal test-time randomness $U = (\omega, \psi) \sim \nu$ (cf. Eq. (2) and Alg. 3), which is resampled on each stochastic *forward* pass; the gradient $\hat{g}_U(\delta)$ is obtained by backpropagating through that same realized forward pass. (Resampling the defense randomness across attack iterations is standard for evaluating stochastic defenses under white-box adversarial attacks; see, e.g., (Athalye et al., 2018; Tramèr et al., 2020).) This can be proved using the following properties.

**Step 0 (finite variance is well-defined).** By assumption, $\mu(\delta) \neq 0$, hence the normalized mean direction $\hat{\mu}(\delta) = \mu(\delta)/\|\mu(\delta)\|_2$ is well-defined and deterministic (it depends on $\delta$ only through the expectation over $U$). Since $\|\hat{\mu}(\delta)\|_2 = 1$ and $Z_U(\delta) = \langle \hat{g}_U(\delta), \hat{\mu}(\delta)\rangle$, the Cauchy–Schwarz inequality (Garling, 2005) gives

$$|Z_U(\delta)| = |\langle \hat{g}_U(\delta), \hat{\mu}(\delta)\rangle| \leq \|\hat{g}_U(\delta)\|_2 \|\hat{\mu}(\delta)\|_2 = \|\hat{g}_U(\delta)\|_2. \tag{36}$$

Therefore $Z_U(\delta)^2 \leq \|\hat{g}_U(\delta)\|_2^2$ almost surely, and $\mathbb{E}_U[Z_U(\delta)^2] \leq \mathbb{E}_U\|\hat{g}_U(\delta)\|_2^2 < \infty$. Consequently, $\mathrm{Var}_U(Z_U(\delta)) < \infty$ and the Cantelli's concentration inequality (Boucheron et al., 2003) in Step 2 applies.

**Step 1 (mean alignment equals $\|\mu(\delta)\|_2$).** Because $\hat{\mu}(\delta)$ is deterministic, linearity of expectation yields

$$\mathbb{E}_U[Z_U(\delta)] = \mathbb{E}_U\big[\langle \hat{g}_U(\delta), \hat{\mu}(\delta)\rangle\big] = \langle \mathbb{E}_U[\hat{g}_U(\delta)], \hat{\mu}(\delta)\rangle = \langle \mu(\delta), \hat{\mu}(\delta)\rangle = \|\mu(\delta)\|_2. \tag{37}$$

For compactness, define $m := \|\mu(\delta)\|_2 > 0$.

**Step 2 (Cantelli's inequality (Boucheron et al., 2003)).** Let $X := Z_U(\delta) - m$. Then $\mathbb{E}[X] = 0$ and $\mathrm{Var}(X) = \mathrm{Var}(Z_U(\delta))$. The misalignment event can be rewritten as

$$\{Z_U(\delta) \leq 0\} = \{Z_U(\delta) - m \leq -m\} = \{X \leq -m\} = \{-X \geq m\}. \tag{38}$$

We bound $\mathbb{P}(-X \geq m)$ using Cantelli's inequality (a standard one-sided Chebyshev bound; see, e.g., (Boucheron et al., 2003)).

*Claim (Cantelli / one-sided Chebyshev; (Boucheron et al., 2003)).* If $Y$ is any random variable with $\mathbb{E}[Y] = 0$ and $\mathrm{Var}(Y) = v < \infty$, then for all $t > 0$,

$$\mathbb{P}(Y \geq t) \leq \frac{v}{v + t^2}. \tag{39}$$

*Proof of the claim.* Fix $\lambda > 0$. On the event $\{Y \geq t\}$ we have $(Y + \lambda)^2 \geq (t + \lambda)^2$, hence by Markov's inequality (see, e.g., (Boucheron et al., 2003)),

$$\mathbb{P}(Y \geq t) = \mathbb{P}\big((Y + \lambda)^2 \geq (t + \lambda)^2\big) \leq \frac{\mathbb{E}\big[(Y + \lambda)^2\big]}{(t + \lambda)^2} = \frac{\mathbb{E}[Y^2] + \lambda^2}{(t + \lambda)^2} = \frac{v + \lambda^2}{(t + \lambda)^2}, \tag{40}$$

where we used $\mathbb{E}[Y] = 0$ to remove the cross term in $\mathbb{E}[(Y + \lambda)^2]$. Minimizing the right-hand side over $\lambda > 0$ yields $\lambda^\star = v/t$, and substituting $\lambda^\star$ gives $\mathbb{P}(Y \geq t) \leq v/(v + t^2)$, proving (39). $\diamond$

Apply (39) to $Y := -X$ (note $\mathbb{E}[-X] = 0$ and $\mathrm{Var}(-X) = \mathrm{Var}(X)$) with $t = m$. Using (38),

$$\mathbb{P}_U\big(Z_U(\delta) \leq 0\big) = \mathbb{P}(-X \geq m) \leq \frac{\mathrm{Var}(-X)}{\mathrm{Var}(-X) + m^2} = \frac{\mathrm{Var}(Z_U(\delta))}{\mathrm{Var}(Z_U(\delta)) + \|\mu(\delta)\|_2^2}, \tag{41}$$

which is exactly (9).

**Step 3 (variance decomposition across `RWAN/SANI` randomness).** Write $U = (\omega, \psi)$ and abbreviate $Z := Z_{\omega,\psi}(\delta)$. By the tower property (law of total expectation) and the identity $\mathrm{Var}(Z) = \mathbb{E}[Z^2] - (\mathbb{E}[Z])^2$ (see, e.g., (Boucheron et al., 2003)),

$$\begin{aligned}
\mathrm{Var}_{\omega,\psi}(Z) &= \mathbb{E}_{\omega,\psi}[Z^2] - \big(\mathbb{E}_{\omega,\psi}[Z]\big)^2 = \mathbb{E}_\omega\big[\mathbb{E}_\psi[Z^2 \mid \omega]\big] - \Big(\mathbb{E}_\omega\big[\mathbb{E}_\psi[Z \mid \omega]\big]\Big)^2 \\
&= \mathbb{E}_\omega\bigg[\underbrace{\Big(\mathbb{E}_\psi[Z^2 \mid \omega] - \big(\mathbb{E}_\psi[Z \mid \omega]\big)^2\Big)}_{\mathrm{Var}_\psi(Z|\omega)} + \big(\mathbb{E}_\psi[Z \mid \omega]\big)^2\bigg] - \Big(\mathbb{E}_\omega\big[\mathbb{E}_\psi[Z \mid \omega]\big]\Big)^2 \\
&= \mathbb{E}_\omega\big[\mathrm{Var}_\psi(Z \mid \omega)\big] + \underbrace{\bigg(\mathbb{E}_\omega\big[\big(\mathbb{E}_\psi[Z \mid \omega]\big)^2\big] - \Big(\mathbb{E}_\omega\big[\mathbb{E}_\psi[Z \mid \omega]\big]\Big)^2\bigg)}_{\mathrm{Var}_\omega\big(\mathbb{E}_\psi[Z|\omega]\big)} \\
&= \mathbb{E}_\omega\Big[\mathrm{Var}_\psi\big(Z_{\omega,\psi}(\delta) \mid \omega\big)\Big] + \mathrm{Var}_\omega\Big(\mathbb{E}_\psi\big[Z_{\omega,\psi}(\delta) \mid \omega\big]\Big),
\end{aligned} \tag{42}$$

which is (10). Both terms are non-negative by definition of variance, matching the interpretation that `SANI` ($\psi$) contributes through the conditional-variance term and `RWAN` ($\omega$) contributes through variability of the conditional mean across $\omega$. $\square$

### A.4. Proofs of Module-wise Properties

A.4.1. PROOF OF LEMMA 3.3 RWAN GATE BOUNDS WEIGHT-PERTURBATION MAGNITUDE

*Proof of Lemma 3.3.* Let $\bar{\alpha} = \bar{\alpha}_{i,j}^{\text{gate}} \in \mathbb{R}^{C_j^i}$ denote the pooled response in Eq. (14), and let $m = \max_c |\bar{\alpha}[c]|$. Then for each channel $c$,

$$g_j^i(x_j^i)[c] = \frac{\bar{\alpha}[c]}{m + b}.$$

Since $|\bar{\alpha}[c]| \leq m$ and $b > 0$, we have

$$|g_j^i(x_j^i)[c]| \leq \frac{m}{m + b} \leq 1,$$

and therefore $\|g_j^i(x_j^i)\|_\infty \leq 1$.

Now let $G_j^i$ be the broadcasted version of $g_j^i$ to the kernel shape $K_j^i \times K_j^i \times C_j^i \times \hat{C}_j^i$, so that each coefficient associated with input-channel $c$ is multiplied by $g_j^i[c]$. By the previous bound, $|G_j^i[\cdot]| \leq 1$ elementwise. Hence,

$$\|\xi_j^i \odot G_j^i\|_F^2 = \sum_r (\xi_j^i[r])^2 (G_j^i[r])^2 \leq \sum_r (\xi_j^i[r])^2 = \|\xi_j^i\|_F^2,$$

which implies $\|\xi_j^i \odot G_j^i\|_F \leq \|\xi_j^i\|_F$. Using $\Delta W_j^i = \rho_j^i(\xi_j^i \odot G_j^i)$ from Eq. (15), we obtain

$$\|\Delta W_j^i\|_F = \rho_j^i \|\xi_j^i \odot G_j^i\|_F \leq \rho_j^i \|\xi_j^i\|_F,$$

which is Eq. (17). $\qquad\square$

### A.4.2. PROOF OF PROPOSITION 3.4 RWAN INDUCES INPUT-CONDITIONED (HETEROSCEDASTIC) GAUSSIAN WEIGHT PERTURBATIONS

*Proof of Proposition 3.4.* From Eq. (15), conditioned on $x_j^i$, the gate $g_j^i(x_j^i)$ and its broadcasted tensor $G_j^i(x_j^i)$ are deterministic. Moreover,

$$\Delta W_j^i = \widetilde{W}_j^i - W_j^i = \rho_j^i(\xi_j^i \odot G_j^i(x_j^i)).$$

Since $\xi_j^i \sim \mathcal{N}(0, \sigma_w^2 I)$ has independent Gaussian entries, multiplying each entry by a deterministic scalar produces a Gaussian random variable with scaled variance. Thus each coefficient of $\Delta W_j^i$ is Gaussian with mean 0, and for any coefficient whose input-channel index is $c$, the corresponding broadcast multiplier is $g_j^i(x_j^i)[c]$, giving

$$\mathrm{Var}(\Delta W_j^i[\cdot] \mid x_j^i) = (\rho_j^i)^2 \sigma_w^2 (g_j^i(x_j^i)[c])^2,$$

which is Eq. (18).

For the conditional second moment,

$$\mathbb{E}[\|\Delta W_j^i\|_F^2 \mid x_j^i] = (\rho_j^i)^2 \mathbb{E}[\|\xi_j^i \odot G_j^i(x_j^i)\|_F^2 \mid x_j^i].$$

Expanding the Frobenius norm and using $\mathbb{E}[(\xi_j^i[r])^2] = \sigma_w^2$,

$$\mathbb{E}[\|\xi_j^i \odot G_j^i\|_F^2 \mid x_j^i] = \sum_r (G_j^i[r])^2 \, \mathbb{E}[(\xi_j^i[r])^2] = \sigma_w^2 \sum_r (G_j^i[r])^2 = \sigma_w^2 \|G_j^i\|_F^2.$$

This yields the identity in Eq. (19). Finally, since Lemma 3.3 implies $|G_j^i[r]| \leq 1$ elementwise and the kernel has $K_j^{i\,2} C_j^i \hat{C}_j^i$ coefficients, we have $\|G_j^i\|_F^2 \leq K_j^{i\,2} C_j^i \hat{C}_j^i$, proving the bound. $\qquad\square$

### A.4.3. PROOF OF LEMMA 3.5 SANI FUSION FORMS A SUB-CONVEX NONNEGATIVE MIXTURE

*Proof of Lemma 3.5.* By Eq. (29), $\lambda_{\text{rec}} = \mathrm{SoftPlus}(\lambda_{\mathcal{P}}) \geq 0$ and $\lambda_{\text{nor}} = \mathrm{SoftPlus}(\lambda_{\mathcal{N}}) \geq 0$. Since $\upsilon > 0$, the normalization constant $\gamma = \lambda_{\text{rec}} + \lambda_{\text{nor}} + \upsilon$ is strictly positive. Therefore $w_{\text{rec}} = \lambda_{\text{rec}}/\gamma \geq 0$ and $w_{\text{nor}} = \lambda_{\text{nor}}/\gamma \geq 0$. Moreover,

$$w_{\text{rec}} + w_{\text{nor}} = \frac{\lambda_{\text{rec}} + \lambda_{\text{nor}}}{\lambda_{\text{rec}} + \lambda_{\text{nor}} + \upsilon} = 1 - \frac{\upsilon}{\gamma} \leq 1.$$

Finally, substituting $w_{\text{rec}}, w_{\text{nor}}$ into Eq. (20) yields $\mathcal{A}(u; \psi) = w_{\text{rec}} \Delta^{\text{rec}} + w_{\text{nor}} \Delta^{\text{nor}}$ as claimed. $\qquad\square$

### A.4.4. PROOF OF PROPOSITION 3.6 (SANI FUSION IS SCALE-CONTROLLED UNDER ANY CONVEX NORM)

*Proof of Proposition 3.6.* Using Lemma 3.5, we can write $\mathcal{A}(u; \psi) = w_{\text{rec}} \Delta^{\text{rec}} + w_{\text{nor}} \Delta^{\text{nor}}$ with $w_{\text{rec}}, w_{\text{nor}} \geq 0$ and $w_{\text{rec}} + w_{\text{nor}} \leq 1$. For any convex norm $\|\cdot\|$, positive homogeneity and the triangle inequality imply

$$\|\mathcal{A}(u; \psi)\| = \|w_{\text{rec}} \Delta^{\text{rec}} + w_{\text{nor}} \Delta^{\text{nor}}\| \leq w_{\text{rec}} \|\Delta^{\text{rec}}\| + w_{\text{nor}} \|\Delta^{\text{nor}}\|.$$

Since both weights are nonnegative,

$$w_{\text{rec}} \|\Delta^{\text{rec}}\| + w_{\text{nor}} \|\Delta^{\text{nor}}\| \leq (w_{\text{rec}} + w_{\text{nor}}) \max\{\|\Delta^{\text{rec}}\|, \|\Delta^{\text{nor}}\|\} \leq \max\{\|\Delta^{\text{rec}}\|, \|\Delta^{\text{nor}}\|\},$$

which proves Eq. (28). $\qquad\square$

### A.5. Recursive Randomized Attention Noise

`RecRAN` is designed to produce *progressive* stochastic smoothing: by applying multiple rounds of attention-modulated perturbations with freshly resampled noise, it yields a sequence of stochastic modulations that progressively refines the representation. This recursion presents a continually shifting optimization landscape, making gradient estimation more difficult for adversarial attackers, while attention gating helps preserve salient structure.

**Deterministic attentions.** Given an input feature map $u \in \mathbb{R}^{H \times W \times C}$, `RecRAN` computes two deterministic attentions in parallel: (i) a local attention to learn local attention map by capturing salient spatial cues and (ii) a channel attention to learn channel attention map by capturing global channel contexts:

$$\alpha_l = \wp(\mathrm{Con}_{1 \times 1}(u)), \quad \alpha_c = \wp(\mathrm{m}(\mathrm{GAP}(u) + \mathrm{GMP}(u))), \tag{43}$$

where $\mathrm{Con}_{1 \times 1}$ denotes a $1 \times 1$ convolution, $\mathrm{m}(\cdot)$ is an `MLP`, `GAP` and `GMP` are global average/max pooling over spatial dimensions, respectively, and $\wp(\cdot)$ is sigmoid. We broadcast $\alpha_c \in [0,1]^C$ across spatial dimensions as needed.

**Progressive noise refinement and recursive modulation.** `RecRAN` injects stochasticity by sampling fresh noises at each recursion step $t = 1, \ldots, T$ (we use $T = 5$). Let $\eta_l^{(t)}$ and $\eta_c^{(t)}$ denote the sampled noises (shaped compatibly with $\alpha_l$ and $\alpha_c$), and let $\pi_l, \pi_c$ be learnable scale tensors. We refine the noise using a self-modulation loop to realize heteroscedastic, data-independent stochasticity with learnable amplitude control. This refinement allows the model to learn which channels (or spatial locations) benefit from stronger randomness versus near-deterministic behavior. Formally,

$$\eta_q^{(t),p} = \eta_q^{(t)} \odot \left( \pi_q + \eta_q^{(t)} \odot \pi_q \right), \qquad q \in \{l, c\}. \tag{44}$$

We then form clipped stochastic attentions

$$\tilde{\alpha}_q^{(t)} = \tau\left( \alpha_q + \eta_q^{(t),p} \right), \qquad \tau(v) = \min\{1, \max\{0, v\}\}, \tag{45}$$

and recursively modulate the representation by multiplying the stochastic attentions across steps:

$$u^{\mathrm{rec}} = u \odot \prod_{t=1}^{T} \left[ \tilde{\alpha}_l^{(t)} \odot \tilde{\alpha}_c^{(t)} \right]. \tag{46}$$

This recursion yields progressive stochastic smoothing with attention-controlled preservation of salient cues.

**Lemma A.3** (`RecRAN` is sign-preserving and $\ell_p$-non-increasing). *Let $u^{\mathrm{rec}}$ be the `RecRAN` output in Eq. (46), where each clipped attention $\tilde{\alpha}_q^{(t)}$ is obtained using $\tau(\cdot)$ as in Eq. (45). Then $u^{\mathrm{rec}}$ is elementwise sign-preserving and magnitude-shrinking:*

$$u^{\mathrm{rec}} \odot u \geq 0 \quad \text{and} \quad |u^{\mathrm{rec}}| \leq |u| \quad \text{(elementwise)}.$$

*Consequently, for all $p \in [1, \infty]$,*

$$\|u^{\mathrm{rec}}\|_p \leq \|u\|_p \qquad \text{and} \qquad \|u^{\mathrm{rec}} - u\|_p \leq \|u\|_p. \tag{47}$$

*Proof.* By definition, $\tau(v) = \min\{1, \max\{0, v\}\}$ clips any scalar to $[0,1]$. Hence each clipped attention satisfies $0 \leq \tilde{\alpha}_q^{(t)} \leq 1$ elementwise for all $t$ and $q \in \{l, c\}$. Therefore the per-step mask $m^{(t)}(u) = \tilde{\alpha}_l^{(t)}(u) \odot \tilde{\alpha}_c^{(t)}(u)$ satisfies $0 \leq m^{(t)}(u) \leq 1$ elementwise, and so does their product

$$m_{\mathrm{rec}}(u) = \prod_{t=1}^{T} m^{(t)}(u), \qquad \text{hence} \qquad 0 \leq m_{\mathrm{rec}}(u) \leq 1 \text{ elementwise}.$$

`RecRAN` outputs $u^{\mathrm{rec}} = u \odot m_{\mathrm{rec}}(u)$ (Eq. (46)). Since $m_{\mathrm{rec}}(u) \geq 0$ elementwise, $u^{\mathrm{rec}} \odot u = u \odot u \odot m_{\mathrm{rec}}(u) \geq 0$. Also $|u^{\mathrm{rec}}| = |u| \odot m_{\mathrm{rec}}(u) \leq |u|$ elementwise because $m_{\mathrm{rec}}(u) \leq 1$.

For any $p \in [1, \infty)$,

$$\|u^{\mathrm{rec}}\|_p^p = \sum_r |u^{\mathrm{rec}}[r]|^p \leq \sum_r |u[r]|^p = \|u\|_p^p,$$

hence $\|u^{\mathrm{rec}}\|_p \leq \|u\|_p$. The case $p = \infty$ follows similarly: $\|u^{\mathrm{rec}}\|_\infty = \max_r |u^{\mathrm{rec}}[r]| \leq \max_r |u[r]| = \|u\|_\infty$.

Finally, $u^{\mathrm{rec}} - u = u \odot (m_{\mathrm{rec}}(u) - \mathbf{1})$. Since $m_{\mathrm{rec}}(u) \in [0,1]$ elementwise, $|m_{\mathrm{rec}}(u) - 1| \leq 1$, giving $|u^{\mathrm{rec}} - u| \leq |u|$ elementwise and thus $\|u^{\mathrm{rec}} - u\|_p \leq \|u\|_p$ for all $p \in [1, \infty]$.

**Proposition A.4** (`RecRAN` is Lipschitz on bounded feature domains). *Fix a recursion depth $T$. Consider `RecRAN` as a (random) mapping $R(\cdot; \psi) : u \mapsto u^{\mathrm{rec}}$ defined by Eqs. (45)–(46), for a fixed draw of its internal randomness $\psi$. Assume each per-step multiplicative mask $m^{(t)}(u) = \tilde{\alpha}_l^{(t)}(u) \odot \tilde{\alpha}_c^{(t)}(u)$ is $L_t$-Lipschitz in $\ell_\infty$, i.e., $\|m^{(t)}(u) - m^{(t)}(v)\|_\infty \le L_t \|u - v\|_\infty$, and that inputs are bounded as $\|u\|_\infty, \|v\|_\infty \le M$. Then `RecRAN` is Lipschitz in $\ell_\infty$ with*

$$\|R(u; \psi) - R(v; \psi)\|_\infty \le \Big( 1 + M \sum_{t=1}^T L_t \Big) \|u - v\|_\infty. \tag{48}$$

*Proof.* Write $m(u) = m_{\mathrm{rec}}(u) = \prod_{t=1}^T m^{(t)}(u)$, so that $R(u; \psi) = u \odot m(u)$. For any $u, v$,

$$R(u; \psi) - R(v; \psi) = (u - v) \odot m(u) \; + \; v \odot \big( m(u) - m(v) \big).$$

Taking $\ell_\infty$ norms and using $0 \le m(u) \le 1$ elementwise gives

$$\|(u - v) \odot m(u)\|_\infty \le \|u - v\|_\infty.$$

Also $\|v \odot (m(u) - m(v))\|_\infty \le \|v\|_\infty \cdot \|m(u) - m(v)\|_\infty \le M \|m(u) - m(v)\|_\infty$.

It remains to bound $\|m(u) - m(v)\|_\infty$. For a fixed coordinate $r$, $m_r(u) = \prod_{t=1}^T m_r^{(t)}(u)$ with each factor in $[0, 1]$. A standard telescoping product identity yields

$$m_r(u) - m_r(v) = \sum_{t=1}^T \Big( \prod_{s<t} m_r^{(s)}(u) \Big) \big( m_r^{(t)}(u) - m_r^{(t)}(v) \big) \Big( \prod_{s>t} m_r^{(s)}(v) \Big).$$

Since all products are bounded by 1 in magnitude, we have $|m_r(u) - m_r(v)| \le \sum_{t=1}^T |m_r^{(t)}(u) - m_r^{(t)}(v)|$, hence

$$\|m(u) - m(v)\|_\infty \le \sum_{t=1}^T \|m^{(t)}(u) - m^{(t)}(v)\|_\infty \le \Big( \sum_{t=1}^T L_t \Big) \|u - v\|_\infty,$$

by the assumed Lipschitz property of each $m^{(t)}$.

Combining bounds,

$$\|R(u; \psi) - R(v; \psi)\|_\infty \le \|u - v\|_\infty + M \Big( \sum_{t=1}^T L_t \Big) \|u - v\|_\infty,$$

which is Eq. (48).

### A.6. Non-Recursive Randomized Attention Noise (`NoRAN`)

`NoRAN` is designed that encourages diversity in stochastic perturbations while avoiding over-smoothing. Specifically, `NoRAN` complements `RecRAN` with a *single-pass* (non-recursive) stochastic attention mechanism that injects complementary channel-aware perturbations driven by multiple global pooling descriptors.

**Complementary channel descriptors and stochastic injection.** Given $u \in \mathbb{R}^{H \times W \times C}$, let $\mathrm{D} = \{\mathtt{MN}, \mathtt{SP}, \mathtt{MP}, \mathtt{AP}\}$ denote global min-, sum-, max-, and average-pooling over spatial dimensions. For each $d \in \mathrm{D}$, we compute a deterministic channel attention vector

$$\alpha_d = \wp(\mathrm{m}(d(u))) \in [0, 1]^C, \tag{49}$$

sample noise $\eta_d \sim \mathcal{N}(0, I)$ with the same shape as $u$, and scale it with learnable coefficients $\beta_d$:

$$\eta_d^N = \beta_d \odot \eta_d. \tag{50}$$

NoRAN outputs the non-recursive stochastic representation

$$u^{\mathrm{nor}} = u + \sum_{d \in \mathrm{D}} \big( \eta_d^N \odot u \odot \alpha_d \big), \tag{51}$$

where $\alpha_d$ is broadcast over spatial dimensions. This yields non-progressive but diverse channel-aware perturbations that complement `RecRAN`'s progressive stochastic smoothing.

**Proposition A.5** (NoRAN perturbation is conditionally Gaussian and heteroscedastic). *Let $u^{\mathrm{nor}}$ be defined by Eq. (51), and define $\Delta^{\mathrm{nor}} = u^{\mathrm{nor}} - u$. Assume for each $d$ that $\eta_d \sim \mathcal{N}(0, I)$ has i.i.d. entries, the noises are independent across $d$, and $\eta_d$ is independent of $u$. Conditioned on $u$, $\Delta^{\mathrm{nor}}$ is Gaussian with mean $0$ and diagonal covariance:*

$$\Delta^{\mathrm{nor}} \mid u \sim \mathcal{N}(0, \, \Sigma(u)), \qquad \Sigma(u) = \mathrm{diag}\left( \sum_{d \in D} \big( \beta_d \odot u \odot \alpha_d(u) \big)^{\odot 2} \right), \tag{52}$$

*where $(\cdot)^{\odot 2}$ denotes elementwise squaring (after broadcasting to the tensor shape). In particular, each feature coordinate has variance scaled by the attention and learnable noise strengths.*

*Proof.* From Eq. (51),

$$\Delta^{\mathrm{nor}} = u^{\mathrm{nor}} - u = \sum_{d \in \mathrm{D}} \big( \eta_d^N \odot u \odot \alpha_d(u) \big), \qquad \eta_d^N = \beta_d \odot \eta_d.$$

Conditioned on $u$, the tensors $u$ and $\alpha_d(u)$ are deterministic. Hence each summand can be written as an elementwise scaling of $\eta_d$ by the deterministic tensor $s_d(u) = \beta_d \odot u \odot \alpha_d(u)$ (broadcasted as needed):

$$\eta_d^N \odot u \odot \alpha_d(u) \; = \; \eta_d \odot s_d(u).$$

Since $\eta_d \sim \mathcal{N}(0, I)$ has independent standard normal entries, it follows that $\eta_d \odot s_d(u)$ is Gaussian with mean $0$ and diagonal covariance $\mathrm{diag}(s_d(u)^{\odot 2})$. Independence across $d$ implies the sum over $d$ is Gaussian with covariance equal to the sum of covariances. Therefore,

$$\Delta^{\mathrm{nor}} \mid u \sim \mathcal{N}\Big(0, \, \sum_{d \in \mathrm{D}} \mathrm{diag}\big(s_d(u)^{\odot 2}\big)\Big) = \mathcal{N}\big(0, \Sigma(u)\big),$$

where $\Sigma(u)$ is exactly Eq. (52). $\qquad \blacksquare$

**Corollary A.6** (Directional concentration of NoRAN perturbations). *Under the assumptions of Proposition A.5, for any fixed unit vector $v$ (vectorized tensor),*

$$\mathbb{P}\big(|\langle v, \Delta^{\mathrm{nor}}\rangle| \geq t \mid u\big) \leq 2\exp\left( -\frac{t^2}{2\, v^\top \Sigma(u)\, v} \right). \tag{53}$$

*Equivalently, $\langle v, \Delta^{\mathrm{nor}}\rangle$ is conditionally sub-Gaussian with variance proxy $v^\top \Sigma(u)v$.*

*Proof.* Conditioned on $u$, Proposition A.5 gives $\Delta^{\mathrm{nor}} \mid u \sim \mathcal{N}(0, \Sigma(u))$. For any fixed vector $v$, the scalar $\langle v, \Delta^{\mathrm{nor}}\rangle$ is therefore Gaussian with mean $0$ and variance

$$\mathrm{Var}(\langle v, \Delta^{\mathrm{nor}}\rangle \mid u) = v^\top \Sigma(u)v.$$

The standard Gaussian tail bound states that if $Z \sim \mathcal{N}(0, \sigma^2)$, then $\mathbb{P}(|Z| \geq t) \leq 2\exp(-t^2/(2\sigma^2))$. Applying this with $\sigma^2 = v^\top \Sigma(u)v$ yields Eq. (53). $\qquad \blacksquare$

## A.7. Experimental Setup

**Benchmarks and evaluation metrics.** We evaluate HySCAN on eight image-classification benchmarks spanning natural and medical imaging: CIFAR-10/100 (Krizhevsky, 2009), ImageNet-1k (Deng et al., 2009), CelebA (Liu et al., 2015), NCT-CRC-HE-100K (Kather et al.), NIH-CXR (Wang et al., 2017), EyePACS (EyePACS, 2015), and HAM10000 (Tschandl et al., 2018). We report robustness under two complementary criteria: (i) *certified* robustness (accuracy) under randomized smoothing ($\ell_2$), and (ii) *empirical* robustness (accuracy) under white-box $\ell_\infty$ attacks.

**HySCAN integration and internal randomness.** We integrate HySCAN by replacing each standard convolutional block in a CNN backbone (e.g., ResNet blocks in the ResNet family (He et al., 2016)) with the RWAN→SANI hybrid module (Sec. 3). Specifically, RWAN replaces every convolutional layer within each residual block, and SANI is applied at the end of the resulting HySCAN block, thereby converting each residual block into a HySCAN block. RWAN injects attention-gated stochastic perturbations in weight space (i.e., randomized kernel perturbations), while SANI injects attention-modulated perturbations into intermediate features (Sec. 3.2.1- 3.3). Both sources of internal randomness are enabled during training and inference, and HySCAN's prediction is defined by marginalizing over this internal joint randomness as in Eq. (2).

**Certified evaluation ($\ell_2$).** We certify `HySCAN` using the standard randomized-smoothing guarantee (Theorem 3.1), applied to the smoothed predictor defined in Eq. 2. Concretely, for each test input $x$, certification requires estimating the top-two class probabilities $(p_A, p_B)$ under the *joint* randomness from (i) Gaussian smoothing noise and (ii) `HySCAN`'s internal stochasticity (RWAN and SANI). Accordingly, each Monte Carlo forward pass uses *fresh* draws of `HySCAN`'s internal randomness (e.g., RWAN's $\omega$ and SANI's $\psi$) together with the smoothing noise, matching the stochastic predictor used throughout the paper (ref. Appendix A.4).

We use a two-stage Monte Carlo routine that is standard in smoothing-based certification: a small pilot sample ($n_0$) to identify the most likely class, followed by $n$ additional samples to form one-sided confidence bounds at confidence $1 - \alpha$; the certified radius is then computed from these bounds via Theorem 3.1. We use $n_0{=}100$, $n{=}100{,}000$, and $\alpha{=}0.001$. We consider Gaussian noise levels $\sigma \in \{0.25,\ 0.50,\ 1.0,\ 2.0\}$ and report certified accuracy at predefined $\ell_2$ radii $r$.

Because certification is sampling-intensive, we certify a dataset-specific subset of the test split using a fixed rule, chosen *a priori* to balance coverage and runtime:

- `ImageNet-1k`: certify every $100^{\text{th}}$ test example (default $n_0{=}100$, $n{=}100{,}000$, $\alpha{=}0.001$).
- `CIFAR-10/100`: certify every $5^{\text{th}}$ test example (default $n_0{=}100$, $n{=}100{,}000$, $\alpha{=}0.001$).
- `CelebA` (ARS-style): certify a *uniform, label-stratified* subset of 200 test images with $n_0{=}100$, $n{=}50{,}000$, and failure probability 0.05 (i.e., 95% confidence).
- `NCT-CRC-HE-100K`, `NIH-CXR`, `EyePACS`: certify a *uniform, label-stratified* subsample per dataset, sized to yield approximately 2,000 certified examples each (default $n_0{=}100$, $n{=}100{,}000$, $\alpha{=}0.001$).
- `HAM10000`: certify the full test split when feasible; otherwise, use a *uniform, label-stratified* subsample (default $n_0{=}100$, $n{=}100{,}000$, $\alpha{=}0.001$).

**Empirical evaluation ($\ell_\infty$).** We evaluate empirical robustness under white-box $\ell_\infty$ attacks using `APGD-20`, `AutoAttack` (AA) (Croce & Hein, 2020), and `PGD` (Madry et al., 2018) at $\epsilon \in \{8/255, 16/255\}$. Here, `APGD-20` denotes the union of $\ell_\infty$-$\text{APGD}_{\text{CE}}$ and $\ell_\infty$-$\text{APGD}_{T\text{-}DLR}$ from (Croce & Hein, 2020), each run for 20 iterations with 5 random restarts. Because `HySCAN` is stochastic at inference time, all empirical attacks keep RWAN/SANI randomness active; each stochastic forward pass draws fresh internal randomness, and gradients are computed through that realization. Beyond the default APGD/AA setting, we stress-test `HySCAN` with stronger `PGD` configurations by increasing both perturbation strength and iteration count (Fig. 3), and with multi-step and adaptive stochastic attacks including `EOT-PGD`, `BPDA`, and `EOT-BPDA` (Figs. 4–5).

**Training details for certified robustness.** We follow an `ARS`-style (Lyu et al., 2024), dataset-specific training recipe and train `HySCAN`-integrated backbones end-to-end under Gaussian noise augmentation at the same smoothing level $\sigma$ used at certification. `HySCAN`'s architectural stochasticity (RWAN and SANI) remains active during training, and its internal randomness is resampled across forward passes in the same manner used at inference and certification. We optimize the standard categorical cross-entropy objective and report top-1 accuracy.

**Architectures.** Consistent with common smoothing baselines (Xia et al., 2024), we use `ResNet-110` for `CIFAR-10/100` and `ResNet-50` for the remaining datasets (`ImageNet-1k`, `CelebA`, `NCT-CRC-HE-100K`, `NIH-CXR`, `EyePACS`, `HAM10000`).

**Optimization.** We use the following settings (one recipe per dataset family):

- `CIFAR-10/100`: 200 epochs, batch size 256, `AdamW` with learning rate $10^{-2}$ and weight decay $10^{-4}$. We use a step learning-rate schedule with step size 30 and multiplicative decay $\Gamma = 0.1$.
- `CelebA`, `NCT-CRC-HE-100K`, `NIH-CXR`, `EyePACS`, `HAM10000`: 200 epochs, batch size 64, `SGD` with learning rate $5 \times 10^{-2}$. We use a step schedule with step size 3 and decay $\Gamma = 0.8$.
- `ImageNet-1k`: 200 epochs, batch size 300, `SGD` with learning rate $10^{-1}$, momentum 0.9, and weight decay $10^{-4}$. We use a short warm-up at the beginning of training, followed by a step schedule with step size 30 and decay $\Gamma = 0.1$.

**Baselines and reporting.** We compare against certified randomized-smoothing baselines (RS (Cohen et al., 2019), IRS (Ugare et al., 2023), DRS (Xia et al., 2024), ARS (Lyu et al., 2024)) and empirical defenses reported in Sec. 4 (e.g., AT (Madry et al., 2018), DCS (Ma et al., 2025), CTRW (Ma et al., 2023), RPF (Dong & Xu, 2023)). All results are averaged

over five random seeds; we report the mean and, where applicable, the standard deviation (denoted as $(\cdot)$). All experiments were run on NVIDIA A100 80GB.

## A.8. Details of the `HySCAN` certification procedure

This section describes how we implement *randomized smoothing* (RS) certification for `HySCAN` when the base network is itself stochastic due to `RWAN` and `SANI`. The key requirement for RS is that the smoothed classifier averages predictions over an *input-independent* noise distribution. We therefore treat `HySCAN`'s internal stochasticity as part of the randomized base classifier and *resample it i.i.d. at every forward pass*, both during training and certification.

**Stochastic base classifier and joint randomness.** Let $s_\Theta(x; \omega, \psi) \in \mathbb{R}^{|\mathcal{Y}|}$ denote the `HySCAN` logits for input $x$, where $\omega$ aggregates all `RWAN` random draws (e.g., sampled weight noises and any gating randomness) and $\psi$ aggregates all `SANI` random draws (e.g., `RecRAN`/`NoRAN` noises used for feature injection). We write $(\omega, \psi) \sim \nu$ for the induced internal randomness distribution and assume $(\omega, \psi)$ is independent of the Gaussian smoothing noise $\varepsilon \sim \mathcal{N}(0, \sigma^2 I)$. The corresponding randomized hard classifier is

$$f_\Theta(x; \omega, \psi) := \arg\max_{c \in \mathcal{Y}} s_{\Theta, c}(x; \omega, \psi).$$

The smoothed classifier used throughout the paper (Eq. (2)) is

$$F_\Theta(x) = \arg\max_{c \in \mathcal{Y}} \mathbb{P}_{\varepsilon \sim \mathcal{N}(0, \sigma^2 I), (\omega, \psi) \sim \nu}\big[f_\Theta(x + \varepsilon; \omega, \psi) = c\big].$$

Because the probability is taken over the *full joint randomness* $(\varepsilon, \omega, \psi)$, the standard RS certificate (Theorem 3.1) applies unchanged; see Appendix A.3 for the product-space argument.

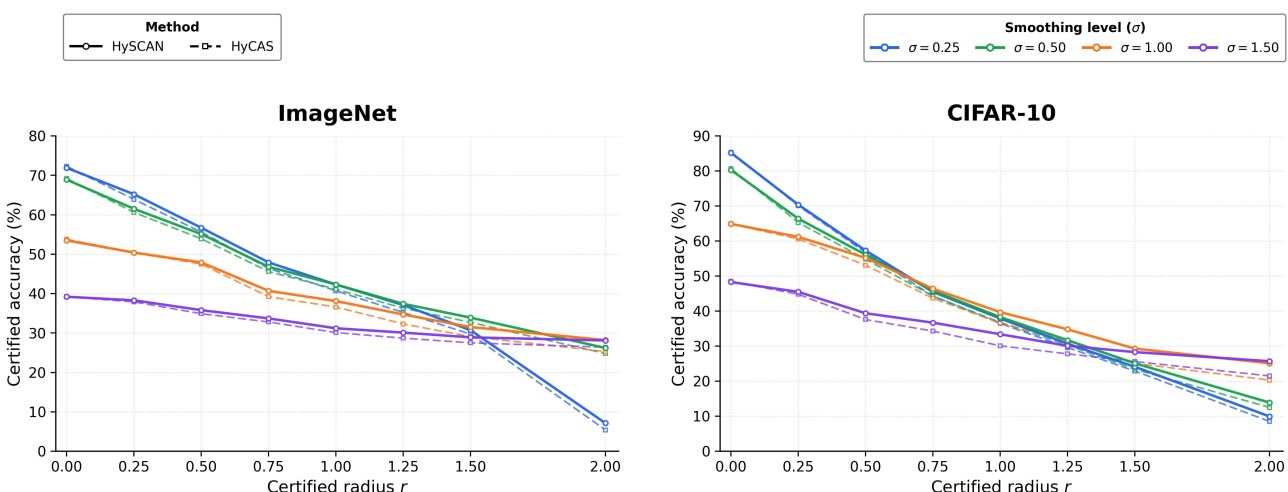

*Figure 7.* Certified accuracy (%) of `HySCAN` versus `HyCAS` on `ImageNet` and `CIFAR-10`. Both methods are evaluated across four smoothing levels, $\sigma \in \{0.25, 0.50, 1.00, 1.50\}$, and certified radii $r$. Solid curves denote `HySCAN`; dashed curves denote `HyCAS`. `HySCAN` shifts the nonzero-radius certified frontier upward while preserving comparable clean accuracy.

### A.8.1. HySCAN TRAINING FOR CERTIFICATION (RS-COMPATIBLE TRAINING)

To obtain valid $\ell_2$ certificates via Theorem 3.1, we train `HySCAN` with an RS-compatible objective that matches the randomness used at certification time. Concretely, for a fixed smoothing level $\sigma$ we minimize the *jointly smoothed risk*

$$\min_\Theta \mathbb{E}_{(x, y) \sim \mathcal{D}} \mathbb{E}_{\varepsilon \sim \mathcal{N}(0, \sigma^2 I), (\omega, \psi) \sim \nu}\Big[\ell\big(s_\Theta(x + \varepsilon; \omega, \psi), y\big)\Big]. \tag{54}$$

This corresponds to Eq. (30) with $\kappa = 0$ (i.e., no adversarial $\ell_\infty$ term). In practice, the nested expectations in Eq. (54) are estimated with Monte Carlo sampling, using fresh draws of $\varepsilon$ and $(\omega, \psi)$ on every forward evaluation. The details of `HySCAN` training with certification guarantees are presented in Algorithm 1.

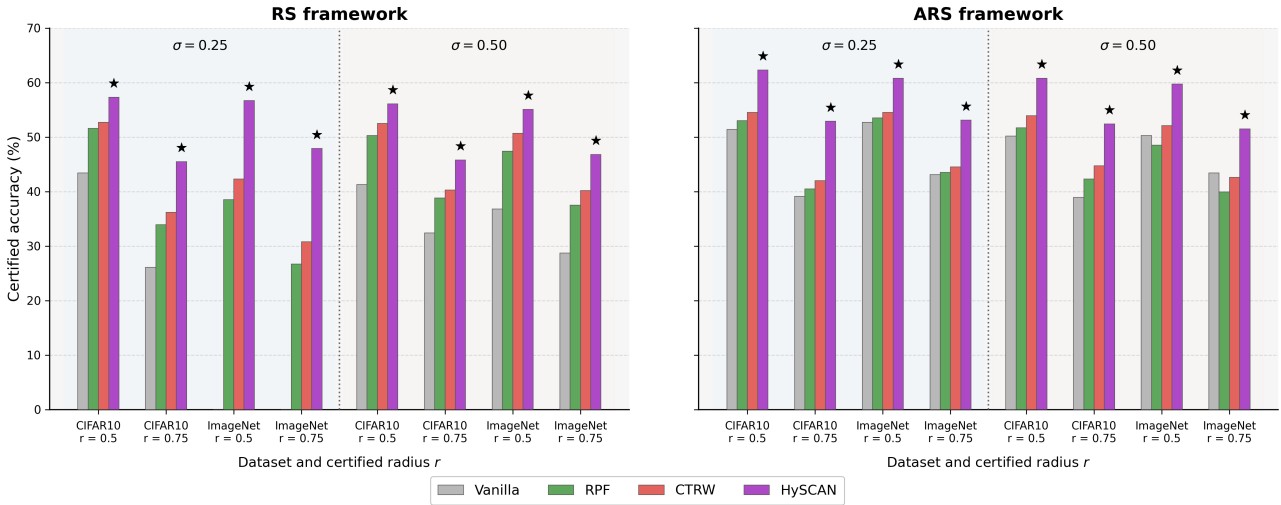

*Figure 8.* Same-framework base-model comparison under `RS` and `ARS`. Certified accuracy is reported on `CIFAR-10` and `ImageNet` at representative radii and smoothing levels. By replacing the base stochastic mechanism with `HySCAN` inside the same certification framework, certified accuracy improves across the evaluated operating points.

---

**Algorithm 1 `RS`-compatible training of `HySCAN` (for certification)**

---

1: **Require:** Dataset $\mathcal{D}$; epochs $E$; batch size $B$; smoothing noise $\sigma$; `HySCAN` logits $s_\Theta(\cdot; \omega, \psi)$; loss $\ell$; optimizer $\texttt{Opt}(\cdot)$.
2: **for** $e = 1, \ldots, E$ **do**
3:     **for** each minibatch $\{(x_i, y_i)\}_{i=1}^B \sim \mathcal{D}$ **do**
4:         Draw $\varepsilon_i \sim \mathcal{N}(0, \sigma^2 I)$ for $i = 1, \ldots, B$ and set $\tilde{x}_i \leftarrow x_i + \varepsilon_i$.    **// Gaussian `RS` noise (one draw per example)**
5:         Draw $(\omega_i, \psi_i) \sim \nu$ for $i = 1, \ldots, B$.    **// `HySCAN` randomness (resampled i.i.d.; realized inside `RWAN`/`SANI`)**
6:         Compute the minibatch loss $\mathcal{L} \leftarrow \dfrac{1}{B} \sum_{i=1}^B \ell\big(s_\Theta(\tilde{x}_i; \omega_i, \psi_i), y_i\big)$.
7:         $\Theta \leftarrow \texttt{Opt}\big(\Theta, \nabla_\Theta \mathcal{L}\big)$.
8:     **end for**
9: **end for**

---

### A.8.2. H Y SCAN INFERENCE-TIME CERTIFICATION (MONTE CARLO RS)

At test time, we certify the smoothed `HySCAN` classifier $F_\Theta$ (Eq. (2)) using the standard two-stage Monte Carlo procedure from randomized smoothing. Every certification query draws from the *full joint randomness*

$$(\varepsilon, \omega, \psi) \sim \mathcal{N}(0, \sigma^2 I) \times \nu,$$

i.e., *each* forward pass uses fresh Gaussian input noise *and* freshly resampled `RWAN`/`SANI` randomness.

**Estimating $p_A$ and forming a valid lower bound.** Fix a test input $x$. Let $c_A$ denote the most likely class under the joint randomness:

$$c_A \in \arg\max_{c \in \mathcal{Y}} p_c(x), \qquad p_c(x) = \mathbb{P}_{\varepsilon, (\omega, \psi)}\big[f_\Theta(x + \varepsilon; \omega, \psi) = c\big].$$

We first identify a candidate $\hat{c}_A$ via a pilot run with $n_0$ samples. We then draw $n$ additional samples and count how often $\hat{c}_A$ occurs. Let $m_A$ be this count. We compute a *one-sided* Clopper–Pearson lower confidence bound $p_A^{\text{LB}}$ on $p_{\hat{c}_A}(x)$ with failure probability $\alpha$.

**Bounding the runner-up probability.** To obtain a *rigorous* certificate for Theorem 3.1 without additional multiple-comparison bookkeeping, we use the standard `RS` bound

$$p_B = \max_{c \neq c_A} p_c(x) \leq 1 - p_A, \qquad \Rightarrow \qquad p_B^{\text{UB}} := 1 - p_A^{\text{LB}}.$$

---

**Algorithm 2 `HySCAN` inference-time `RS` certification under joint randomness** $(\varepsilon, \omega, \psi)$

---

1: **Require:** Trained logits $s_\Theta(\cdot; \omega, \psi)$; test point $x$; noise level $\sigma$; pilot samples $n_0$; certification samples $n$; failure probability $\alpha$.

2: **Output:** Certified label $\hat{y}$ and radius $\hat{r}_2(x)$, or ABSTAIN.

3: **for** $t = 1, \ldots, n_0$ **do**

                                                             **// (1) Pilot: identify the top class**

4:    Draw $\varepsilon^{(t)} \sim \mathcal{N}(0, \sigma^2 I)$.

5:    Draw $(\omega^{(t)}, \psi^{(t)}) \sim \nu$.

6:    $c^{(t)} \leftarrow \arg\max_k s_{\Theta,k}(x + \varepsilon^{(t)}; \omega^{(t)}, \psi^{(t)})$.

7: **end for**

8: $\hat{c}_A \leftarrow \arg\max_c \text{COUNT}\big(\{c^{(t)}\}_{t=1}^{n_0} = c\big)$.

9: **for** $t = 1, \ldots, n$ **do**

                                                              **// (2) Estimation: estimate $p_A$ for $\hat{c}_A$**

10:    Draw $\varepsilon^{(t)} \sim \mathcal{N}(0, \sigma^2 I)$.

11:    Draw $(\omega^{(t)}, \psi^{(t)}) \sim \nu$.

12:    $c^{(t)} \leftarrow \arg\max_k s_{\Theta,k}(x + \varepsilon^{(t)}; \omega^{(t)}, \psi^{(t)})$.

13: **end for**

14: $m_A \leftarrow \text{COUNT}\big(\{c^{(t)}\}_{t=1}^{n} = \hat{c}_A\big)$.

15: $p_A^{\text{LB}} \leftarrow \text{CLOPPERPEARSONLOWER}(m_A, n, 1 - \alpha)$.

16: $p_B^{\text{UB}} \leftarrow 1 - p_A^{\text{LB}}$.                                                     **// Valid bound on $\max_{c \neq \hat{c}_A} p_c(x)$**

17: **if** $p_A^{\text{LB}} \leq 1/2$ **then**

18:    **return** ABSTAIN

19: **else**

20:    $\hat{r}_2(x) \leftarrow \frac{\sigma}{2}\Big(\Phi^{-1}(p_A^{\text{LB}}) - \Phi^{-1}(p_B^{\text{UB}})\Big)$. {Equivalently, $\hat{r}_2(x) = \sigma \, \Phi^{-1}(p_A^{\text{LB}})$ since $p_B^{\text{UB}} = 1 - p_A^{\text{LB}}$.}

21:    **return** $(\hat{y} = \hat{c}_A, \, \hat{r}_2(x))$

22: **end if**

---

This yields a valid certificate with overall failure probability at most $\alpha$.

**Certified radius.** Given $(p_A^{\text{LB}}, p_B^{\text{UB}})$, we compute the certified radius using Theorem 3.1:

$$\hat{r}_2(x) \;=\; \frac{\sigma}{2}\Big(\Phi^{-1}(p_A^{\text{LB}}) \;-\; \Phi^{-1}(p_B^{\text{UB}})\Big), \qquad p_B^{\text{UB}} = 1 - p_A^{\text{LB}}. \tag{55}$$

The details of `HySCAN` inference time with certification guarantees are presented in Algorithm 2.

## A.9. `HySCAN` with Adversarial Training for Empirical $\ell_\infty$ Robustness.

Let $f_\Theta(\cdot; \omega, \psi) : \mathbb{R}^d \to \mathbb{R}^C$ denote the `HySCAN`-integrated classifier parameterized by $\Theta$, where $\omega$ and $\psi$ capture the internal randomness induced by `RWAN` and `SANI`, respectively. In the empirical robustness component of our hybrid objective Eq. (30), we generate $\ell_\infty$-bounded adversarial examples on-the-fly and update $\Theta$ using the resulting worst-case loss.

For a labeled sample $(x, y)$ and perturbation budget $\epsilon_\infty$, an adversarial example is defined as

$$x^\star \;=\; \max_{x' : \|x' - x\|_\infty \leq \epsilon_\infty} \ell\big(f_\Theta(x'; \omega^{[\text{Att}]}, \psi^{[\text{Att}]}), y\big), \tag{56}$$

where [Att] denotes the *attack phase*: gradients are taken with respect to the input (equivalently, $\delta = x' - x$), while $\Theta$ is held fixed.

`HySCAN`'s stochasticity is part of the deployed predictor: `RWAN` and `SANI` randomness are *freshly resampled per forward pass during training and inference*. Consequently, during adversarial example generation we compute each `PGD` gradient step using a fresh draw of $(\omega, \psi)$, matching the forward-pass stochasticity used throughout the model.

Adversarial training then solves the standard min–max problem associated with the empirical term in Eq. (30):

$$\min_{\Theta^{[\text{Inf}]}} \; \max_{x' : \|x' - x\|_\infty \leq \epsilon_\infty} \ell\big(f_\Theta(x'; \omega^{[\text{Att}]}, \psi^{[\text{Att}]}), y\big), \tag{57}$$

---

**Algorithm 3** Adversarial Training with `HySCAN` for Empirical $\ell_\infty$ Robustness

---

1: **Require:** dataset $\mathcal{D}$; `HySCAN` network $f_\Theta(\cdot; \omega, \psi)$; $\ell_\infty$ budget $\epsilon_\infty$; PGD steps $K$; step size $a$; optimizer `Opt`$(\cdot)$.
2: **while** not converged **do**
3:  Sample a minibatch $\{(x_i, y_i)\}_{i=1}^B \sim \mathcal{D}$.
4:  Initialize $\delta_i^0 \sim \text{Unif}([-\epsilon_\infty, \epsilon_\infty])$ **or** $\delta_i^0 = 0$.
5:  **for** $t = 0, \ldots, K - 1$ **do**

**// attack phase** [Att]

6:    Draw fresh $(\omega_i^t, \psi_i^t)$ for each forward pass.
7:    $g_i^t \leftarrow \nabla_\delta \, \ell\big(f_\Theta(x_i + \delta_i^t; \omega_i^t, \psi_i^t), y_i\big)$.
8:    $\delta_i^{t+1} \leftarrow \Pi_{\|\delta\|_\infty \leq \epsilon_\infty}\big(\delta_i^t + a \cdot \text{sign}(g_i^t)\big)$.
9:    $x_i + \delta_i^{t+1} \leftarrow \text{Clip}_{[0,1]}(x_i + \delta_i^{t+1})$.
10:  **end for**
11:  $x_i^{\text{adv}} \leftarrow x_i + \delta_i^K$.
12:  Draw fresh $(\omega_i, \psi_i)$ for the parameter update.    **// update (inference) phase** [Inf]
13:  Compute $\mathcal{L} \leftarrow \mathcal{L}_{\texttt{HySCAN}}\big(\Theta; \{(x_i, y_i, x_i^{\text{adv}})\}_{i=1}^B\big)$ using Eqs. (4)–(5).
14:  $\Theta \leftarrow \texttt{Opt}\big(\Theta, \nabla_\Theta \mathcal{L}\big)$.
15: **end while**

---

where [Inf] denotes the *update (inference) phase*: gradients are taken with respect to $\Theta$. In practice, we approximate the inner maximization with multi-step projected gradient ascent (`PGD`) and then update $\Theta$ by minimizing the `HySCAN` objective in Eqs. (30)–(57), which combines the `RWAN/SANI` module-wise losses via the learnable weights $(\phi, \zeta)$. The details of `HySCAN` with adversarial training for empirical adversarial robustness are presented in Algorithm 3.

## A.10. Extended Experimental Results.

### A.10.1. EXTENDED CERTIFIED ROBUSTNESS ON NATURAL VISION

Figure 7 gives a direct certified-robustness comparison with the prior `HyCAS` baseline. The gain is primarily a robust-radius gain: `HyCAS` can be comparable or slightly higher at $r = 0$, but `HySCAN` improves the nonzero-radius frontier across `ImageNet` and `CIFAR-10`. This behavior is consistent with the methodological distinction between the two defenses. `HyCAS` uses hybrid streams with separately controlled stochastic components, whereas `HySCAN` uses the same attention-noise mechanism to coordinate `RWAN`'s weight-space stochasticity and `SANI`'s feature-space stochasticity inside one randomized base predictor. As a result, the `RS`-style classifier marginalizes over both input smoothing noise and internally resampled `RWAN/SANI` randomness, producing a stronger certified frontier rather than merely improving clean accuracy.

Figure 8 controls for the certification framework by comparing the base model inside the same `RS` and `ARS` pipelines. The consistent improvement of `HySCAN` over vanilla, `RPF`, and `CTRW` indicates that the gain is not inherited only from the smoothing framework. Instead, it comes from the randomized base predictor: `RWAN` perturbs the convolutional operator through attention-noise-induced weight-space stochasticity, while `SANI` perturbs the intermediate representation through attention-noise-induced feature-space stochasticity. The same trend under both `RS` and `ARS` supports the claim that `HySCAN` is a framework-compatible base architecture rather than a framework-specific tuning artifact.

### A.10.2. CERTIFIED ROBUSTNESS ON MEDICAL BENCHMARKS.

Across `EyePACS` and `NCT-CRC-HE-100K`, `HySCAN` consistently improves certified accuracy over `ARS/DRS` for every preset radius $r \in \{0, 0.25, 0.5, 0.75, 1.0\}$ at both smoothing levels $\sigma \in \{0.25, 0.5\}$ (see Table 7). On `EyePACS`, `HySCAN` improves over `ARS` by +4.4 pp at $r = 0.25$ (67.3 vs 62.9) and +4.3 pp at $r = 1.0$ (40.5 vs 36.2) for $\sigma = 0.25$, and remains ahead under $\sigma = 0.5$ (+4.0 pp at $r = 0.25$ and +4.2 pp at $r = 1.0$). On `NCT-CRC-HE-100K`, `HySCAN` yields consistent mid-radius gains (e.g., +4.8 pp at $r = 0.5$ for $\sigma = 0.25$, 64.2 vs 59.4) and strengthens the high-radius regime under larger noise (+3.0 pp at $r = 1.0$ for $\sigma = 0.5$, 37.2 vs 34.2). We attribute these improvements to `HySCAN`'s *distributed, attention-conditioned* stochasticity: `RWAN` introduces input-dependent (heteroscedastic) perturbations in weight space, while `SANI` injects additive attention-modulated feature noise, jointly stabilizing predictions under the *same joint randomness* used by certification. Since the randomized-smoothing radius satisfies $r_2(x) = \frac{\sigma}{2}\big(\Phi^{-1}(p_A) - \Phi^{-1}(p_B)\big)$ with

$(p_A, p_B)$ computed under $(\varepsilon, \omega, \psi)$, HySCAN's structured multi-level randomization enlarges the top-two probability gap and increases the fraction of examples whose certificates exceed a target radius. This mechanism is especially beneficial for medical imagery, where discriminative cues can be spatially localized: attention gating concentrates stochasticity on salient regions/channels while avoiding indiscriminate corruption of fine-grained structure. Finally, the notably smaller standard deviations (e.g., at $r = 0.25$) suggest the gains are reproducible across seeds, consistent with systematically improved margins rather than seed-specific effects.

A.10.3. EMPIRICAL $\ell_\infty$ ROBUSTNESS UNDER **WHITE-BOX** ATTACKS.

**Multi-step Attack.** Figure 9 studies the stability of HySCAN under EOT-PGD on CIFAR-10. Since HySCAN resamples RWAN/SANI randomness at inference time, EOT estimates the attack gradient by averaging over multiple stochastic forward passes, matching the evaluation protocol used for stochastic defenses. We use 20 PGD steps and report a sweep over 0–10 EOT trials; the main adaptive setting uses 7 EOT trials for consistency with the stochastic-deformable-convolution baseline. After EOT is enabled, robust accuracy remains stable across the sweep (mean 74.72% over 1–10 trials), rather than degrading monotonically as more stochastic samples are used. This indicates that HySCAN's empirical robustness is not an artifact of a particular EOT choice, but is consistent with its attention-noise-induced weight- and feature-space stochasticity.

**Extended Results.** Table 6 reports top-1 accuracy under a standard $\ell_\infty$ threat model at $\epsilon = 8/255$, evaluated with **white-box** adversarial attacks (PGD-20 and AA-20) on CIFAR-10 and ImageNet. Across *both* datasets and *both* attack instantiations, **HySCAN achieves the strongest empirical robustness while also attaining the highest clean accuracy**, indicating that the gains do not arise from trading away nominal generalization for robustness.

On CIFAR-10, HySCAN reaches **90.93%** clean accuracy and the best robust accuracies (**74.63%** under PGD-20 and **78.47%** under AA-20). Relative to the strongest competing baseline in this table (DCS), HySCAN improves by **+0.51 pp** on clean accuracy, **+1.11 pp** under PGD-20, and **+2.39 pp** under AA-20. The margin over other recent defenses is similarly consistent: compared to CTRW, HySCAN gains **+3.51 pp** (PGD-20) and **+4.39 pp** (AA-20), while also improving clean accuracy. Notably, the improvements are *larger* under adversarial evaluation than on clean data, suggesting that HySCAN strengthens decision stability specifically in the adversarially relevant regime rather than merely improving standard accuracy.

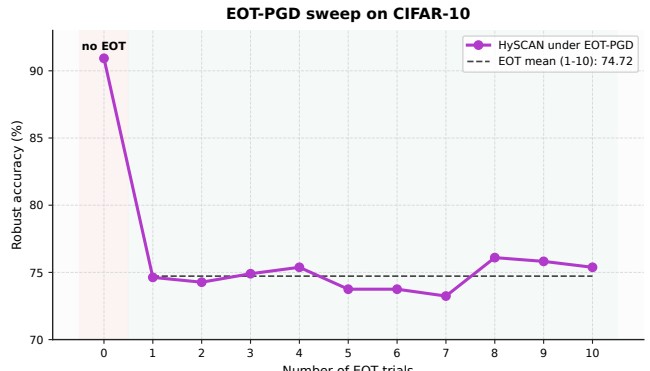

*Figure 9.* HySCAN stability under multi-step EOT-PGD evaluation on CIFAR-10. PGD is fixed to 20 steps while EOT trials are varied from 0 to 10.

On ImageNet, HySCAN again sets the best clean accuracy (**74.24%**) and the best robustness (**60.69%** PGD-20, **68.32%** AA-20). The gains are especially pronounced over smoothing-based certified defenses: HySCAN improves over ARS by **+15.91 pp** (PGD-20) and **+22.67 pp** (AA-20), and over DRS by **+20.06 pp** (PGD-20) and **+25.50 pp** (AA-20). These results support the central objective of HySCAN: narrowing the long-observed gap between provable robustness pipelines and practical robustness against strong **white-box** $\ell_\infty$ attacks, *without* incurring a clean-accuracy regression.

*Table 6.* Empirical Robustness (accuracy (%)) against $\ell_\infty$ adversarial attacks (PGD-20 and AA-20) on CIFAR-10 (Krizhevsky, 2009) (left) and ImageNet (Deng et al., 2009) (right) at $\epsilon \in \{8/255\}$.

| Method | CIFAR-10 | | | ImageNet | | |
|---|---|---|---|---|---|---|
| | Clean | PGD–20 | AA–20 | Clean | PGD–20 | AA–20 |
| DRS (Xia et al., 2024) | 88.31 | 63.86 | 65.51 | 72.23 | 40.63 | 42.82 |
| ARS (Lyu et al., 2024) | 89.10 | 66.74 | 68.13 | 73.13 | 44.78 | 45.65 |
| CTRW (Ma et al., 2023) | 90.18 | 71.12 | 74.08 | 73.87 | 60.16 | – |
| RPF (Dong & Xu, 2023) | 89.85 | 68.85 | 69.31 | 73.24 | 56.56 | – |
| DCS (Ma et al., 2025) | 90.42 | 73.52 | 76.08 | 73.87 | 52.38 | 66.79 |
| **HySCAN** (Ours) | **90.93** | **74.63** | **78.47** | **74.24** | **60.69** | **68.32** |

**Why HySCAN improves over prior defenses.** We attribute these consistent gains to HySCAN's *hybrid, space-aware stochasticity* that hardens both (i) **how** features are computed and (ii) **what** representations look like. RWAN introduces *implicit* stochasticity by perturbing convolutional operators in a structured manner, while SANI

*Table 7.* `HySCAN` vs. state-of-the-art certified defenses: `ARS` and `DRS` to evaluate certified accuracy on `EyePacs` and `NCT-CRC-HE-100K` benchmarks. The best performance is shown as **bold**.

| Approach | $\sigma$ | EyePacs | | | | | NCT-CRC-HE-100K | | | | |
|---|---|---|---|---|---|---|---|---|---|---|---|
| | | Certified accuracy at predetermined $\ell_2$ radius $r$ (%) | | | | | Certified accuracy at predetermined $\ell_2$ radius $r$ (%) | | | | |
| | | 0.00 | 0.25 | 0.50 | 0.75 | 1.00 | 0.00 | 0.25 | 0.50 | 0.75 | 1.00 |
| DRS | 0.25 | 81.3 ± 1.92 | 60.1 ± 2.83 | 50.1 ± 1.53 | 40.9 ± 2.04 | 26.2 ± 2.69 | 89.5 ± 2.78 | 67.6 ± 2.63 | 56.7 ± 3.35 | 45.3 ± 2.27 | 30.5 ± 3.54 |
| | 0.5 | 78.9 ± 0.91 | 57.7 ± 2.82 | 51.8 ± 1.34 | 42.1 ± 0.96 | 30.4 ± 3.74 | 85.2 ± 1.67 | 65.6 ± 1.89 | 56.2 ± 1.23 | 48.2 ± 1.67 | 33.1 ± 1.20 |
| ARS | 0.25 | 83.1 ± 1.35 | 62.9 ± 2.93 | 47.9 ± 1.94 | 40.7 ± 2.32 | 36.2 ± 3.73 | 91.7 ± 1.84 | 69.3 ± 2.07 | 59.4 ± 2.37 | 48.3 ± 3.91 | 31.9 ± 2.63 |
| | 0.5 | 80.9 ± 1.14 | 60.7 ± 2.27 | 51.5 ± 1.42 | 42.2 ± 2.68 | 37.5 ± 0.98 | 87.8 ± 2.25 | 68.4 ± 1.91 | 60.1 ± 1.01 | 50.7 ± 0.53 | 34.2 ± 1.39 |
| HyCAS | 0.25 | 86.7 ± 0.97 | 66.1 ± 1.62 | 51.4 ± 2.74 | 45.7 ± 1.27 | 39.2 ± 2.61 | 95.4 ± 2.02 | 72.9 ± 1.63 | 63.1 ± 1.59 | 51.7 ± 3.11 | **33.2 ± 1.84** |
| | 0.5 | 82.6 ± 1.89 | 63.9 ±1.74 | 53.2 ±2.81 | 46.3 ±1.45 | 41.5 ±1.67 | **92.3 ± 0.61** | 71.7 ± 1.11 | 63.4 ± 1.36 | 52.2 ± 1.21 | 36.9 ± 2.57 |
| HySCAN | 0.25 | **87.1 ±0.82** | **67.3 ±0.59** | 52.2 ±1.93 | 46.4 ±0.88 | **40.5 ±1.87** | **95.9 ±1.62** | **73.4 ±0.56** | **64.2 ±1.52** | **52.5 ±1.45** | 32.6 ±0.69 |
| | 0.5 | **83.3 ±0.58** | **64.7 ±0.51** | **53.5±1.92** | **46.8±0.97** | **41.7±1.31** | 91.9 ± 1.03 | 71.4 ± 0.95 | **64.2 ±0.83** | **52.7 ±0.91** | **37.2 ±1.69** |

introduces *explicit* stochasticity by injecting attention-modulated noise into intermediate features. This two-level design is crucial: operator perturbations discourage brittle reliance on a single fixed set of filters, whereas feature perturbations encourage smoother intermediate geometry under perturbations. Attention conditioning further ensures that the injected randomness is *selective* rather than indiscriminate—preserving discriminative structure and explaining why `HySCAN` improves clean accuracy while also raising robust accuracy. Ablations in the paper corroborate this complementarity: enabling both `RWAN` and `SANI` yields the strongest robustness, exceeding either component in isolation, consistent with the view that single-location randomization is helpful but incomplete.

**Robustness-Efficiency Tradeoff.** Figure 11 summarizes the robustness–cost profile of `HySCAN`. The figure shows that `HySCAN` occupies the highest-robustness region for both certified and empirical evaluations, while incurring additional computation from its architecture-level stochasticity. This cost is expected: `RWAN` replaces deterministic convolutions with attention-noise-conditioned stochastic kernels, and `SANI` is repeatedly invoked for both per-layer weight-space gating and block-level feature-space injection. Importantly, this design choice is motivated by safety-sensitive robustness rather than efficiency alone. In settings such as clinical decision support, robustness failures can have a higher cost than additional computation, so the overhead is a deliberate trade-off for stronger adversarial reliability. The same accounting also explains the larger overhead on `ImageNet`-scale `ResNet-50` than on `CIFAR-10`-scale `ResNet-110`, where feature-map resolution and repeated attention-noise operations dominate the cost.

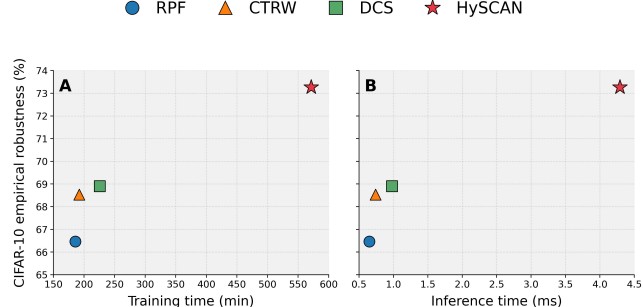

*Figure 10.* Training time (A) and inference time (B) versus empirical robustness on `CIFAR-10`. `HySCAN` achieves the highest empirical robustness.

Figure 10 makes the `CIFAR-10` runtime trade-off explicit. `RPF`, `CTRW`, and `DCS` require less training and inference time, but their empirical robustness remains lower than `HySCAN`. The higher runtime of `HySCAN` follows directly from its deployment-time stochasticity: each forward pass samples weight-space randomness, computes attention-noise gates, and injects feature-space stochasticity, while adversarial training further resamples this randomness across attack steps. Thus, the added time is not incidental overhead; it is the computational price of the same mechanism that improves robustness. For security- and safety-sensitive settings, this is a reasonable design point: `HySCAN` prioritizes adversarial reliability over minimal latency, while still exposing the cost for transparent comparison.

A.10.4. CERTIFIED–EMPIRICAL ROBUSTNESS PROFILES UNDER INCREASING PERTURBATION STRENGTH.

Figure 12 summarizes `HySCAN`'s behavior across the two robustness regimes studied in this work: (i) *certified* robustness via randomized smoothing (certified accuracy as a function of the target $\ell_2$ radius $r$, bottom axis), and (ii) *empirical* robustness under strong white-box $\ell_\infty$ attacks (robust accuracy as a function of attack strength $\epsilon$, top axis). We emphasize that these

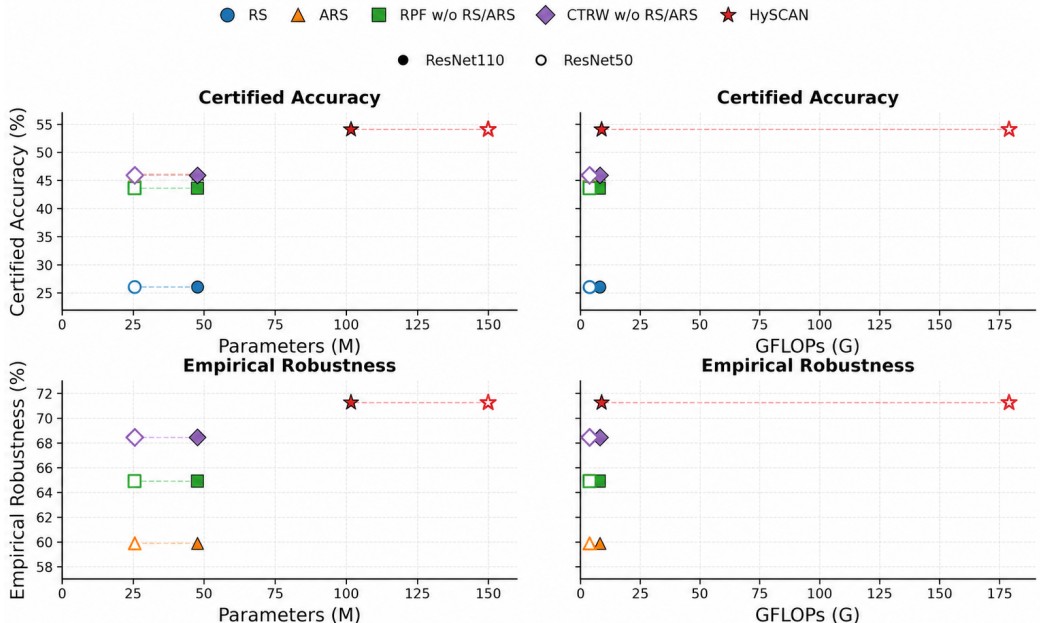

*Figure 11.* Computation–robustness trade-off across certified and empirical settings. The plots compare robustness against parameters and FLOPs for representative pipelines. `HySCAN` obtains the highest certified and empirical robustness averages, while incurring additional computational cost from attention-noise-induced weight- and feature-space stochasticity.

curves are not intended to imply a zero-sum relationship; instead, they provide a compact *joint characterization* of how performance evolves as perturbations become stronger under each evaluation protocol.

**CIFAR-10: certificates vs. empirical robustness.** On `CIFAR-10` (left), both certified and empirical accuracy decrease smoothly as perturbation strength increases, but the certified curve decays more rapidly. This separation is expected: randomized-smoothing certificates are *sufficient* guarantees under an $\ell_2$ threat model and can be substantially more conservative than observed robustness under $\ell_\infty$ attacks. Importantly, `HySCAN` maintains a gradual, monotone degradation in the empirical curve even in regimes where certificates become small, indicating that the defense's architectural stochasticity remains effective under stronger attacks rather than exhibiting brittle regime changes.

**ImageNet: closer alignment across regimes.**

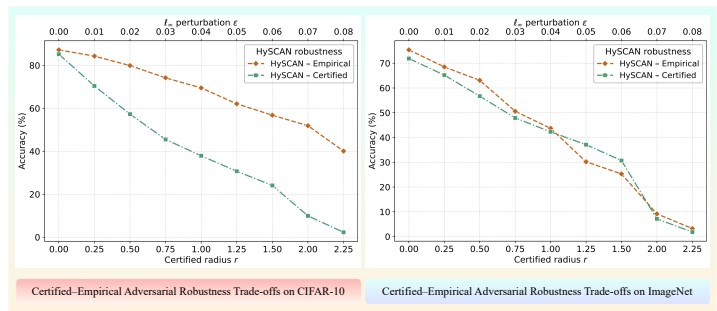

*Figure 12.* Certified ($\ell_2$) and empirical ($\ell_\infty$) adversarial robustness of `HySCAN` on `CIFAR-10` (left) and `ImageNet` (right) as perturbation strength increases (bottom axis: certified radius $r$; top axis: $\ell_\infty$ attack strength $\epsilon$).

On `ImageNet` (right), the certified and empirical curves track each other more closely over a broad mid-range of perturbation strengths and intersect at moderate radii, suggesting that the stability improvements captured by certification are reflected in the model's practical resistance to strong first-order $\ell_\infty$ attacks as well. As perturbation strength approaches the most stringent settings, both curves converge toward low accuracy, providing a clear and coherent stress-test signal: robustness degrades in a controlled manner as the task becomes inherently harder.

Overall, Fig. 12 provides a concise robustness "fingerprint" for `HySCAN`: as perturbations strengthen, certified and empirical accuracies decrease smoothly on both `CIFAR-10` and ImageNet, with (i) an interpretable certificate–empirical gap on `CIFAR-10` due to conservativeness and norm mismatch, and (ii) noticeably tighter alignment on ImageNet. This joint view is consistent with `HySCAN`'s core objective of narrowing the practical gap between provable $\ell_2$ certification and empirical robustness under modern $\ell_\infty$ attacks by distributing freshly resampled, attention-modulated stochasticity throughout the network.

A.10.5. COMPUTATIONAL COST ANALYSIS.

Table 8 quantifies the computational cost of replacing a vanilla backbone with its `HySCAN`-integrated counterpart. For `ResNet-110` on CIFAR-10 (32×32), `HySCAN` increases parameters from 47.6M to 101.6M (2.13×) and `FLOPs` from 8.10G to 8.75G (1.08×), while raising activation memory from 105.33MB to 387.71MB (3.68×) and inference time from 0.33ms to 4.29ms (13.0×). For `ResNet-50` on `ImageNet` (224×224), the overhead is larger: parameters increase from 25.6M to 149.8M (5.85×), `FLOPs` from 3.8G to 179.03G (47.1×), memory from 98.89MB to 571.4MB (5.78×), and inference time from 2.29ms to 15.7ms (6.86×). This increase is inherent to `HySCAN`'s architecture-level randomization: `RWAN` replaces *every* convolution with an attention-gated stochastic kernel, requiring (i) sampling weight noise and (ii) computing per-layer gates via `SANI`'s attention-noise learner (Eqs. (12)–(15), Eq. (24)), while `SANI` additionally performs an explicit feature-space

*Table 8.* Computational cost comparison between vanilla backbones and their `HySCAN`-integrated counterparts. We report the number of parameters (M), `FLOPs` (G), activation memory (MB), and inference time (ms) for `ResNet-50` on `ImageNet-1K` ($224 \times 224 \times 3$) and `ResNet-110` on CIFAR-10 ($32 \times 32 \times 3$).

| Backbone | Variant | Inputs | Parameters (M) | FLOPs (G) | Memory (MB) | Inference Time (ms) |
|---|---|---|---|---|---|---|
| ResNet-50 | Vanilla | $224 \times 224 \times 3$ | 25.6 | 3.8 | 98.89 | 2.29 |
| | HySCAN | | 149.8 | 179.03 | 571.4 | 15.7 |
| ResNet-110 | Vanilla | $32 \times 32 \times 3$ | 47.6 | 8.10 | 105.33 | 0.33 |
| | HySCAN | | 101.6 | 8.75 | 387.71 | 4.29 |

injection at the end of each block (Eq. (26)). Because the same dual-branch learner $A(\cdot)$ (RecRAN+NoRAN; Eq. (23)) is invoked repeatedly (per `RWAN` layer for gating and per block for injection) and includes recursive modulation and multi-descriptor pooling, `HySCAN` introduces extra parameters and intermediate activations whose cost scales with channel width and, especially, feature-map resolution—explaining the substantially larger `FLOP`/memory growth for `ResNet-50` on `ImageNet`-scale inputs.

A.10.6. EXTENDED DETAILS ON FIGURE 3

Robust accuracy (i.e., empirical robustness) under $\ell_\infty$ PGD stress tests on CIFAR-10 and CIFAR-100, where we strengthen the adversary along two axes: (i) increasing the perturbation budget ($\epsilon$ sweep) from 0.01 to 0.08 and (ii) increasing the number of PGD iterations (step sweep) from 10–100. Across these settings, `HySCAN` remains the most robust among the compared defenses, and the gap is most visible in the largest-$\epsilon$ and highest-step regimes.

## A.11. Ablation Study

This section isolates **which components of `HySCAN` are responsible for the gains in certified and empirical robustness**, and—crucially—*why* those components matter given `HySCAN`'s hybrid design (implicit weight-space stochasticity + explicit feature-space stochasticity).

A.11.1. EXPERIMENTAL PROTOCOL AND ABLATION RATIONALE

We ablate `HySCAN` along the two axes that define the method:

1. **Block-level hybridization (`RWAN vs. SANI`)**:
   `HySCAN`'s core claim is that *distributed* stochasticity—**implicit** randomization in the operator (`RWAN`) plus **explicit** randomization in the representation (`SANI`)—is necessary to improve both (i) **RS-certified** robustness and (ii) **empirical** robustness under strong $\ell_\infty$ attacks.
   Therefore Table 9(a) disables `RWAN` and `SANI` **independently and jointly**, holding everything else fixed, to attribute gains cleanly.

2. **Branch-level structure inside `SANI` (`RecRAN vs. NoRAN`)**:
   `SANI` is intentionally *not* a single noise injector: it fuses **progressive recursive stochastic modulation** (`RecRAN`) and **single-pass diverse channel-aware perturbations** (`NoRAN`). These branches target different failure modes (over-smoothing vs. insufficient diversity / localized stochasticity).
   Table 9(b) therefore ablates each branch to verify (i) neither is redundant and (ii) the fusion is justified.

**Setting.** All ablations are evaluated on **CIFAR-10** with **Gaussian smoothing** $\sigma = 0.25$, reporting:

- **Certified accuracy at preset radius** $r = 0.75$ (`RS`-style certificate; depends on the top-two probability gap under the *full joint randomness*), and

| Components | | Accuracy | |
|---|---|---|---|
| RWAN | SANI | Certified (r = 0.75) | PGD-20 ($\epsilon = 8/255$) |
| ✗ | ✗ | 26.1 | 59.8 |
| ✗ | ✓ | 37.5 | 67.9 |
| ✓ | ✗ | 40.1 | 69.7 |
| ✓ | ✓ | 45.5 | 74.3 |

**(a) Ablations of `RWAN` and `SANI` modules**

| Components | | Accuracy | |
|---|---|---|---|
| RecRAN | NoRAN | Certified (r = 0.75) | PGD-20 ($\epsilon = 8/255$) |
| ✗ | ✗ | 40.1 | 69.7 |
| ✗ | ✓ | 42.4 | 72.5 |
| ✓ | ✗ | 43.9 | 73.2 |
| ✓ | ✓ | 45.5 | 74.3 |

**(b) Ablations of `RecRAN` and `NoRAN` (within `SANI`)**

*Table 9.* Ablation studies of integrated components of the `HySCAN` block on `CIFAR-10` dataset for certified accuracy ($r = 0.75$) and empirical robustness (i.e., `PGD-20` ($\epsilon = 8/255$)): (a) ablations of RWAN and SANI modules on certified accuracy ($r = 0.75$) and empirical robustness (i.e., `PGD-20` ($\epsilon = 8/255$)); (b) ablations of RecRAN and NoRAN modules (within SANI). Here, we have used $\sigma = 0.25$. Note: All other components are remain fixed.

- **Empirical robust accuracy under `PGD-20` at** $\epsilon = 8/255$ (a standard $\ell_\infty$ threat model on `CIFAR-10`).

This $(\sigma, r)$ pair is a practically informative regime: it is challenging enough that certification is non-trivial, yet not so large that all methods collapse—making component contributions measurable rather than saturated.

A.11.2. RWAN vs. SANI: BOTH ARE NECESSARY, AND THEY ARE COMPLEMENTARY

Table 9(a) shows a clear, monotonic story:

- **No `RWAN`, no `SANI`**: This is the baseline without `HySCAN`'s internal stochastic mechanisms (i.e., `RS` style).

- **`SANI` only.** SANI injects **attention-modulated feature noise** at the end of each block, explicitly smoothing intermediate representations. Under randomized smoothing, certification improves when the model's class-probability gap $(p_A - p_B)$ under the joint randomness is enlarged; feature-space smoothing increases invariance of learned features to both Gaussian corruption and small adversarial shifts, thereby making $p_A$ more stable and reducing mass leakage to competing classes. Empirically, the same mechanism makes the loss landscape less exploitable by $\ell_\infty$ PGD because adversarial perturbations must remain effective *despite* stochastic feature perturbations.

- **`RWAN` only.** RWAN injects **attention-gated stochasticity directly into convolutional weights**, creating an input-conditioned, heteroscedastic "implicit ensemble" of operators across forward passes. This tends to be particularly beneficial for certification because it reduces brittle reliance on a single deterministic set of filters: the smoothed classifier aggregates predictions over many perturbed operators, and a more stable operator distribution increases the probability gap underlying the RS certificate. For empirical attacks, weight perturbations disrupt gradient estimation because the effective filters vary across queries, forcing PGD to optimize against a moving target rather than a fixed deterministic operator.

- **`HySCAN` (`RWAN` + `SANI`).** Relative to the no-`HySCAN` baseline, this is **+19.4 pp certified** and **+14.5 pp PGD**.

Why hybridization matters? The complete model is best because RWAN and SANI operate on *different* failure modes:

- RWAN primarily hardens the **operator** (filters) by injecting structured randomness into *how features are computed*.

- SANI primarily hardens the **representation geometry** by injecting structured randomness into *what features look like*.

This ablation study is the motivation for a hybrid design: **single-location stochasticity is helpful but incomplete**, whereas distributing stochasticity across weights *and* features yields the strongest—and most consistent—robustness improvements.

A.11.3. (RECRAN vs. NORAN INSIDE SANI: EACH BRANCH CONTRIBUTES, AND THE FUSION IS JUSTIFIED

Table 9(b) starts from the RWAN-only configuration and then selectively enables the two SANI branches:

- **`NoRAN` only.** NoRAN introduces **diverse, channel-aware stochastic perturbations in a single pass**, using multiple global pooling descriptors to drive complementary noise patterns. This diversity is particularly relevant for empirical

robustness: PGD must craft perturbations that survive not just one noise pattern but a distribution of channel-structured perturbations.

- **RecRAN only.** RecRAN performs **progressive stochastic modulation** across multiple recursive steps, repeatedly applying attention-modulated perturbations with fresh randomness. This "multi-step internal smoothing" naturally aligns with certification: it encourages stability of the representation under repeated stochastic transformations, which translates into a larger probability gap under the joint noise used by certification. Empirically, recursion also makes optimization harder because gradients must remain useful across a sequence of stochastic modulations rather than a single perturbation.

- **RecRAN + NoRAN (SANI).** The combined improvement beyond either branch alone validates the design choice to **fuse two qualitatively different stochastic mechanisms**. RecRAN provides *progressive smoothing* (stability), while NoRAN provides *diverse perturbation directions* (coverage). Their combination yields a strictly stronger defense because it improves robustness to both (i) distributional smoothing noise used by certification and (ii) worst-case $\ell_\infty$ perturbations optimized by PGD.

Finally, this ablation supports the motivation for SANI's **normalized, learnable fusion** of branch increments: the model benefits from *both* sources of stochasticity and can learn their relative contributions without uncontrolled noise amplification.

### A.11.4. INTERNAL WEIGHT-SPACE NOISE ABLATION

Figure 13 isolates the role of RWAN's internal weight-space noise. When $\sigma_w = 0$, HySCAN keeps the remaining architecture but removes the stochastic kernel perturbation. The certified frontier drops across radii and smoothing levels, showing that the weight noise is not merely decorative. Mechanistically, RWAN creates an input-conditioned distribution over effective convolutional filters; the smoothed classifier then aggregates predictions over this operator distribution together with Gaussian input noise. Removing this source of randomness reduces the stability of the top-class probability gap and weakens the resulting certificate.

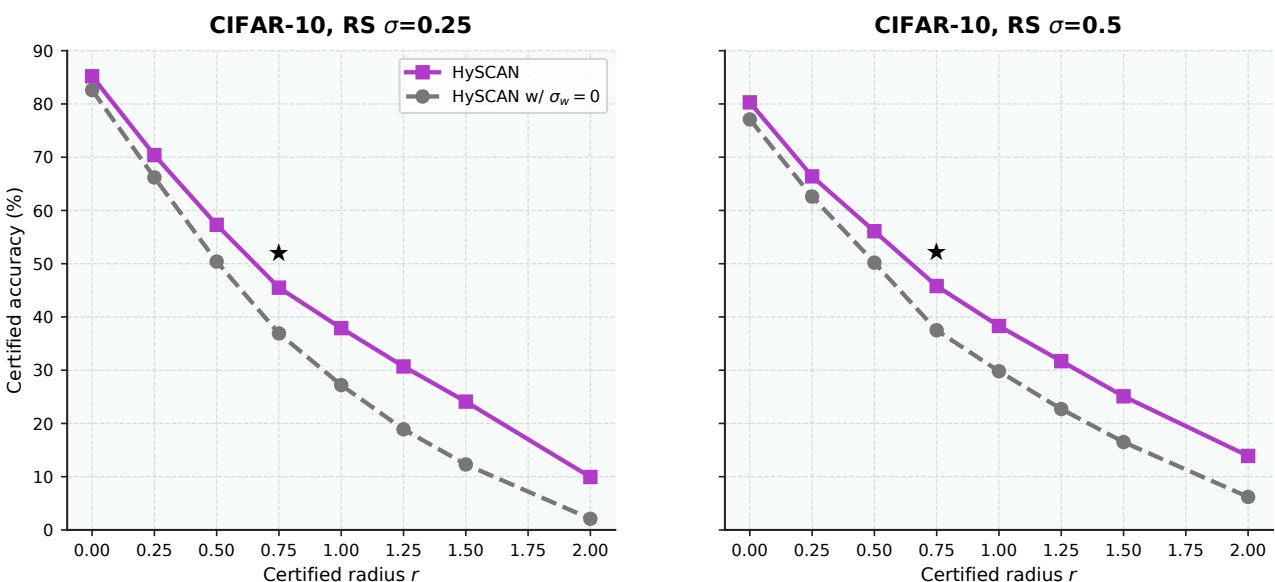

*Figure 13.* Internal weight-space noise ablation on CIFAR-10. Removing RWAN's weight-space noise by setting $\sigma_w = 0$ lowers the certified frontier for both $\sigma = 0.25$ and $\sigma = 0.50$. This shows that attention-noise-induced weight-space stochasticity is an essential part of HySCAN's certified robustness.

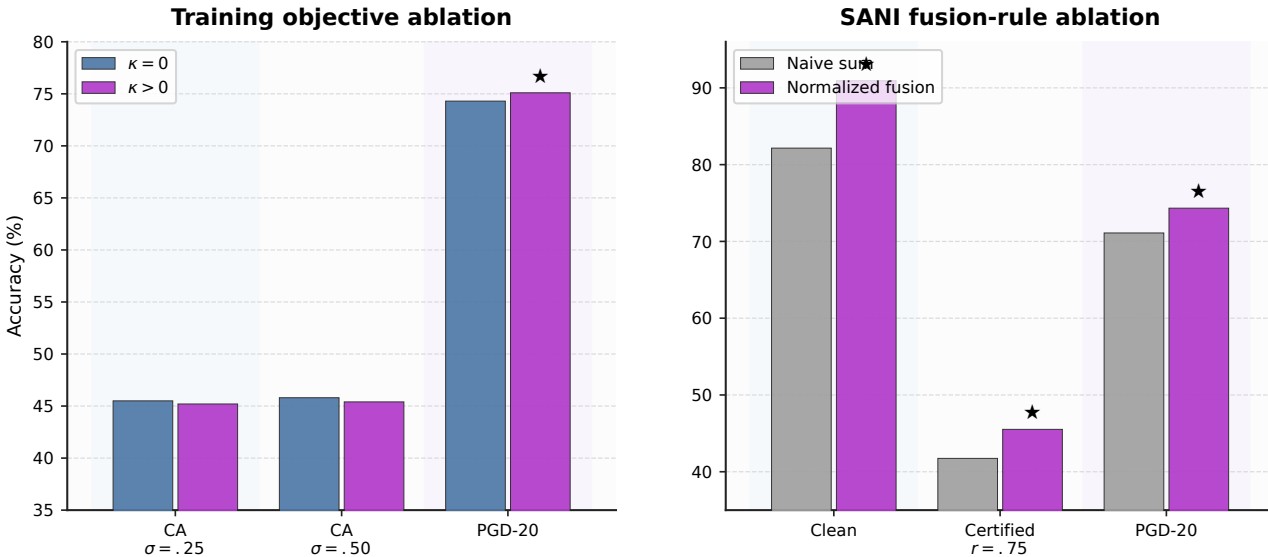

*Figure 14.* Objective and fusion-rule ablations on `CIFAR-10`. Left: $\kappa > 0$ improves empirical `PGD-20` robustness while preserving nearly the same certified accuracy as the $\kappa = 0$ `RS`-compatible objective. Right: normalized `SANI` fusion improves clean accuracy, certified accuracy, and `PGD-20` robustness over a naive unnormalized sum.

### A.11.5. ABLATION ON TRAINING OBJECTIVE AND FUSION RULE

Figure 14 shows two choices. First, the $\kappa > 0$ objective adds adversarial training pressure to the `RS`-compatible loss. This slightly prioritizes empirical $\ell_\infty$ robustness while keeping certified accuracy nearly unchanged, which is the intended certified–empirical trade-off. Second, normalized `SANI` fusion is critical: a naive sum increases perturbation magnitude without scale control, harming clean accuracy and weakening the robustness–utility balance. The normalized fusion keeps the `RecRAN`/`NoRAN` branch increments in a scale-controlled sub-convex mixture, matching Lemma 3.5 and Proposition 3.6.

### A.12. Discusion

`HySCAN` is motivated by a persistent and practically relevant mismatch in robustness research: the most scalable *provable* defenses for modern vision models (randomized smoothing) predominantly certify under an $\ell_2$ threat model, while *deployment-facing* adversarial evaluations are routinely performed under strong $\ell_\infty$ attacks (e.g., `APGD`/`AutoAttack`). `HySCAN`'s central design hypothesis is that narrowing this gap requires *distributed stochasticity* that is (i) active at both training and inference, (ii) *structured* rather than indiscriminate, and (iii) incorporated into certification via the *joint randomness* formulation (Eq. 2). Concretely, `HySCAN` couples **implicit weight-space stochasticity** (`RWAN`) with **explicit feature-space stochasticity** (`SANI`), where both are *attention-conditioned* and *resampled per forward pass*. Below we discuss (i) why this hybridization improves both certified and empirical robustness, (ii) what the experimental evidence reveals, (iii) asymptotic/computational considerations, and (iv) broader social impact implications.

### A.12.1. WHY HYBRID SPACE-AWARE STOCHASTICITY IMPROVES BOTH CERTIFIED AND EMPIRICAL ROBUSTNESS

**Certificates remain valid under internal randomization, and can become stronger when randomness is modeled jointly.** A frequent pitfall in stochastic defenses is to add internal noise as a heuristic while certifying only with respect to input noise, thereby creating a mismatch between the certified distribution and the deployed predictor. `HySCAN` avoids this by *defining* the smoothed predictor as the marginal over the *full* randomness sources: Gaussian input smoothing noise and `HySCAN`'s internal randomness (`RWAN`/`SANI`). Since randomized smoothing guarantees depend only on the induced class probabilities under the smoothing distribution (Theorem 3.1), the certificate remains valid when the base classifier is stochastic, provided the certification procedure estimates the top-two class probabilities under the *joint* randomness (Eq. 2). In other words, certification and inference correspond to the same stochastic predictor, ensuring that reported radii remain meaningful and deployment-consistent.

**RWAN and SANI target complementary failure modes: "how features are computed" vs. "what representations look like."** Empirically strong adversaries often exploit brittle reliance on a small set of deterministic filters and unstable intermediate features. HySCAN addresses these vulnerabilities at two distinct loci:

- **RWAN perturbs the operator.** RWAN injects attention-gated Gaussian weight perturbations into convolution kernels, yielding an input-conditioned ensemble of operators sampled on each forward pass. The normalized gate (Lemma 3.3) constrains perturbation magnitude, and the induced heteroscedasticity (Proposition 3.4) discourages over-reliance on a single deterministic kernel realization.

- **SANI perturbs the representation.** SANI injects additive attention-modulated noise into block outputs, explicitly smoothing intermediate geometry while preserving discriminative signal through attention conditioning. Its nonnegative, normalized mixing (Lemma 3.5) and the scale-control property (Proposition 3.6) prevent uncontrolled amplification of injected noise and help maintain stable feature magnitudes.

This separation is crucial: *weight-only* randomization can still allow adversaries to exploit representation-level invariants that persist across filter samples, while *feature-only* randomization can be averaged out by sufficiently strong stochastic-gradient attacks. The most consistent gains arise when both mechanisms are active, supporting the view that robustness benefits from simultaneously hardening *operators* and *representations*.

**A mechanistic lens: probability-gap enlargement for certification and gradient dispersion for empirical robustness.** From the certification perspective, the $\ell_2$ radius (Eq. 4) grows with the gap between the top and runner-up class probabilities under the smoothing distribution. HySCAN's structured stochasticity can increase this gap by making correct-class predictions more stable under joint randomness while suppressing probability leakage to competing classes. From the empirical perspective, HySCAN increases *directional gradient dispersion* under resampled internal randomness (Proposition 3.2), which weakens the attacker's ability to follow a single stable descent direction. Importantly, HySCAN is evaluated in the recommended regime for stochastic defenses: internal randomness is active at test time and resampled across attack iterations, so the attacker must optimize under a *moving* stochastic landscape rather than a fixed deterministic network. The persistence of robustness under stronger PGD sweeps (larger step counts and perturbation budgets; Fig. 3) supports that improvements are not simply an artifact of weak evaluation.

A.12.2. WHAT THE EXPERIMENTS SHOW, AND WHAT THEY IMPLY

**Certified robustness: HySCAN shifts the $\sigma$–$r$ frontier upward.** Across natural-image benchmarks, HySCAN consistently yields the strongest certified accuracy over a wide range of $(\sigma, r)$ operating points (Table 1). Two experimental patterns are especially diagnostic:

**(i) The low-noise / high-radius regime is where many methods collapse.** At $(\sigma, r) = (0.25, 2.0)$, HySCAN is the only method with non-zero certified accuracy on both ImageNet (7.15%) and CIFAR-10 (9.94%). This regime is informative because small $\sigma$ typically preserves clean accuracy but makes large-radius certification difficult; HySCAN's non-zero performance suggests that *internal, structured* randomization improves stability beyond what input-only smoothing achieves.

**(ii) Gains persist across radii rather than concentrating at a single operating point.** For CIFAR-10 at $\sigma = 0.25$, HySCAN reaches 30.7% certified accuracy at $r = 1.25$ versus 21.1% for ARS (+9.6 pp), and 24.1% at $r = 1.5$ versus 16.2% (+7.9 pp). For ImageNet at $\sigma = 0.25$, HySCAN improves certified accuracy at $r = 1.0$ (42.3% vs. 39.1%) and remains higher up to $r = 1.5$ (30.7% vs. 26.7%). These consistent improvements indicate that HySCAN shifts the certified robustness frontier upward rather than trading off performance at one radius for another.

**Generalization beyond natural images is a salient empirical strength.** On NIH-CXR, EyePACS, HAM10000, and NCT-CRC-HE-100K, HySCAN improves certified accuracy across $\sigma \in \{0.25, 0.50, 1.0, 2.0\}$ and across radii (Table 2), including higher-radius settings where discriminative cues can be subtle. For example, at $(\sigma, r) = (1.0, 1.0)$ HySCAN achieves 42.9% on NIH-CXR and 39.5% on HAM10000, outperforming ARS by +8.8 and +4.9 pp, respectively. These results suggest that attention-conditioned stochasticity remains beneficial even when salient features are localized and dataset statistics differ substantially from natural-image benchmarks.

**Empirical robustness: strong $\ell_\infty$ robustness while preserving clean accuracy.** HySCAN also demonstrates strong empirical robustness against modern $\ell_\infty$ attack suites (APGD-20 and AA-20) on both natural-image and medical benchmarks

(Tables 3–4; Table 6). Notably, HySCAN sustains robust accuracy in the range of roughly 55.2–90.7% across the medical datasets while maintaining high clean accuracy.

**Medical benchmarks: consistent gains over strong empirical defenses.** On EyePACS under AA-20 at $\epsilon = 8/255$, HySCAN attains 73.1% robust accuracy compared to 69.7% for CTRW (+3.4 pp) and 65.3% for DCS (+7.8 pp). On NIH-CXR at $\epsilon = 16/255$ under APGD-20, HySCAN reaches 78.1% versus 73.1% for CTRW (+5.0 pp). Together with the reported standard deviations, these improvements indicate reproducible benefits rather than seed-specific effects.

**Natural-image benchmarks: robustness gains persist under strong evaluation.** On CIFAR-10 at $\epsilon = 8/255$, HySCAN achieves 90.93% clean accuracy with 74.63% robust accuracy under PGD-20 and 78.47% under AA-20 (Table 6). On ImageNet at $\epsilon = 8/255$, HySCAN achieves 74.24% clean accuracy with 60.69% (PGD-20) and 68.32% (AA-20). These results are consistent with the paper's core hypothesis: structured internal stochasticity can improve robustness under strong $\ell_\infty$ attacks while remaining compatible with smoothing-based certification.

**Robustness under stronger PGD does not collapse abruptly.** The stronger-PGD sweeps (Fig. 3) show HySCAN remains consistently above competitors as attack strength increases. This is a practical stress test against "fragile" stochastic defenses; the persistence of the gap at larger step counts is consistent with HySCAN increasing the difficulty of first-order optimization rather than merely confounding weak attacks.

**Ablations validate the hybrid design and the two-branch SANI structure.** The ablations provide a clean causal explanation: removing either source of stochasticity weakens robustness, and combining them is strictly better than either alone (Table 9). On CIFAR-10 at $\sigma = 0.5$ (certified at $r = 0.75$) and PGD-20 at $\epsilon = 8/255$, the progression *(no RWAN/no SANI) → (SANI only) → (RWAN only) → (RWAN+SANI)* is monotone for both certified and empirical robustness. Within SANI, RecRAN and NoRAN ablations further show that combining progressive smoothing (RecRAN) with diverse channel-aware perturbations (NoRAN) yields the most reliable gains, aligning with the design rationale that these branches address distinct failure modes (over-smoothing vs. insufficient diversity).

### A.12.3. ASYMPTOTIC ANALYSIS AND COMPUTATIONAL CONSIDERATIONS

**What must be accounted for.** A correct cost model for HySCAN must reflect that RWAN's channel-wise weight-perturbation gate is produced by the same SANI attention–noise learner used for explicit feature injection: SANI provides the attention–noise responses used by RWAN to gate weight perturbations, and SANI also injects feature-space noise at the end of each HySCAN block. This implies that HySCAN invokes the dual-branch learner $A(\cdot)$ repeatedly: *per RWAN layer for gating* and *per block for injection*. This repeated invocation is highlighted as the architectural reason for the measured overhead in Appendix A.10.5 and Table 8. *Therefore, the dominant overhead is not merely sampling $\Delta W$, but repeatedly evaluating SANI (RecRAN+NoRAN)* (See Eqs. (12)–(15), Eq. (24), Eq. (26); Appendix A.10.5.).

**Notation.** Let $u \in \mathbb{R}^{H \times W \times C}$ be a feature map. Consider a dense convolution with kernel $W \in \mathbb{R}^{K \times K \times C_{\text{in}} \times C_{\text{out}}}$ producing an output of spatial size $H \times W$. We write $\text{MLP}(C; d)$ for the arithmetic cost of the channel MLP $m(\cdot)$ used in attention computation when applied to a $C$-dimensional descriptor with hidden width $d$ (e.g., a two-layer bottleneck MLP would satisfy $\text{MLP}(C; d) = \Theta(Cd + dC)$).

**Baseline convolution.** A standard dense convolution costs

$$\text{Cost}_{\text{conv}} = \Theta\left(HW\,K^2\,C_{\text{in}}C_{\text{out}}\right).$$

**RWAN: kernel noise is cheap; the per-layer SANI gate is the main overhead.** RWAN replaces each convolution with a stochastic kernel $\widetilde{W} = W + \Delta W$ whose perturbation is Gaussian and scaled by a learned noise-strength and an input-dependent gate (Eqs. (12)–(15)). Its additional costs decompose into:

- **Kernel-noise sampling and elementwise mixing.** Sampling $\Delta W \in \mathbb{R}^{K \times K \times C_{\text{in}} \times C_{\text{out}}}$ and forming $\widetilde{W}$ adds

$$\Theta\left(K^2\,C_{\text{in}}C_{\text{out}}\right)$$

random draws and elementwise operations, typically lower order than $\text{Cost}_{\text{conv}}$.

- **Gate pooling/normalization.** `RWAN` pools the `SANI` response to a per-channel gate (mean over batch and spatial dimensions, followed by max-absolute normalization; ( Eqs. (13)–(14), adding $\Theta(HW\,C_{\mathrm{in}})$.

- **Gate computation via `SANI`.** Crucially, computing the gate requires evaluating `SANI`'s dual-branch learner $A(u;\psi)$ (`RecRAN`+`NoRAN`; Eq. (20)), and this evaluation occurs *for every RWAN layer* (Eq. (24)), in addition to `SANI`'s *explicit block-level injection* (Eq. (26);Appendix A.10.5).

Thus, for a single `RWAN`-convolution layer,

$$\mathrm{Cost}_{\mathrm{RWAN}} \;=\; \mathrm{Cost}_{\mathrm{conv}} \;+\; \Theta\!\left(K^2\,C_{\mathrm{in}}C_{\mathrm{out}} + HW\,C_{\mathrm{in}}\right) \;+\; \mathrm{Cost}_{\mathrm{SANI}}(H,W,C_{\mathrm{in}}).$$

**`SANI`: dual-branch attention–noise learner (`RecRAN` + `NoRAN`).** Given $u \in \mathbb{R}^{H\times W\times C}$, `SANI` forms stochastic attention noise $A(u;\psi)$ as a normalized nonnegative mixture of a *recursive* branch (`RecRAN`) and a *non-recursive* branch (`NoRAN`) (Eq. (20)), producing residual increments relative to $u$ (Eq. (20)–(21)). This same learner is used (i) to provide `RWAN` gating responses and (ii) to perform explicit feature-space injection at block ends.

**`RecRAN` cost.** `RecRAN` first computes deterministic local and channel attentions (Eq. (43)):

$$\alpha_\ell = \sigma(\mathrm{Con}_{1\times 1}(u)), \qquad \alpha_c = \sigma(m(\mathrm{GAP}(u) + \mathrm{GMP}(u))),$$

then performs $T$ recursive steps (we use $T = 5$) with freshly resampled noises and elementwise refinement/masking (Eqs. (44)–(46)). A faithful cost decomposition is:

$$\mathrm{Cost}_{\mathrm{RecRAN}} = \underbrace{\Theta(HW\,C^2)}_{1\,\times\,1\text{ conv from }C\text{ to }C} + \underbrace{\Theta(HW\,C)}_{\mathrm{GAP/GMP\ pooling}} + \underbrace{\mathrm{MLP}(C;d)}_{\text{channel MLP}} + \underbrace{\Theta(T\,HW\,C)}_{T\text{ rounds of elementwise refinement/masks}}.$$

(The $\Theta(HW\,C^2)$ term follows from the dense $1 \times 1$ convolution that outputs a $C$-channel attention map.)

**`NoRAN` cost.** `NoRAN` uses multiple global pooling descriptors $D = \{\mathrm{MN}, \mathrm{SP}, \mathrm{MP}, \mathrm{AP}\}$ (min/sum/max/avg pooling) and, for each $d \in D$, computes a channel attention $\alpha_d = \sigma(m(d(u)))$ and injects noise via elementwise modulation (Eqs. (49)–(51)). With $|D| = 4$ in the paper, we obtain:

$$\mathrm{Cost}_{\mathrm{NoRAN}} = \underbrace{\Theta(|D|\,HW\,C)}_{\text{pooling and elementwise injections}} + \underbrace{|D| \cdot \mathrm{MLP}(C;d)}_{\text{per-descriptor channel MLP}}.$$

**`SANI` cost summary.** Up to lower-order mixture/normalization terms, a single call to `SANI` satisfies:

$$\mathrm{Cost}_{\mathrm{SANI}}(H,W,C) = \Theta(HW\,C^2) + \Theta((T + |D|)\,HW\,C) + (1 + |D|)\,\mathrm{MLP}(C;d).$$

This expression directly reflects the module definitions: a dense $1 \times 1$ attention convolution (`RecRAN`), $T$ recursive elementwise refinement steps (`RecRAN`), and $|D|$ pooling descriptors with per-descriptor channel attentions (`NoRAN`).

**Block-level scaling: why the overhead multiplies.** If a `HySCAN` block contains $J$ convolutions, `SANI` is invoked $J$ times for `RWAN` gating and once more for explicit feature injection at block end. Therefore, the per-block cost is approximately:

$$\mathrm{Cost}_{\mathrm{block}} \approx \sum_{j=1}^{J} \mathrm{Cost}_{\mathrm{conv}}^{(j)} + (J+1)\,\mathrm{Cost}_{\mathrm{SANI}}(H,W,C_j) + \sum_{j=1}^{J} \Theta\!\left(K_j^2 C_{\mathrm{in},j} C_{\mathrm{out},j} + HW\,C_{\mathrm{in},j}\right).$$

This makes explicit the multiplicative effect of *per-layer gating* plus *per-block injection*.

**Implications for training, evaluation, and certification.** *(a) Adversarial training (empirical $\ell_\infty$ robustness).* `HySCAN`'s internal randomness is part of the deployed predictor and is resampled per forward pass; during `PGD`-based adversarial example generation, each gradient step uses a fresh draw of $(\omega, \psi)$. Consequently, $K$-step `PGD` multiplies the forward/backward cost by roughly $K$ under this resampling protocol.

**(b) `RS` certification (certified $\ell_2$ robustness).** Inference-time certification samples from the *full joint randomness*: each query draws fresh Gaussian input noise and freshly resampled `RWAN`/`SANI` randomness (Algorithm 2). With Monte Carlo sample counts $(n_0, n)$, the per-example certification cost is

$$\Theta((n_0 + n) \cdot \text{Cost}_{\text{forward}}),$$

where $\text{Cost}_{\text{forward}}$ is the full `HySCAN` forward-pass cost (including `RWAN` and `SANI`). Several datasets use the default $(n_0, n, \alpha) = (100, 100{,}000, 0.001)$, which makes certification expensive by construction.

**Measured overhead aligns with this accounting.** Table 8 reports that integrating `HySCAN` into `ResNet-50` on ImageNet-scale inputs increases `FLOPs` from 3.8G to 179.03G ($47.1\times$) and inference time from 2.29ms to 15.7ms, consistent with the repeated invocation of the attention–noise learner for per-layer gating plus per-block injection. On `CIFAR`-scale `ResNet-110`, `FLOPs` increase only mildly (8.10G to 8.75G), but activation memory and inference time still increase substantially, reflecting the additional intermediate activations introduced by `SANI` and recursion.

**Opportunities for efficiency.** This decomposition suggests direct efficiency levers that do not change `HySCAN`'s certification semantics: reducing `RecRAN` recursion depth $T$, computing gates at coarser granularity (e.g., per block/stage rather than per layer), or selectively enabling `HySCAN` in later low-resolution stages where $HW$ is smaller.

A.12.4. DISCUSSION ON SOCIAL IMPACT IMPLICATIONS OF `HySCAN`

`HySCAN` is motivated by the practical security and reliability challenges posed by adversarial perturbations in modern vision systems, which are especially concerning in *high-stakes* settings (e.g., clinical decision support) and other real-world decision pipelines. In such contexts, robustness failures can translate into degraded trust, safety concerns, or increased operational risk.

**Potential positive impact: improved robustness where reliability matters.** `HySCAN`'s central goal is to narrow the gap between (i) *provable* robustness under randomized-smoothing $\ell_2$ certificates and (ii) *practical* robustness under strong $\ell_\infty$ adversarial evaluations, by distributing freshly resampled, attention-modulated stochasticity throughout the network. Empirically, `HySCAN` is evaluated not only on natural-image benchmarks but also across multiple medical-imaging datasets, where robustness is frequently reported to be more challenging. In these domains, stronger robustness can reduce susceptibility to both malicious manipulations and accidental small perturbations, and can serve as an additional safeguard when vision models are used as decision-support components.

**Scope and responsible interpretation of guarantees.** `HySCAN` provides formal guarantees only under the certification setting it is evaluated with (randomized-smoothing $\ell_2$ certification at a chosen noise level and confidence), while empirical robustness is assessed under $\ell_\infty$ attack suites. For responsible communication and deployment, it is important to present certificates as *sufficient* (and often conservative) guarantees within the certified threat model, rather than as a blanket guarantee against all perturbations (e.g., semantic or physical-world attacks). In safety-critical deployments, `HySCAN` should therefore be treated as one component in a defense-in-depth strategy (including monitoring, validation under domain-specific corruptions, and human oversight where appropriate).

**Dual-use considerations.** Robustness methods can be dual-use: techniques that harden beneficial systems can also harden harmful systems against auditing or detection. To mitigate misuse, releases should emphasize clear threat-model documentation, evaluation protocols (including how randomness is handled during attack), and known operating assumptions, enabling downstream users to adopt the method with appropriate safeguards.

**Environmental and accessibility considerations.** `HySCAN`'s architecture-level stochasticity increases compute and memory relative to a vanilla backbone, which can raise the energy cost of training and the cost of Monte Carlo certification. This has implications for the environmental footprint of robustness research and for accessibility. Continued work on compute-efficient variants and more sample-efficient certification would improve the societal footprint and broaden participation.

