# OpenReview forum: "Does a Hybrid Space-Aware Randomized Defense Improve Empirical and Certified Adversarial Robustness?"
_ICML.cc/2026/Conference — ICML 2026 regular_

### Official Review · Reviewer_2UE1 · 2026-02-17

**Soundness:** 3
**Presentation:** 3
**Significance:** 2
**Originality:** 2
**Overall Recommendation:** 4
**Confidence:** 2

**Summary:**

Focusing on both empirical and certified robustness, this paper proposes Hybrid Space-aware Stochastic Convolution Attention Noise(HySCAN), a hybrid randomized adversarial defense framework that injects internal randomness at two complementary levels: implicit weight-space perturbations and explicit feature-space noise smoothing. HySCAN enables meaningful certified guarantees while improving empirical robustness in practice. The experiments on multiple datasets and attack scenarios show that the proposed method is effective in improving both empirical and certified robustness.

**Compliance With Llm Reviewing Policy:**

Affirmed.

**Final Justification:**

W1–W3 have been resolved, while W4 has been partially addressed. I believe fully resolving this issue would likely necessitate substantial revisions, and the current explanation remains insufficient for a complete assessment. So, I currently maintain a borderline assessment of the paper.

**Key Questions For Authors:**

See Weaknesses.

**Limitations:**

See Weaknesses.

**Strengths And Weaknesses:**

Strengths:
1. The goal of the paper is clearly stated: improving both empirical and certified robustness, reliably across both natural image and medical-vision benchmarks.

2. The proposed method is evaluated on multiple datasets and attack scenarios.

3. The paper includes ablation studies and analysis to support the choices of the integrated components.



Weaknesses:
1. Lack of some baseline on empirical robustness. For example, the current diffusion-based method is very powerful for improving empirical robustness, and it would be helpful to include a comparison with some recent diffusion-based methods.

2. Lack of inference-time comparison with other methods. Although the paper reports the inference time of HySCAN, it does not provide a direct comparison with other methods. In addition, I would like to clarify whether the inference time reported in Table 8 corresponds to the total processing time for a single adversarial example (including all stochastic sampling procedures), or only the time required for a single forward inference pass.


3. Missing evaluation under EOT for randomness. Since the proposed method introduces randomness factors, it is essential to evaluate robustness using the EoT attack. The authors should provide results under EOT attack to more comprehensively evaluate the empirical robustness of the method and to discuss the effects of obfuscated gradients.

4. The inclusion of medical image benchmarks is a good point of the paper. However, the current evaluation remains somewhat limited. Medical images require stricter fidelity preservation compared to natural images. In natural image classification, losing some details or semantic information may still be acceptable, whereas in medical imaging, subtle details can be clinically significant. So, I believe that the authors should provide additional analysis of image quality preservation on medical datasets, along with qualitative visualizations, to better understand how the proposed method affects these key information.

---

> ### Author Rebuttal · Authors · 2026-03-29
>
> Thank you, Reviewer **2UE1** for the opportunity to address the **weaknesses (W)**.
>
> **W1. (Comparison with Diffusion).** Thank you for the suggestion. We added recent diffusion-based purification baselines (e.g., FreqPure [1] and AGDM [2]) in Table **R4-Tab-1**. HySCAN still achieves the best robust accuracy across all reported attack settings: on CIFAR-10, 74.63/78.47 under PGD-20/AA-20 and 73.2/72.9 under stronger and adaptive attacks [3] e.g., EOT-PGD/BPDA+EOT [4, 5]; on ImageNet, 60.69/68.32 under PGD-20/AA-20. Thus, HySCAN remains stronger than such prior diffusion baselines in adversarial robustness, while remaining competitive in clean accuracy.
>
> **Table R4-Tab-1: Empirical robustness (%) under PGD, AutoAttack (AA), EOT-PGD, and BPDA+EOT [4, 5] on CIFAR-10 (C10) and ImageNet (IN) datasets with $\epsilon = 8/255$.**
>
> | Pipeline | Clean | C10@PGD | C10@AA | C10@BPDA+EOT | C10@EOT-PGD | Clean | IN@PGD | IN@AA |
> | - | -: | -: | -: | -: | -: | -: | -: | -: |
> | FreqPure [1] | 92.19 | 71.39 | 77.35 | 72.42 | 70.84 | 71.88 | 59.77 | 64.91 |
> | AGDM [2] | 90.82 | 70.12 | 76.61 | 71.88 | 68.27 | 68.75 | 54.82 | 61.30 |
> | **HySCAN** | **90.93** | **74.63** | **78.47** | **72.9** | **73.2** | **74.24** | **60.69** | **68.32** |
>
> **W2 (Inference-time Comparison.)** Table **R4-Tab-2** reports the inference-time comparison on CIFAR-10. We clarify that the reported HySCAN inference time corresponds to a single stochastic forward pass per adversarial example, with both the implicit stochasticity from RWAN and the explicit stochasticity from SANI enabled during inference. We agree that HySCAN is more computationally expensive than the baselines. However, its objective is not to optimize efficiency but to improve adversarial robustness significantly for security-sensitive AI applications (e.g., clinical decision support), which often require substantial robustness that outweighs modest cost savings. The practical value of a defense method therefore depends on whether it delivers substantial robustness gains, even at high computational cost. HySCAN outperforms baselines by up to **10.44% (Tables R1-Tab-1–R1-Tab-6, & R2-Tab-1; see also Reviewers 6HY3 & DbTh)**.
>
> **Table R4-Tab-2: Training and inference time comparison on CIFAR-10.**
> | Method | Training time (min) | Inference time (ms) |
> | - | -: | -: |
> | RPF | 185.8 | 0.65 |
> | CTRW | 192.3 | 0.74 |
> | DCS | 225.8 | 0.98 |
> | **HySCAN** | **571.4** | **4.29** |
>
> **W3 (Evaluation under EOT).** To address the reviewer’s concern, we additionally evaluate HySCAN under EOT-PGD and adaptive attacks (BPDA and BPDA+EOT). Since HySCAN keeps internal randomness—implicit stochasticity via RWAN and explicit stochasticity via SANI—active at both training and inference time, with fresh randomness resampled across the inference phase. As shown in **Table R4-Tab-3**, HySCAN retains the best robust accuracy on both NIH-CXR and CIFAR-10, improving over prior baselines by up to +9.5. These results alleviate the reviewer’s concern regarding obfuscated gradients. **For further discussion, see **Reviewer v96S** W2 response.**
>
> **Table R4-Tab-3: Empirical robustness (%) under EOT-PGD (EP) & adaptive attacks (BPDA and BPDA+EOT (BE)) on NIH-CXR (NC) & CIFAR-10 (C10).**
>
> | Pipeline | NC@EP | NC@BPDA | NC@BE | C10@EP |C10@BPDA | C10@BE |
> | - | -: | -: | -: | -: | -: | -: |
> | HyCAS [6] | 83.6 | 82.2 | 78.3 | 68.4 | 68.1 | 66.5 |
> | DCS | 81.1 | 76.7 | 74.9 | 69.8 | 69.3 | 67.6 |
> | RPF | 82.2 | 77.3 | 75.3 | 66.1 | 67.9 | 65.4 |
> | CTRW | 81.1 | 76.7 | 74.9 | 68.8 | 69.4 | 67.4 |
> | **HySCAN** | **88.3** | **85.3** | **84.4** | **73.2** | **73.7** | **72.9** |
>
> **W4. (Qualitative Analysis).** **Figure R4-Fig-1** (see https://github.com/Epimone/Rebuttal-HySCAN) provides a qualitative example on NIH-CXR. For a chest X-ray originally labeled **No Finding**, a PGD adversarial perturbation changes the prediction to **Pneumothorax**, whereas HySCAN predicts **No Finding**. This qualitative behavior is consistent with HySCAN’s design, where RWAN and SANI introduce freshly resampled internal stochasticity at the weight and feature spaces during both training and inference. This figure complements the quantitative NIH-CXR certified and empirical robustness results reported in **Tables 2 and 3 (see main paper)**.
>
> **[1] Pei et al. Diffusion-based Adversarial Purification from the Perspective of the Frequency Domain, ICML, 2025.**
>
> **[2] Lin et al. Adversarial guided diffusion models for adversarial purification, Neural Networks, 2025.**
>
> **[3] Athalye et al. Obfuscated gradients give a false sense of security: Circumventing defenses to adversarial examples, ICML, 2018**
>
> **[4] Tramer et al. On adaptive attacks to adversarial example defenses, NeurIPS, 2020.**
>
> **[5] Hill et al. Stochastic security: Adversarial defense using long-run dynamics of energy-based models, ICLR, 2021.**
>
> **[6] Dhar, el. al. Certified vs. Empirical Adversarial Robustness via Hybrid Convolutions with Attention Stochasticity, ICLR, 2026**

---

> > ### Author Rebuttal · Reviewer_2UE1 · 2026-04-03
> >
> > Thank you for the rebuttal. W1–W3 have been resolved, while W4 has been partially addressed. I believe fully resolving this issue would likely necessitate substantial revisions, and the current explanation remains insufficient for a complete assessment. So, I currently maintain a borderline assessment of the paper.

---

> > > ### Author Response · Authors · 2026-04-08
> > >
> > > Thank you so much, **Reviewer 2UE1**, for the helpful follow-up, the **score increase to 4 (i.e., Weak Accept)**, and for confirming that W1–W3 have been resolved. On **W4**, our intent in including **NIH-CXR** and **HAM10000** was to evaluate whether HySCAN generalizes beyond natural-image benchmarks under the same certified and empirical robustness protocols used throughout the paper. The added **NIH-CXR qualitative example is complementary evidence in that direction**. We therefore interpret the **medical-benchmark results as a first-step robustness evaluation on medical data rather than a broader clinical conclusion**. We therefore fully agree that medical-image robustness should be framed as a careful **first step rather than a definitive clinical claim**, and we will revise the final version to reflect that limitation explicitly.

---

### Official Review · Reviewer_v96S · 2026-02-17

**Soundness:** 1
**Presentation:** 3
**Significance:** 2
**Originality:** 2
**Overall Recommendation:** 4
**Confidence:** 4

**Summary:**

This work proposes a convolution layer that is specialized to simultaneously (1) Improve the certified accuracy-robustness tradeo-off of randomized smoothing against $\ell_2$ pertrubations and (2) improve empirical robustness to $\ell_\infty$ perturbations.

The proposed layer consists of two components.
The first one (RWAN) randomized the weights of the convolution kernel via a learned, attention-based transformation of the original weights and random noise.
The second one (SANI) uses a similar attention-based mechanism to introduce noise into intermediate representations between layers.

The authors prove that combining this model with classic randomized smoothing preserves the usual robustness guarantees.
They further show an upper bound on the misalignment between the expected gradient (under internal model randomness) and a randomly drawn gradient at a fixed adversarial input. This is meant to theoretically justify the effectiveness of the model in improving empirical robustness.

Their experiments demonstrate improved certified accuracy (under $\ell_2$ perturbations) and empirical robustness (under $\ell_\infty$ perturbations) compared to prior work that targets these objectives individually.

**Compliance With Llm Reviewing Policy:**

Affirmed.

**Final Justification:**

See below. Major concerns addressed. My main concern remains connection between theory and empirical merits of the method. But since ICML does not allow for updates to the manuscript during the review period, this cannot really be resolved.

Overall, **I no longer think there's anything egregiously wrong with the paper that would warrant rejection**. But it is, ultimately, an empirical tweak that modifies a particular model architecture in a particular domain (CNNs, image data). It does not really advance probabilistic certification / randomized smoothing as a whole. So I don't really feel like the description for Score 5 ("High impact [...], good-to-excellent evaluation") fits here.

**Key Questions For Authors:**

How exactly do you compare your method to "RS" in Table 2. Is "RS" the same randomized smoothing approach, but with standard convolutions and no internal noise?

What would you expect to be the effect of eliminating internal noise ($\sigma_w = 0$) on certified accuracy and empirical robustness?

In Proposition 3.2, why does upper-bounding the misalignment (i.e., admitting small misalignment) provide evidence for improved empirical robustness?

**Limitations:**

**No**. Neither the apppendix nor the main text discuss limitations of the method (e.g., giving up the generality of randomized smoothing for a specific model architecture (CNNs), increased memory footprint, cost of needing to sample internal noise, lack of specialized GPU kernels compared to standard convolutions, etc.)

**Strengths And Weaknesses:**

## Strengths
* The studied problem is interesting and relevant for the adversarial robustness / safety community (improving both certified robustness and empirical robustness; improving robustness under multiple perturbation models)
* Presentation (clarity of figures, writing style, overall structure of the text) is good
* Related work is discussed in sufficient depth and detail
* The authors ablate different components of their proposed architecture
* Experiments are run with multiple seeds, standard deviations are reported.

## Weaknesses

### Main weakness
The main weakness of the paper is that it proposes a heuristic architectural improvement for better certified accuracy and empirical robustness. However, neither of these two claims is substantiated by the experimental evaluation.

**Improved certified robustness:**

The method is compared against standard randomized smoothing (Cohen et al., 2019) [1] and adaptive randomized smoothing (Lyu et al., 2019) [2]. However, this is conflating **two entirely different types of works**: Standard RS and adaptive RS are general frameworks that can be applied to arbitrary base models. This work proposes a concrete instantiation of the base model. To obtain its robustness guarantees in Section 4.1, this work directly applies standard randomized smoothing to the proposed base model (see Theorem 3.1).
**Therefore, the claim "HyScan is better than ARS / RS" from Section 4.1 is not well-founded.**

Of course, this does not preclude that this specific base model could potentially offer an improved accuracy-robustness trade-off compared to other base models (convolution-based or other), but this would require a comparison across a wide range of different model architectures. Afterall, the proposed work introduces a variety of new components that increase model capacity / expressiveness (skip connections, attention mechanisms, ...). Simply comparing to standard convolutions (as appears to be done in the ablations in Appendix A.10) is not sufficient.

**Improved empirical robustness:**

This work only evaluates empirical robustness by applying two standard (and relatively old) attack methods / toolsets to a novel empirical defense. This violates a fundamental principle of defense evaluation: **Empirical defenses have to be evaluated against adaptive attacks** [3,4,5], i.e., the authors should make an extensive and best-effort attempt to circumvent the new attack strategy. This is not done here, meaning there is no reason to believe that the proposed defense couldn't be (trivially) circumvented by a follow-up work.

### Other weaknesses

**Missing ablation of internal noise**
The proposed architecture introduces, by design, additional latent noise via parameters $\sigma_w$. Since this additional noise does not strengthen the robustness certificate, and increased noise usually worsens accuracy, conducting experiments with $\sigma_w=0$ would be necessary to substantiate the claim that internal noise improves certified and empirical robustness.

**Missing ablation of training objective**
The work proposes a new training objective. Again, not ablation is conducted to verify whether this explains the improved certified and empirical robustness.

**Superfluous theory**
The main section is padded with various Lemmata and Propositions that do not substantiate any of the main claims, do not contribute to the overall narrative, and in the case of Lemma 3.5 and Proposition 3.6 are not even discussed in the text. (Why do we need to know the conditional law of $\Delta W$? Why is it important to know that SANI is a sub-convex nonnegative mixture?, ...)

**Limitations are not discussed**.
Neither the apppendix nor the main text discuss limitations of the method (e.g., giving up the generality of randomized smoothing for a specific model architecture (CNNs), increased memory footprint, cost of needing to sample internal noise, lack of specialized GPU kernels compared to standard convolutions, etc.)

### Minor comments
* Proposition 3.2 shows that the misalignment is upper-bounded by a function of the model's internal noise. It is not clear why this should improve empirical robustness. It appears like a lower bound would be needed that a gradient-based attack is misguided by the proposed method.
* In Tables 3 and 4, the difference between HyScan and the baselines is often small relative to their respective standard deviation, i.e., there is often no strong evidence that the proposed method is actually better.
* The contribution would be stronger / be of broader interest if the internal randomness was actually used for deriving stronger robustness guarantees.
* The lengthy proof in Appendix A.2.1 is not needed, the certificates from Cohen et al. (2019) [1] (and the underlying Neyman-Pearson lemma) already admit random base classifiers.

## Conclusion
To summarize the work fails to substantiate its main claims. Furthermore, it introduces a variety of additional components and complications, many of which are not ablated. This makes it hard to gain any generalizable insights from this work. For this reason, I would not recommend acceptance of the current version of the manuscript.

---

[1] Cohen et al. Certified Adversarial Robustness via Randomized Smoothing. ICML 2019
[2] Lyu et al. Adaptive Randomized Smoothing: Certified Adversarial Robustness for Multi-Step Defences. NeurIPS 2024
[3] Carlini et al. On Evaluating Adversarial Robustness. https://arxiv.org/abs/1902.06705
[4] Athalye et al.. Obfuscated Gradients Give a False Sense of Security: Circumventing Defenses to Adversarial Examples. ICML 2018.
[5] Nasr et al. The Attacker Moves Second: Stronger Adaptive Attacks Bypass Defenses Against Llm Jailbreaks and Prompt Injections. https://arxiv.org/abs/2510.09023

---

> ### Author Rebuttal · Authors · 2026-03-29
>
> Thank you, **Reviewer v96S** for the opportunity to address the **major weaknesses (W), minor comments (C) & Questions (Q)**.
>
> **W1/Q1 (Certified Robustness).** RS & ARS are certification frameworks. HySCAN is a stochastic base model certified under these same wrappers while applying implicit & explicit stochasticity into network. Thus, our claim is not that HySCAN outperforms RS/ARS as frameworks, but that—under such wrappers—a HySCAN-based pipeline surpasses alternative base models. Under such identical wrappers, hence, we compare various base models (Vanilla, RPF, CTRW, HySCAN). HySCAN achieves the best certified accuracy on CIFAR-10 (C10) / ImageNet (IN) for σ∈{0.25,0.5}, r∈{0.5,0.75}, with gains up to 17.1% over baselines **(see R3-Tab-1)**. This supports our design hypothesis: hybrid weight-space stochasticity (via RWAN) plus feature-space stochasticity (via SANI) outperforms prior base models with no or single-locus stochasticity. HySCAN replaces ResNet's conv. blocks; the residual skip is inherited from backbone. **Note: ARS basis results are in R3-Tab-1-1 at https://github.com/Epimone/Rebuttal-HySCAN**.
>
> **R3-Tab-1: base model vs base model comparison under same frameworks.**
> | Pipeline | (σ=.25) C10@.5 | C10@.75 | IN@.5 | IN@.75 | (σ=.5) C10@.50 | C10@.75 | IN@.5 | IN@.75 |
> | - | -: | -: | -: | -: | -: | -: | -: | -: |
> | Vanilla+RS | 43.4 | 26.1 | 0.0 | 0.0 | 41.3 | 32.4 | 36.8 | 28.7 |
> | RPF+RS | 51.6 | 33.9 | 38.5 | 26.7 | 50.3 | 38.8 | 47.4 | 37.5 |
> | CTRW+RS | 52.7 | 36.2 | 42.3 | 30.8 | 52.5 | 40.3 | 50.7 | 40.2 |
> | HySCAN+RS | 57.3 | 45.5 | 56.7 | 47.9 | 56.1 | 45.8 | 55.1 | 46.8 |
>
> **W2 (Empirical Robustness).** We have used EOT-PGD (EOT steps =7), **adaptive attacks: BPDA [3-5 (see Reviewer 2UE1)]** (ϵ=8/255, max steps=20, learning rate=0.5), & **BPDA+EOT [3-5 (see Reviewer 2UE1)]** (same settings of BPDA + EOT steps =3) on NIH-CXR & CIFAR-10. **HySCAN tops in R4-Tab-3 (see Reviewer 2UE1 W3 reply), gaining ≤ +9.5 pp over prior defenses.** Such results therefore strengthen the empirical-robustness evaluation against attacks that explicitly account for defense randomness. This supports our design hypothesis: RWAN injects attention noise aware weight-space stochasticity, SANI injects feature-space stochasticity, thereby distributing randomness across filters & representations instead of single locus.
>
> **W3/Q2 (Internal-noise ablation).** We ablated internal noise by setting $σ_w = 0$ while keeping HySCAN unchanged (**R3-Tab-2**). On CIFAR-10, disabling RWAN’s weight noise lowers certified accuracy by up to 11.8 pp for σ ∈ {0.25, 0.5}, & by 8.3–8.6 pp at r = 0.75. Thus, this noise in weight-space is essential to HySCAN’s certified gains. Empirical robustness shows the same trend: the RWAN/SANI ablation in **Table 9 (see paper)** drops PGD accuracy from 74.3 to 67.9 at σ = 0.5.
>
> **R3-Tab-2: Internal-noise $σ_w$ ablation**
> | RS (σ) | Variant | (r=0) | (r=0.25) | (r=0.5) | (r=0.75) | (r=1) | (r=1.25) | (r=1.5) | (r=2) |
> | ----------- | ------------------------------- | -------: | -------: | -------: | -------: | -------: | -------: | -------: | ------: |
> | 0.25 | HySCAN | 85.2 | 70.4 | 57.3 | 45.5 | 37.9 | 30.7 | 24.1 | 9.94 |
> | 0.25 | HySCAN w/ ($σ_w$=0) | 82.6 | 66.2 | 50.4 | 36.9 | 27.2 | 18.9 | 12.3 | 2.1 |
> | 0.5 | HySCAN | 80.3 | 66.4 | 56.1 | 45.8 | 38.3 | 31.7 | 25.1 | 13.9 |
> | 0.5 | HySCAN w/ ($σ_w$=0) | 77.1 | 62.6 | 50.2 | 37.5 | 29.8 | 22.7 | 16.5 | 6.2 |
>
> **W4 (Train obj. ablation).** We add a fixed-architecture objective ablation on CIFAR-10 by comparing κ=0 & κ>0 in Eq. 30. The added control indicates that the proposed objective changes certified accuracy **(CA)** only marginally, while yielding a modest gain in empirical robustness **(see R3-Tab-3)**.
>
> **R3-Tab-3: Training objective ablation**
> | Objective | CA@(σ=0.25, r=0.75) | CA@(σ=0.50, r=0.75) | PGD-20@(8/255) |
> | - | -: | -: | -: |
> | $\kappa=0$ (RS-compatible) | 45.5 | 45.8 | 74.3 |
> | $\kappa>0$ | 45.2 | 45.4 | 75.1 |
>
> **W5 (Theory).** Prop. 3.4 & Lem. 3.5/Prop. 3.6 formalize that HySCAN’s internal randomness is structured, not arbitrary: Prop. 3.4 shows RWAN as inducing attention noise aware weight perturbations instead of indiscriminate noise injection while Lem. 3.5 & Prop. 3.6 show that SANI fuses its two branches through a normalized, nonnegative, scale-controlled mixture whose norm remains bounded by that of the larger branch increment. Together, these support HySCAN as controlled stochastic smoothing in both weight & feature space.
>
> **W6 (Limitations).** HySCAN serves as a plug-and-play drop-in replacement for standard conv. blocks. However, its attention-noise–aware implicit & explicit stochasticity—resampled on each forward pass during both train & infer—introduces non-trivial cost in param, FLOPs, memory, & runtime.
>
> **C1/Q3 (Prop 3.2) — see Reviewer DbTh’s W3 response.**
>
> **Other (C2-C4) could not be included here due to space constraints & will be integrated, along with the above revisions, in the final version.**

---

> > ### Author Rebuttal · Reviewer_v96S · 2026-04-01
> >
> > Thank you for the rebuttal! This addresses most of my concerns (missing ablations, lack of adaptive attacks, distinction between randomized smoothing framework and probabilistic base models). **I have accordingly increased my score by 2 points**.
> >
> > It would be great if you could update the method names in all tables as you did here (i.e. Method + RS), since this disentangle the base model and the certification framework.
> >
> > My main concern remains that the connection between the theory and actual method is not clear and detracts from the narrative flow. For example, **why** any of the following statements from the rebuttal matter / how they contribute to improved robustness-utility trade-off is still not clear to me:
> > "normalized, nonnegative, scale-controlled mixture whose norm remains bounded by that of the larger branch increment. Together, these support HySCAN as controlled stochastic smoothing in both weight & feature space."
> >
> > But since that is, to some extend, a stylistic choice that cannot be addressed during the rebuttal, there is no need to further belabor this point during the discussion period.

---

> > > ### Author Response · Authors · 2026-04-03
> > >
> > > Thank you so much, **Reviewer v96S**, for this helpful point and for **raising your score to 4 (Weak Accept).** We agree and will revise the paper accordingly. We will also update all certification tables to use the “Method + RS” naming convention, consistent with the rebuttal, so that the stochastic base model and the certification framework are clearly separated.
> > >
> > > We clarify that Lemma 3.5 and Proposition 3.6 show that SANI’s dual-branch fusion design helps prevent the robustness–utility trade-off from deteriorating by keeping the injected feature perturbation controlled rather than excessive, thereby avoiding degradation in both empirical and certified robustness while preserving clean accuracy **(see Table R3-Tab1-1)**.
> > >
> > > More specifically, HySCAN introduces selective, scale-controlled stochasticity in both the weight and feature spaces. Within SANI, RecRAN and NoRAN are combined in a normalized manner, enabling the model to inject useful stochasticity into the network without over-perturbing the representation in a way that would otherwise degrades this trade-off and clean accuracy. RecRAN contributes progressive smoothing and stability, whereas NoRAN introduces diverse, channel-aware perturbations. If the two branches were naively summed, the resulting perturbation could become unnecessarily large and degrades the robustness–utility trade-off. Instead, SANI fuses the RecRAN and NoRAN modules using nonnegative normalized weights, i.e., a normalized nonnegative mixture, which bounds the resulting perturbation by the stronger branch rather than by the sum of the two branches. In other words, this fusion scheme is designed to keep the fused perturbation from growing like an unnormalized dual-branch sum. This allows HySCAN to benefit from both stability and diversity and helps reduce the risk of over-perturbation that could otherwise degrades clean accuracy, empirical robustness, and certified robustness, thereby improving robustness across certified and empirical regimes while preserving utility better than unstructured noise injection. Moreover, the same attention-noise learner also modulates RWAN, so this selective control extends to both feature-space and weight-space stochasticity. As a result, this design improves attack resistance through bounded and selective stochasticity, without injecting excessive noise that would otherwise undermine the robustness–utility trade-off and clean accuracy.
> > >
> > > **Table R3-Tab1-1: Fusion-rule ablation within SANI on CIFAR-10, with both RecRAN and NoRAN enabled. We compare a naive unnormalized sum with our normalized fusion while keeping the remaining HySCAN setting unchanged. The normalized fusion yields higher clean accuracy, certified accuracy at r=0.75, and PGD-20 robust accuracy at 8/255, supporting the claim that scale-controlled fusion helps avoid over-perturbation and better preserves the robustness–utility trade-off.**
> > >
> > > | SANI fusion rule          | Clean accuracy | Certified acc. @ (r=0.75) | PGD-20 @ (8/255) |
> > > | - | -: | -: | -: |
> > > | Naive unnormalized sum    | 82.15 |             41.74 |    71.10 |
> > > | Normalized fusion (Ours) | **90.93** |                  **45.52** |         **74.33** |

---

### Official Review · Reviewer_6HY3 · 2026-03-09

**Soundness:** 3
**Presentation:** 3
**Significance:** 2
**Originality:** 3
**Overall Recommendation:** 4
**Confidence:** 3

**Summary:**

The paper proposes a method to improve the provable robustness and empirical robustness, so that the gap between these can be reduced. This method utilizes two attention-modulated stochastic modules, RWAN and SANI, to smooth both parameters and representations in the models. Authors provide sound motivations and reasoning to support their designs. The experiments establish the improvements in robustness on natural and medical datasets.

**Compliance With Llm Reviewing Policy:**

Affirmed.

**Final Justification:**

After the rebuttal, both the initial and subsequent concerns have been adequately addressed, leading me to change my recommendation from a weak rejection to full acceptance. Therefore, I revise my score to 4 and recommend accepting the paper.

**Key Questions For Authors:**

Q1. Are there differences if the classifier and HySCAN module are trained from scratch?
Q2. What is the training time compared to other methods?
Q3. Since HySCAN needs adversarial training, how many iterations (steps) of PGD attack are used during training (the value of K in Algorithm 3)?
Q4. What is the training/inference time on the ImageNet dataset?

**Limitations:**

The evaluations only include limited attacks to demonstrate its effectiveness while stronger attacks are not verified (e.g., PGD+EOT is not included).

**Strengths And Weaknesses:**

strengths:
1. The method is with strong motivations, careful designs, and supportive theorems. The presentation is clear for readers to understand the proposed method.
2. The experiments show the effectiveness of the proposed method across natural and medical datasets, even if the experiments only include limited attacks.
3. The proposed method reduces the gap between provable robustness and empirical robustness, as claimed.

weaknesses:
1. The authors show the performance gain against APGD–20 and AA-20. However, according to [1,2,3,4], when a model includes randomness as a defence scheme, Expectation over Transformation (EOT) should be used to compute the gradient and to attack to avoid overestimating robustness. PGD-EOT or stronger attacks should be included in evaluations.
2. Proposition 3.2 is not referenced in the main text but is referenced in the Appendix. To make the paragraph more connected, the authors should either put Proposition 3.2 in the Appendix or at least use one or two sentences to describe it in the main text.

[1] Athalye, Anish, et al. "Synthesizing robust adversarial examples." International conference on machine learning. PMLR, 2018.
[2] Athalye, Anish, Nicholas Carlini, and David Wagner. "Obfuscated gradients give a false sense of security: Circumventing defenses to adversarial examples." International conference on machine learning. PMLR, 2018.
[3] Croce, Francesco, et al. "Evaluating the adversarial robustness of adaptive test-time defenses." International Conference on Machine Learning. PMLR, 2022.
[4] Lee, Minjong, and Dongwoo Kim. "Robust evaluation of diffusion-based adversarial purification." Proceedings of the IEEE/CVF International Conference on Computer Vision. 2023.

---

> ### Author Rebuttal · Authors · 2026-03-28
>
> Thank you, Reviewer **6HY3** for the opportunity to address the **weaknesses (W)**, **Questions (Q)**, and **Limitations (L)**.
>
> **W1/L1 (PGD-EOT or Stronger Attacks).** We evaluate our HySCAN against EOT-PGD & stronger attacks i.e., adaptive attacks—BPDA, & BPDA+EOT [2-4], where HySCAN remains leading adversarial robustness, improving over prior baselines: HyCAS [1], DCS, RPF, CTRW by up to 9.5% **(see Table R2-Tab-1)**.
>
> **Table R2-Tab-1: Empirical robustness (%) under EOT-PGD (EP) & adaptive attacks (BPDA and BPDA+EOT (BE)) on NIH-CXR (NC) & CIFAR-10 (C10).**
>
> | Pipeline | NC@EP | NC@BPDA | NC@BE | C10@EP |C10@BPDA | C10@BE |
> | - | -: | -: | -: | -: | -: | -: |
> | HyCAS | 83.6 | 82.2 | 78.3 | 68.4 | 68.1 | 66.5 |
> | DCS | 81.1 | 76.7 | 74.9 | 69.8 | 69.3 | 67.6 |
> | RPF | 82.2 | 77.3 | 75.3 | 66.1 | 67.9 | 65.4 |
> | CTRW | 81.1 | 76.7 | 74.9 | 68.8 | 69.4 | 67.4 |
> | **HySCAN** | **88.3** | **85.3** | **84.4** | **73.2** | **73.7** | **72.9** |
>
> **W2. (Proposition 3.2).** Thank you for this helpful suggestion. We agree that Proposition 3.2 should be explicitly connected in the main text. Rather than moving it to the Appendix, we will keep it in Sec. 3.1 and add the following sentence before Proposition 3.2: “Proposition 3.2 explains how HySCAN’s resampled internal randomness can make empirical gradient-based attacks less stable and, consequently, less effective by causing attack gradients to vary in direction across stochastic passes.”
>
> **Q1. (Differences between Classifier and HySCAN).** In our study, HySCAN is not trained as a separate post-hoc module; it replaces the backbone’s convolutional blocks, and the entire model is optimized jointly end-to-end. Hence, the relevant comparison is the plain backbone versus the HySCAN-integrated backbone under the same training setting. As shown in Table 9(a) (main paper's Appendix), on CIFAR-10 (σ=0.5), enabling RWAN+SANI improves certified accuracy at r=0.75 from 26.1 to 45.5 and PGD-20 robust accuracy from 59.8 to 74.3.
>
>
>
> **Q2 (Training time comparison).** For the training and inference time comparison **(see Table R2-Tab-2)**, we observe that HySCAN is a high-latency, computationally intensive pipeline. However, HySCAN’s objective is not to optimize efficiency but to improve adversarial robustness significantly for security-sensitive AI applications (e.g., clinical decision support), which often require substantial robustness that outweighs modest cost savings. The practical value of a defense method therefore depends on whether it delivers substantial robustness gains, even at high computational cost. Under this design choice, HySCAN outperforms baselines by up to **10.44%** **(see Tables R1-Tab-1–R1-Tab-6 at https://github.com/Epimone/Rebuttal-HySCAN
>  and Table R2-Tab-1)** across certified and empirical regimes, including **adaptive attacks**.
>
> **Table R2-Tab-2: Training and inference time comparison on CIFAR-10.**
>
> | Method | Training time (min) | Inference time (ms) |
> | - | -: | -: |
> | RPF | 185.8 | 0.65 |
> | CTRW | 192.3 | 0.74 |
> | DCS | 225.8 | 0.98 |
> | **HySCAN** | **571.4** | **4.29** |
>
> **Q3. (Steps of PGD attack).** Thank you for pointing this out. For adversarial training, HySCAN uses 10-step PGD to generate adversarial examples.
>
> **Q4. (Training & Inference time).** On ImageNet dataset, HySCAN takes training time including adversarial training: 8912.5 min and inference time: 15.7 ms.
>
> **[1] Dhar, el. al. Certified vs. Empirical Adversarial Robustness via Hybrid Convolutions with Attention Stochasticity, ICLR, 2026**
>
> **[2] Athalye et al. Obfuscated gradients give a false sense of security: Circumventing defenses to adversarial examples, ICML, 2018**
>
> **[3] Tramer et al. On adaptive attacks to adversarial example defenses, NeurIPS, 2020.**
>
> **[4] Hill et al. Stochastic security: Adversarial defense using long-run dynamics of energy-based models, ICLR, 2021.**

---

> > ### Author Rebuttal · Reviewer_6HY3 · 2026-04-02
> >
> > Without full knowledge of the setting of attacks, it would be diffcult to judge the effectiveness of the proposed method. Therefore, what's the hyperparameter of EOT-PGD (i.e., number of PGD steps and number of EOT trials)? It would be more informative for readers to understanding the strength of the proposed method. Other than that the authors have resolved most of my questions and concerns.

---

> > > ### Author Response · Authors · 2026-04-02
> > >
> > > Thank you very much, **Reviewer 6HY3**, for your kind follow-up and for acknowledging that we have addressed most of your concerns and for **raising your score to 4 (Weak Accept).** We apologize for not specifying the EOT-PGD hyperparameters. In our evaluation, we used **EOT-PGD with 20 PGD steps and 7 EOT trials**. We agree that these details are important for assessing the attack’s strength, and we will include them in the final version accordingly.
> > >
> > > In our evaluation, EOT-PGD was run with 20 PGD steps and 7 EOT trials. To clarify that this EOT trial is a sufficiently strong and conservative choice, we additionally fixed the PGD attack to 20 steps and varied the number of EOT trials from 1 to 10 on CIFAR-10. The corresponding robust accuracies of HySCAN were 74.63, 74.27, 74.90, 75.38, 73.75, 73.75, 73.24, 76.10, 75.82, and 75.38, respectively (see R2-Table-1-1).
> > >
> > > Importantly, the lowest robust accuracy in this sweep is obtained at 7 EOT trials (73.24%), which is the value reported as 73.2% in Table R2-Tab-1 after rounding. Therefore, our use of 7 EOT trials is not a favorable setting for HySCAN; rather, it is a conservative choice within the tested range, since larger trial counts (8–10) did not produce a stronger attack in this evaluation.
> > >
> > > We selected 7 EOT trials for two reasons: (i) it follows the prior DCS evaluation protocol [1], enabling a fair comparison with stochastic-defense baselines, and (ii) in our additional 1–10 trial sweep, it gives the strongest observed EOT-PGD attack against HySCAN. We agree that these details are important, and we will include the EOT-PGD hyperparameters and this analysis in the final version for clarity.
> > >
> > > **Table R2-Table-1-1: Robust accuracy (%) of HySCAN under EOT-PGD with 20 PGD steps on CIFAR-10, while varying the number of EOT trials from 1 to 10. This result indicates that our HySCAN remains robust even under multi-step EOT-PGD attacks.**
> > >
> > > | EOT Steps | Empirical Robustness (in Accuracy %)
> > > | - | -: |
> > > | 0 | 90.93 |
> > > | 1 | 74.63 |
> > > | 2 | 74.27 |
> > > | 3 | 74.90 |
> > > | 4 | 75.38 |
> > > | 5 | 73.75 |
> > > | 6 | 73.75 |
> > > | 7 | 73.24 |
> > > | 8 | 76.10 |
> > > | 9 | 75.82 |
> > > | 10 | 75.38 |
> > >
> > >
> > >
> > >
> > > [1]. Ma et al. Adversarial Robustness via Deformable Convolution with Stochasticity, ICML, 2025.

---

### Official Review · Reviewer_DbTh · 2026-03-12

**Soundness:** 2
**Presentation:** 3
**Significance:** 3
**Originality:** 2
**Overall Recommendation:** 3
**Confidence:** 4

**Summary:**

This paper proposed HySCAN, a hybrid randomized defense that injects stochasticity at two levels: implicit weight-space randomness via Random Weights with Attention Noise (RWAN), and explicit feature-space randomness via stochastic attention noise injection (SANI). The authors claim that this method simultaneously improves certified $\ell_2$ robustness under randomized smoothing and empirical robustness against $\ell_\infty$ attcks, while maintaining good clean accuracy across both general image and medical image bencgmarks.

**Compliance With Llm Reviewing Policy:**

Affirmed.

**Ethical Review Concerns:**

Impact Statement is missing.

**Final Justification:**

I don't think the authors have resolved my concerns regarding comparison to HyCAS. The datasets, baselines, evaluation protocols, attack methods, training procedures, backbone architectures, and certification methods all appear to be identical to those used in HyCAS, yet the authors do not discuss this in the submission, which I find concerning. The key difference is that HySCAN adopts weight-space and feature-space perturbations (RWAN+SANI), whereas HyCAS is built on a Lipschitz-controlled architecture. While this is a substantive modification, it seems incremental rather than conceptually original. Therefore I maintain my original score of 3.

**Key Questions For Authors:**

See Weaknesses.

**Limitations:**

Impact Statement or limitations discussion is missing.

**Strengths And Weaknesses:**

Strengths:

The gap between certified $\ell_2$ robustness and empirical $\ell_\infty$ robustness is a well known limitation of randomized smoothing. Many defenses help either certification of empirical robustness, but not both. This paper is well motivated by this issue. The proposed RWAN+SANI combination is more effective than simply adding noise at one location.

The empirical results are strong and thorough. On certified robustness, HySAN achieves best results on $\sigma=0.25,0.5$. The empirical robustness also achieves better results over prior defenses across medical benchmarks. And the ablation is informative, for example, Table 9 shows that RWAN and SANI contribute independently  and the combination leads to improvement on both certified and empirical robustness.

Weaknesses:

- On the certified robustness experiments, the authors use small $\sigma=0.25,0.5$, however, this is not enough to claim the improvement of certified robustness. As many baselines did, the comparisons of $\sigma=1,1.5$ are needed.

- It is well known that RS based method are computationally high, so a comparison of computational cost with previous core baselines are needed to address the computation-performance trade off. If vanilla in Table 8 refers to standard RS, then HySAN has 47 $\times$ FLOPs increase and ~6 $\times$ memory and inference time on ResNet50/ImageNet. This is not negligible.

- Theory for empirical robustness is limited. If I understand correctly, Proposition 3.2 is an intuitive gradient dispersion bound, not a robustness guarantee. The authors argued that HySAN increases directional gradient dispersion under resampled internal randomness, but they do not evaluate whether their specific noise distributioon indeed defends adaptive attackers in practice. APGD and AutoAttack already incorporate restarts and step size adaptation, but they do not specifically exploit HySCAN's attention gated noise. A more carefully designed adaptive attack, e.g. one that accounts for the distribution of $(w, \psi)$ during gradient estimation, should be discussed.

- While the paper presents HySCAN as a new hybrid robustness framework, its contribution over HyCAS [1] is limited. The two works share essentially the same high level motivation: narrowing the gap between certified $\ell_2$ and empirical $\ell_\infty$ robustness using internal stochasticity. The main difference is that HySCAN change the stochastic design to weight-space and feature-space perturbations (RWAN+SANI), whereas HyCAS uses a Lipschitz controlled architecture. This is a meaningful design variation, but it feels incremental rather than conceptually new. Moreover, HyCAS seems to offer more distinctive theoretical contribution via Lipschitz based certificate, while HySCAN mostly inherits the standard RS under joint randomness. In addition, the datasets, baselines, evaluation protocols, attack methods, training recipes, backbone choices, certification procedures are the same, so a clear discussion/comparison with HyCAS should be addressed. Overall, I think HySCAN's main contribution is an empirical and architectural refinement over HyCAS, rather than a clearly new conceptual/theoretical advance.

[1] Joy Dhar, el. al. Certified vs. Empirical Adversarial Robustness via Hybrid Convolutions with Attention Stochasticity.  International Conference on Learning Representations (ICLR) 2026

---

> ### Author Rebuttal · Authors · 2026-03-27
>
> Thank you, Reviewer **DbTh** for the opportunity to address the weaknesses (W).
>
> **Note: Tables R1-Tab-1–R1-Tab-6 are available on https://github.com/Epimone/Rebuttal-HySCAN due to space constraints.**
>
> **W1 (σ evaluation).** In **Tables R1-Tab-1–R1-Tab-2 (available at above link)** for each σ = 1.0, 1.5, HySCAN beats HyCAS [1] by up to 4.8%, confirming robustness gain comes from its dual attention-noise aware weight & feature spaces stochasticity.
>
> **Table R1-Tab-1: Certified accuracy (%) on ImageNet.**
> | Method | σ | r=0 | r=0.25 | r=0.5 | r=0.75 | r=1 | r=1.25 | r=1.5 | r=2 |
> |:--|:--:|--:|--:|--:|--:|--:|--:|--:|--:|
> | HyCAS | 0.25 | 72.3 | 63.9 | 55.6 | 46.4 | 40.7 | 35.2 | 29.7 | 5.42 |
> | HyCAS | 0.5 | 69.2 | 60.6 | 53.9 | 45.6 | 41.1 | 36.3 | 32.7 | 24.8 |
> | HyCAS | 1 | 53.8 | 50.4 | 47.5 | 39.2 | 36.6 | 32.3 | 29.1 | 25.3 |
> | HyCAS | 1.5 | 39.2 | 37.9 | 34.9 | 32.8 | 30.1 | 28.7 | 27.5 | 26.4 |
> | Ours | 0.25 | 71.9 | 65.2 | 56.7 | 47.9 | 42.3 | 37.1 | 30.7 | 7.15 |
> | Ours | 0.5 | 68.9 | 61.5 | 55.1 | 46.8 | 42.3 | 37.4 | 33.9 | 26.2 |
> | Ours | 1 | 53.5 | 50.4 | 47.9 | 40.7 | 38.1 | 34.7 | 31.5 | 28.2 |
> | Ours | 1.5 | 39.2 | 38.3 | 35.8 | 33.7 | 31.2 | 30.1 | 28.9 | 28.1 |
>
> **W2 (efficiency–robustness trade-off).** HySCAN achieves leading certified and empirical robustness, beating baselines by up to 10.44% **(see R1-Tab-3, available at above link)**. These gains come with higher computational cost. Our objective in HySCAN is not to optimize efficiency, but to improve adversarial robustness in security-sensitive AI applications (e.g., clinical decision support), where robustness is often more important than savings such costs.  Because practical value of a defense method depends on whether it provides substantial robustness gains, even at high computational cost.
>
> **W3 (Theorem & Adaptive Attacks).** We agree that Proposition 3.2 is a mechanistic gradient-dispersion statement, not a empirical robustness guarantee. We add **Theorem 1** as empirical ℓ∞ robustness guarantee. We evaluate HySCAN against EOT-PGD & adaptive attacks **[3-5; see Reviewer 2UE1]** to achieve gains by up to 9.5% **(see Reviewer 6HY3 W1's R2-Tab-1)**.
>
> **Theorem 1 (Empirical robustness of HySCAN).** Let f_Θ(·; U) = HySCAN base classifier, where U = (ω, ψ)~ν is joint internal randomness, & let ℓ be loss objective. For x, y, define worst-case adversarial loss
>
> $$
> L_{x,y}^{adv}(\Theta)=
> \sup_{\delta \in {B_\infty(\epsilon_\infty)}} E_{U \sim \nu} (\ell (f_{\Theta} (x+ \delta; U), y) ).
> $$
>
> Let the corresponding randomized prediction be
>
> $$
> \hat y_{\Theta}(x+\delta;U) =\arg\max_k f_{\Theta,k}(x+\delta;U).
> $$
>
> Then adversarial loss upper-bounds worst-case empirical misclassification probability
>
> $$
> \sup_{\delta\in B_\infty(\epsilon_\infty)}
> P_{U}\left[\hat y_\Theta(x+\delta;U)\neq y\right]
> \;\le\;
> \frac{L^{adv}_{x,y}(\Theta)}{\log 2}.
> $$
>
> To further control such loss, define stochastic loss
>
> $$ L_{x,y}(\delta) = E_{U \sim \nu}(\ell (f_\Theta(x+\delta;U),y ) ) $$
>
> & its worst-case local $(\ell_1)$ sensitivity
>
> $$
> G_\infty(x,y)
> :=
> \sup_{\delta\in B_\infty(\epsilon_\infty)}
> \left\|\nabla_\delta L_{x,y}(\delta)\right\|_1,
> $$
>
> assuming $L_{x,y}$ is differentiable on $B_\infty(\epsilon_\infty)$. Then adversarial loss satisfies
>
> $$
> L_{x,y}^{adv}(\Theta)
> \le
> L_{x,y}(0)
> +
> \epsilon_\infty , G_\infty (x, y).
> $$
>
> Combining two bounds yields
>
> $$
> \sup_{\delta\in B_\infty(\epsilon_\infty)}
> P_{U}\left[\hat y_\Theta(x+\delta;U)\neq y\right]
> \le
> \frac{L_{x,y}(0)+\epsilon_\infty\,G_\infty(x,y)}{\log 2}.
> $$
>
> Hence, HySCAN ensures an empirical $(\ell_\infty)$ robustness guarantee.
>
> **W4 (HyCAS vs HySCAN).** HyCAS is indeed closest prior work, however, we respectfully disagree that HySCAN is an incremental variant.
>
> *Conceptual distinction.* HyCAS studies how far deterministic Lip. control can be combined with additional randomness: it couples a Lip.-constrained deterministic control with stochastic modules. HySCAN, in contrast, studies whether attention-conditioned stochasticity across two complementary spaces (implicit attention-noise-conditioned weight-space perturbations via RWAN & explicit attention-modulated feature-space stochasticity via SANI) can provide an alternative route to robustness within backbones. Thus, our claim is not that HySCAN replaces HyCAS’s Lip. theory, but that it contributes a different conceptual route: rather than extending deterministic Lip. control plus stochastic smoothing, instead HySCAN studies hybrid space-aware stochasticity across both spaces.
>
> *Comparison.* With identical datasets & protocols, HySCAN achieves robustness gains by up to 4.8% (on certified) **(R1-Tab-1 - R1-Tab-2)** & by up to 5.11% (empirical) **(R1-Tab-4 - R1-Tab-6)** on average over HyCAS. We attribute this gain to HySCAN’s joint, dual attention-conditioned randomization across both spaces, which appears harder for stronger attacks to exploit than HyCAS’s single-space deterministic-stochastic design.
>
> **Limitations. See Reviewer v96S W6 response.**

---

> > ### Author Rebuttal · Reviewer_DbTh · 2026-04-02
> >
> > Thanks the reviewer for the additional results. However, my concerns regarding W4 remains. The datasets, baselines, evaluation protocols, attack methods, training recipes, backbone choices, certification procedures are the same as HyCAS, and the authors omit discussing this in their submission pdf, which looks suspicious to me. The authors should clearly state their contribution when comparing to HyCAS.

---

> > > ### Author Response · Authors · 2026-04-04
> > >
> > > We thank Reviewer for pointing out HyCAS. **HyCAS accepted publicly by ICLR on Jan. 26, 2026 while our ICML submission was made on Jan. 28, 2026, which did not allow sufficient time to perform a careful implementation/controlled empirical comparison before submission**
> > >
> > > **That said, our main claim is not based on timing but on methodological distinction. We set new state-of-the-art defense (see Table R1-1-1 & Fig. R1-Fig-2)**
> > >
> > > **Table & above Fig. in https://github.com/Epimone/Rebuttal-HySCAN compares HyCAS vs HySCAN**
> > >
> > > **HyCAS is a deterministic Lipschitz-controlled multi-stream design (FDPAN+SNCAN+RPFAN with RANI, Fig R1-Fig-2(a)): weight-space stochasticity comes from RPFAN’s RPF, while feature-space stochasticity is injected separately by RANI.
> > > HySCAN differs in how these stochastic loci are organized/coupled. It introduces a **hybrid cross-space coupled stochastic** block inside network (Fig. R1-Fig-2(b))**: RWAN applies attention noise conditioned heteroscedastic perturbations in weight space & SANI injects attention-modulated perturbations in feature space. The key distinction is that these two sources of randomness are not independent design add-ons: they are coordinated via the same attention-noise mechanism A(u;ψ), rather than being combined only at the stream level. A(u;ψ) plays two complementary roles in HySCAN: per-layer, it produces the response that forms RWAN’s gate for weight perturbation; & per-block, it produces the additive perturbation used by SANI for feature injection. This is the core methodological difference we emphasize. In HyCAS, weight & feature-space noise are present, but they are not presented as an explicitly shared, attention-conditioned cross-space controller. In HySCAN, by contrast, weight-space & feature-space stochasticity are jointly coordinated via the same attention-noise learner, which is why we describe the method as a hybrid space-aware defense.
> > >
> > > Unlike RANI, SANI ≠ a single noise injector. It is dual-branch attention-noise learner (RecRAN+NoRAN) whose normalized fusion ensures that the perturbation remains scale-controlled & does not grow uncontrollably as in an unnormalized dual-branch sum. This design allows HySCAN to benefit from progressive smoothing while reducing over-perturbation, which would otherwise degrade clean accuracy & the robustness-utility trade-off.
> > >
> > > This cross-space coupling is the key reason HySCAN gains robustness in both regimes **(see Tables R2-Tab-1 & R1-Tab-1–R1-Tab-2 (rebuttal)**. RWAN perturbs how features are computed, while SANI perturbs what intermediate representations look like & the shared attention-noise learner aligns these two effects without injecting uncontrolled noise. Ablations confirm the benefit: RWAN-only & SANI-only beat the no-HySCAN baseline, but the RWAN+SANI design is the best; similarly, RecRAN+NoRAN outperforms either branch alone. Additional adaptive-attack results further support that this benefit is not from generic randomness alone but from coordinated stochastic design that remains leading adversarial robustness under EOT-PGD, BPDA, & BPDA+EOT **(Reviewer 6HY3 W1's R2-Tab-1)**.
> > > HyCAS is primarily framed around a Lipschitz margin certificate while HySCAN retains standard randomized smoothing under joint Gaussian noise/our internal randomness. Thus, we claim HySCAN introduces an attention-driven hybrid cross-space coupling between weight- & feature-space stochasticity.
> > >
> > > **Table R1-1-1: Architectural differences between HyCAS & HySCAN**
> > > | Contribution | HyCAS | HySCAN | HySCAN novelty
> > > | - | -: |  -: | -: |
> > > | **Design** | Hybrid defense built on **Lipschitz-controlled multi-stream design (FDPAN+SNCAN+RPFAN+RANI)** that integrates filter & feature space randomization | Hybrid space defense built on a **hybrid cross-space-coupled stochastic plug-in block (RWAN→SANI)** | Hybrid cross-space coupling of stochasticity, rather than multi-stream fusion with randomness injected at multiple points |
> > > | **weight-space randomization** | RPFAN introduces weight-space stochasticity via random-projection filters, without an attention-controlled noise learner | **RWAN** applies **attention-gated, input-conditioned, heteroscedastic perturbations to convolution weights** | Attention noise conditioned weight perturbation, rather than RPF without attention-based noise control |
> > > | **Feature-space stochasticity** | **RANI** injects bounded attention-noise residuals into features | **SANI** injects explicit feature perturbations via **RecRAN + NoRAN** with normalized fusion | Dual-branch, scale-controlled feature-space stochasticity for a better robustness–accuracy trade-off **(see Reviewer v96S, Table R3-Tab1-1)** |
> > > | **Cross-space coupling** | Filter-space randomness & feature-space noise coexist, but are **not governed by a shared attention-noise controller.** | The **same attention-noise learner** is used for **RWAN gating** & **SANI injection** | **Attention-noise-driven cross-space coupling** between the two stochastic spaces. |

---

### Decision · Program_Chairs · 2026-04-30

**Decision:**

Accept (regular)

**Comment:**

The AC has read all the reviews and the authors' responses. The rebuttal addresses many of the reviewers' concerns. After the rebuttal, Reviewer DbTh still had concerns about the difference between this work and HyCAS (published at ICLR 2026) and maintained a weak reject. The authors provided a detailed and solid comparison between the two works, which seems acceptable. Other reviewers acknowledged that many of their concerns have been addressed and recommended a weak acceptance. However, they also noted that some issues remain unresolved, including the unclear connection between the theory and the actual method, and the somewhat limited evaluation. Considering all the reviews and the authors' responses, the AC recommends a weak accept, and a more thorough revision is required to resolve all the concerns.